# THE ROLE OF LABEL NOISE IN THE FEATURE LEARNING PROCESS

## ABSTRACT

Deep learning with noisy labels presents significant challenges. In this work, we theoretically characterize the role of label noise in training neural networks from a feature learning perspective. Specifically, we consider a *signal-noise* data distribution, where each data point comprises a label-dependent signal and label-independent noise, and rigorously analyze the training dynamics of a two-layer convolutional neural network under this data setting, along with the presence of label noise. Particularly, we identify two stages in which the dynamics exhibit distinct patterns. In *Stage I*, the model perfectly fits all the clean samples (i.e., samples without label noise) while ignoring the noisy ones (i.e., samples with noisy labels). In the first stage, the model learns the signal from the clean samples, which generalizes well on unseen data. In *Stage II*, as the training loss converges, the gradient in the direction of noise surpasses that of the signal, leading to overfitting on noisy samples. Eventually, the model memorizes the noise present in the noisy samples, which degrades its generalization ability. In contrast, when training without label noise, the dynamics do not exhibit this two-stage pattern. Furthermore, our results provide theoretical supports for two widely used techniques for tackling label noise: early stopping and sample selection. Experiments on both synthetic and real-world datasets confirm our theoretical findings.

## 1 INTRODUCTION

One of the key challenges in deep learning lies in its susceptibility to label noise (Angluin & Laird, 1988). The success of deep learning stems from its exceptional ability to approximate arbitrary functions (Hornik et al., 1989; Funahashi, 1989), yet this ability becomes problematic in the presence of noisy labels. Over-parameterized neural networks, which have sufficient capacity to memorize training data, tend to over-fit noisy labels, leading to poor generalization on unseen data. Although many studies (Patrini et al., 2017; Ma et al., 2018; Yu et al., 2019; Tanaka et al., 2018; Han et al., 2018; Liu et al., 2023b; Chen et al., 2023; Xia et al., 2024) have developed methods to mitigate the effects of label noise in practice, our theoretical understanding of label noise remains limited. Particularly, a crucial step towards advancing our understanding is to formulate a comprehensive theory that explains the learning dynamics of neural networks in the presence of label noise.

Existing works (Li et al., 2020; Liu et al., 2023a) has attempted to theoretically analyze the effects of label noise on the training dynamics of neural networks; however, many of these studies rely on unrealistic assumptions. Li et al. (2020) assumed the lazy training regime, which constrains the distance between model weights and their initialization. Similarly, Liu et al. (2023a) adopted an infinitely-wide neural network, where the dynamics become linear and can be described by a static kernel function. Exploring the training dynamics with label noise beyond these assumptions remains an active area of research.

Recently, a new line of research (Allen-Zhu & Li, 2020; Frei et al., 2022; Cao et al., 2022; Kou et al., 2023; Xu et al., 2024) have developed the feature learning theory to better understand the training dynamics and generalization of neural networks under more realistic settings. The core idea of the feature learning theory, which dates back to Rumelhart et al. (1986), is to assume a simplified data distribution and analyze how neural networks learn useful *features*, or *representations*, of this data during the training process. Unlike the lazy training regime, this theoretical framework allows model weights to evolve over a larger distance, capturing a broader range of nonlinear behaviours in the training process. While the feature learning theory has successfully modeled the complex

dynamics of neural networks across various settings, few attempts have been made to characterize their dynamics in the presence of label noise using this theory. Here, we pose the question:

*How does label noise affect the training dynamics of neural networks?*

In this work, we answer the above question by establishing a *feature learning framework* for training neural networks with gradient descent in the presence of label noise, unveiling a two-stage behaviour in the training dynamics. Similar to Cao et al. (2022); Kou et al. (2023), the feature learning framework is based on a *signal-noise* data distribution, where each data point consists of a label-dependent signal and a label-independent noise[1]. Specifically, we consider a binary classification dataset $\{(\mathbf{x}_i, \tilde{y}_i)\}_{i=1}^n$, where $\mathbf{x}_i \in \mathbb{R}^{2d}$ is the $i$-th input and $y_i \in \{-1, 1\}$ is the corresponding ground-truth label. The input are defined as $\mathbf{x} = [\mathbf{x}^{(1)}, \mathbf{x}^{(2)}] = [y\boldsymbol{\mu}, \boldsymbol{\xi}]$, with the fixed vector $\boldsymbol{\mu} \in \mathbb{R}^d$ representing the *signal*, and the random vector $\boldsymbol{\xi} \in \mathbb{R}^d \sim \mathcal{N}(0, \sigma_\xi^2 \mathbf{I}_d)$ representing the *noise*. Additionally, we introduce the label noise in the sense that the observed label $\tilde{y}$ flips the true label $y$ (i.e., $\tilde{y} = -y$) with a certain probability. We rigorously analyze the training dynamics of a two-layer convolutional neural network under above setup, identifying a *two-stage* picture:

- *Stage I.* Initially, the model perfectly fits all the clean samples while ignoring the noisy ones[2]. The model mainly learns the signal from clean samples, which generalizes well on unseen data.

- *Stage II.* Then, the training loss converges and the model over-fits to the noisy samples. The model memorizes the noise features from noisy samples, degrading its generalization.

For comparison, we also show that when training without label noise, the model tends to fit all training samples throughout the training process, and the test error remains well-bounded. See Table 1 for a summary. This two-stage picture is well-supported by existing empirical findings (Arpit et al., 2017; Han et al., 2018), which show that neural networks tend to first learn simple patterns from clean samples and then proceed to memorize the noisy ones, and further validated by our experiments under both synthetic and real-world scenarios.

| | | |
|---|---|---|
| **w/ label noise** | *Stage I.* fit all clean samples and ignore noisy ones; learn signal. | Theorem 4.1 |
| | *Stage II.* over-fit noisy samples; learn noise; degrade generalization. | Theorem 4.2 |
| **w/o label noise** | the loss converges; fit all clean samples; generalize well. | Theorem 4.3 |

Table 1: Overview of the two-phase picture and corresponding theoretical results.

Subsequently, based on our theoretical analysis, we provide an explanation for the effectiveness of two common techniques used to address the label noise problem, i.e., *early stopping* (Liu et al., 2020; Bai et al., 2021) and *sample selection* (Han et al., 2018; 2020). i) For early stopping, we show that stopping the training process at the end of the first stage ensures a low test error, even in the presence of label noise. ii) For sample selection, we verify that the small-loss criterion[3] (Han et al., 2018) can provably identify the clean samples from noisy ones.

In summary, our work focuses on the role of label noise in the optimization process of neural networks. We prove that the models inevitably learn the noise features from data when the label noise is present, which leads to degraded generalization compared to noise-free training. Our results highlight the importance of taking a feature-learning viewpoint in studying questions related to training dynamics.

## 2 RELATED WORK

**Feature learning theory.** Recent works (Allen-Zhu & Li, 2020; Frei et al., 2022; Allen-Zhu & Li, 2022; Cao et al., 2022; Kou et al., 2023; Zou et al., 2023; Xu et al., 2024) has developed the feature learning theory to analyze the training dynamics and generalization of neural networks. The core idea of the feature learning theory is to simplify the data setup and describe how neural network learns useful features of this data during training. The feature learning theory has achieved great successes in understanding a large range of architecture such as graph neural network (Huang et al., 2023),

---

[1] Both signal and noise are "features" in data.

[2] To clarify, "clean samples" refers to samples without noisy labels, while "noisy samples" refers to those with noisy labels. The same terminology applies for the rest of this paper.

[3] The samples with small loss are more likely to be the ones which are correctly labeled, i.e., the clean samples.

convolutional neural network (Cao et al., 2022; Kou et al., 2023), vision transformer (Jelassi et al., 2022; Li et al., 2023), as well as various training schemes including gradient descent with momentum (Jelassi & Li, 2022), Adam (Zou et al., 2021), sharpness-aware minimization (Chen et al., 2024), and Mixup (Zou et al., 2023; Chidambaram et al., 2023).

**Theoretical advances on learning with label noise.** Existing works (Liu et al., 2023a; Li et al., 2020; Frei et al., 2021) have attempted to theoretically analyze the training dynamics of neural networks in the presence of label noise. However, many of these studies rely on unrealistic assumptions. Li et al. (2020); Liu et al. (2023a) assumed the lazy-training regime (Jacot et al., 2018; Chizat et al., 2019), constraining the distance between model weights and their initialization, and thus cannot capture the process of learning *features* of data (Rumelhart et al., 1986; Damian et al., 2022). On the other hand, Frei et al. (2021) focused on the early dynamics of neural networks trained with SGD, which can be largely explained by the behavior of a linear classifier (Kalimeris et al., 2019; Hu et al., 2020), thereby overlooking the adverse effects of label noise on generalization in later training stages. *In comparison*, our work characterizes the whole training process of neural networks with label noise from a feature-learning perspective, unveiling a novel two-stage behaviour in the training dynamics.

**Comparison with** Kou et al. (2023); Meng et al. (2023); Xu et al. (2024). Despite not specifically focusing on the label noise problem, these works analyze the feature learning process of neural networks while incorporating label noise in their data setups. However, their results differ significantly from ours. Meng et al. (2023); Xu et al. (2024) focused on specific XOR-type classifications tasks, whereas our analysis considers a more standard setting. Moreover, Kou et al. (2023) considered a setting where $n \cdot \mathrm{SNR}^2 = o(1)$, with $\mathrm{SNR} := \|\boldsymbol{\mu}\|_2/(\sigma_\xi \sqrt{d})$ representing the signal-to-noise ratio. In contrast, our two-stage picture can only be derived when $n \cdot \mathrm{SNR}^2 = \Theta(1)$.

## 3 PROBLEM SETUP

**Notation.** We use bold letters for vectors and matrices, and scalars otherwise. The Euclidean norm of a vector and spectral norm of a matrix are denoted by $\|\cdot\|_2$, and the Frobenius norm of a matrix by $\|\cdot\|_F$. We use $y$ to represent the true label and $\tilde{y}$ to represent the observed label. For a logical variable $a$, let $\mathbb{1}(a) = 1$ if $a$ is true, otherwise $\mathbb{1}(a) = 0$. Denote $\mathbf{I}_d$ as the $d \times d$ identity matrix and $[n] = \{1, 2, \ldots, n\}$.

**Training set.** Following Allen-Zhu & Li (2020); Cao et al. (2022); Kou et al. (2023); Xu et al. (2024); Huang et al. (2023), we consider a binary-classification data distribution $\mathcal{D}_{\mathrm{tr}}$, where each data point consists of a label-dependent *signal* and a label-independent *noise*. More precisely, let $\boldsymbol{\mu} \in \mathbb{R}^d$ be a fixed vector representing the signal and let $\boldsymbol{\xi} \in \mathbb{R}^d$ be a random vector sampled from the Gaussian distribution $\mathcal{N}(\mathbf{0}, \sigma_\xi^2 \mathbf{I}_d)$ representing the noise. Then each data point $\mathbf{x} \in \mathbb{R}^{2d}$ is defined as

$$\mathbf{x} = [\mathbf{x}^{(1)}, \mathbf{x}^{(2)}], \quad \text{where one of } \mathbf{x}^{(1)}, \mathbf{x}^{(2)} \text{ is } y\boldsymbol{\mu} \text{ and the other is } \boldsymbol{\xi},$$

where $y \in \{-1, 1\}$ is the corresponding label, generated from the Rademacher distribution, i.e., $\mathbb{P}(y = 1) = \mathbb{P}(y = -1) = 1/2$. We sample the training set $\{\mathbf{x}_i, y_i\}_{i=1}^n$ from $\mathcal{D}_{\mathrm{tr}}$. Let $\mathcal{S}_1 := \{i \in [n] : y_i = 1\}$ and $\mathcal{S}_{-1} := \{i \in [n] : y_i = -1\}$, we assume the training set is balanced without loss of generality, i.e., $|\mathcal{S}_1| = |\mathcal{S}_{-1}| = n/2$.

**Label noise.** In this work, we introduce the label noise for each training sample, where the observed label $\tilde{y}$ may differ from the ground-truth label $y$[4]. Two common label noise settings are: i) *random classification noise* (Angluin & Laird, 1988), where labels are flipped with probability $\tau$; and ii) *class-conditional noise*, where label flips depend on the class, with samples from some classes being more likely to be mislabeled than others. Our theoretical analysis covers both settings. Specifically, let $\tau_+, \tau_- \in (0, 1/2)$[5] represent the label flipping probabilities for samples from the positive and negative classes in the binary classification task, i.e., $\tau_+ = \mathbb{P}(\tilde{y} = -1|y = 1)$ and $\tau_- = \mathbb{P}(\tilde{y} = 1|y = -1)$, respectively. We consider both cases where $\tau_+ = \tau_-$ (i.e., the random classification noise) and $\tau_+ \neq \tau_-$ (i.e., the class-conditional noise). Additionally, we denote the clean sample set as $\mathcal{S}_t := \{i \in [n] : \tilde{y}_i = y_i\}$ and noisy sample set as $\mathcal{S}_f := \{i \in [n] : \tilde{y}_i \neq y_i\}$

---

[4]Notice that in experiments, the ground-truth labels are inaccessible, and we can only evaluate models based on the observed labels.

[5]The upper bound on $\tau_+$ and $\tau_-$ ensures that the neural network can still learn the signal from data despite the presence of label noise.

**Network.** We consider a two-layer convolutional neural network with ReLU activation. Formally, given the input data $\mathbf{x}$, the output of the neural network is defined as $f(\mathbf{W}, \mathbf{x}) = F_{+1}(\mathbf{W}_{+1}, \mathbf{x}) - F_{-1}(\mathbf{W}_{-1}, \mathbf{x})$, where $F_{+1}(\mathbf{W}_{+1}, \mathbf{x})$ and $F_{-1}(\mathbf{W}_{+1}, \mathbf{x})$ are given by

$$F_j(\mathbf{W}_j, \mathbf{x}) = \frac{1}{m}\sum_{r=1}^{m}\Big(\sigma\big(\langle\mathbf{w}_{j,r}, y\boldsymbol{\mu}\rangle\big) + \sigma\big(\langle\mathbf{w}_{j,r}, \boldsymbol{\xi}\rangle\big)\Big), \quad j = \pm 1.$$

Here, $\sigma(\cdot)$ is ReLU activation function, defined as $\sigma(x) = \max\{0, x\}$. We initialize the entries of $\mathbf{W}$ independently from a zero-mean Gaussian distribution with variance $\sigma_0^2$, i.e., $\mathbf{w}_{j,r}^{(0)} \sim \mathcal{N}(0, \sigma_0^2\mathbf{I}_d)$ for all $j = \pm 1, r \in [m]$. Additionally, we adopt the common practice of fixing the second layer weights to uniformly $\pm 1$ for simplifying analysis (Arora et al., 2019; Cao et al., 2022; Kou et al., 2023).

**Objective.** We employ the logistic loss $\ell(f, \tilde{y}) = \log\big(1 + \exp(-f \cdot \tilde{y})\big)$ for training neural networks. Then the training loss, or the empirical risk, can be written as:

$$L_S(\mathbf{W}) = \frac{1}{n}\sum_{i=1}^{n}\ell\big(f(\mathbf{W}, \mathbf{x}_i), \tilde{y}_i\big),$$

To minimize this empirical risk, we use gradient descent (GD) with a constant learning rate $\eta > 0$,

$$\mathbf{w}_{j,r}^{(t+1)} = \mathbf{w}_{j,r}^{(t)} - \eta \cdot \nabla_{\mathbf{w}_{j,r}}L_S(\mathbf{W}^{(t)})$$

$$= \mathbf{w}_{j,r}^{(t)} - \frac{\eta}{nm}\sum_{i=1}^{n}\ell_i'^{(t)}\sigma'(\langle\mathbf{w}_{j,r}^{(t)}, \boldsymbol{\xi}_i\rangle)j\tilde{y}_i\boldsymbol{\xi}_i - \frac{\eta}{nm}\sum_{i=1}^{n}\ell_i'^{(t)}\sigma'(\langle\mathbf{w}_{j,r}^{(t)}, y_i\boldsymbol{\mu}\rangle)jy_i\tilde{y}_i\boldsymbol{\mu}, \quad (1)$$

where we define the loss derivative as $\ell_i'^{(t)} := \ell'(f(\mathbf{W}^{(t)}, \mathbf{x}_i), \tilde{y}_i) = -\frac{1}{1+\exp(\tilde{y}_i f(\mathbf{W}^{(t)}, \mathbf{x}_i))}$.

**Genaralization.** We characterize the generalization abilities of models by evaluating the 0-1 error on the unseen data distribution $\mathcal{D}_{\text{test}}$:

$$L_D^{0-1}(\mathbf{W}) = \mathbb{P}_{(\mathbf{x},y)\sim\mathcal{D}_{\text{test}}}(y \cdot f(\mathbf{W}, \mathbf{x}) < 0)$$

The test distribution $\mathcal{D}_{\text{test}}$ mainly follows the settings of $\mathcal{D}_{\text{tr}}$; however, to simulate spurious features in real-world scenarios, for any $(\mathbf{x}, y) \in \mathcal{D}_{\text{test}}$, we define $\mathbf{x} = [y\boldsymbol{\mu}, \boldsymbol{\xi} + \boldsymbol{\zeta}]$, where $\boldsymbol{\xi} \sim \text{Unif}(\{\boldsymbol{\xi}_i\}_{i=1}^n)$ and $\boldsymbol{\zeta} \sim \mathcal{N}(0, \sigma_\xi^2\mathbf{I})$. Here, $\text{Unif}(\{\boldsymbol{\xi}_i\}_{i=1}^n)$ denotes the uniform distribution over $\{\boldsymbol{\xi}_i\}_{i=1}^n$. In real-world scenarios, spurious features exist, which occur in both the training and test set but lack causal relationships with the ground-truth label $y$, such as the "background" information in image classification tasks (Sagawa et al., 2020; Zhou et al., 2021; Singla & Feizi, 2021; Izmailov et al., 2022). Therefore, we consider label-independent noise $\boldsymbol{\xi}$ in the training set as spurious features and randomly incorporate them into the data points from the test distribution.

**Signal-noise decomposition.** In our analysis, we utilize a proof technique termed *signal-noise decomposition*, which has been widely adopted by (Li et al., 2019; Allen-Zhu & Li, 2020; 2022; Cao et al., 2022; Kou et al., 2023). The signal-noise decomposition breaks down the weight $\mathbf{w}_{j,r}^{(t)}$ into signal and noise components. Formally, we express:

$$\mathbf{w}_{j,r}^{(t)} = \mathbf{w}_{j,r}^{(0)} + j\gamma_{j,r}^{(t)}\|\boldsymbol{\mu}\|_2^{-2}\boldsymbol{\mu} + \sum_{i=1}^{n}\rho_{j,r,i}^{(t)}\|\boldsymbol{\xi}_i\|_2^{-2}\boldsymbol{\xi}_i, \quad (2)$$

where $\gamma_{j,r}^{(t)}$ and $\rho_{j,r,i}^{(t)}$ represent the signal and noise coefficients, respectively. The normalization factors $\|\boldsymbol{\mu}\|_2^{-2}$ and $\|\boldsymbol{\xi}_i\|_2^{-2}$ ensure that $\gamma_{j,r}^{(t)} \approx \langle\mathbf{w}_{j,r}^{(t)}, \boldsymbol{\mu}\rangle$, and $\rho_{j,r,i}^{(t)} \approx \langle\mathbf{w}_{j,r}^{(t)}, \boldsymbol{\xi}_i\rangle$. Naturally, $\gamma_{j,r}^{(t)}$ characterizes the process of signal learning, while $\rho_{j,r,i}^{(t)}$ captures the memorization of noise.

To facilitate a finer-grained analysis of the evolution of the noise coefficients, we introduce the notations $\overline{\rho}_{j,r,i}^{(t)} := \rho_{j,r,i}^{(t)}\mathbb{1}(\rho_{j,r,i}^{(t)} \geq 0)$, $\underline{\rho}_{j,r,i}^{(t)} := \rho_{j,r,i}^{(t)}\mathbb{1}(\rho_{j,r,i}^{(t)} \leq 0)$, following Cao et al. (2022). Consequently, the weight decomposition can be further expressed as:

$$\mathbf{w}_{j,r}^{(t)} = \mathbf{w}_{j,r}^{(0)} + j\gamma_{j,r}^{(t)}\|\boldsymbol{\mu}\|_2^{-2}\boldsymbol{\mu} + \sum_{i=1}^{n}\overline{\rho}_{j,r,i}^{(t)}\|\boldsymbol{\xi}_i\|_2^{-2}\boldsymbol{\xi}_i + \sum_{i=1}^{n}\underline{\rho}_{j,r,i}^{(t)}\|\boldsymbol{\xi}_i\|_2^{-2}\boldsymbol{\xi}_i. \quad (3)$$

Based on the gradient descent dynamics (1) and the signal-noise decomposition (3), we introduce the Lemma 3.1, which outlines the iterative update rules for the signal and noise coefficients.

**Lemma 3.1.** *The coefficients $\gamma_{j,r}^{(t)}, \overline{\rho}_{j,r,i}^{(t)}, \underline{\rho}_{j,r,i}^{(t)}$ in decomposition (3) satisfy $\gamma_{j,r}^{(0)}, \overline{\rho}_{j,r,i}^{(0)}, \underline{\rho}_{j,r,i}^{(0)} = 0$ and admit the following iterative update rule:*

$$\gamma_{j,r}^{(t+1)} = \gamma_{j,r}^{(t)} - \frac{\eta}{nm} \sum_{i=1}^{n} \ell_i'^{(t)} \sigma'(\langle \mathbf{w}_{j,r}^{(t)}, y_i \boldsymbol{\mu} \rangle) y_i \tilde{y}_i \|\boldsymbol{\mu}\|_2^2,$$

$$\overline{\rho}_{j,r,i}^{(t+1)} = \overline{\rho}_{j,r,i}^{(t)} - \frac{\eta}{nm} \ell_i'^{(t)} \sigma'(\langle \mathbf{w}_{j,r}^{(t)}, \boldsymbol{\xi}_i \rangle) \|\boldsymbol{\xi}_i\|_2^2 \mathbb{1}(\tilde{y}_i = j),$$

$$\underline{\rho}_{j,r,i}^{(t+1)} = \underline{\rho}_{j,r,i}^{(t)} + \frac{\eta}{nm} \ell_i'^{(t)} \sigma'(\langle \mathbf{w}_{j,r}^{(t)}, \boldsymbol{\xi}_i \rangle) \|\boldsymbol{\xi}_i\|_2^2 \mathbb{1}(\tilde{y}_i = -j).$$

Lemma 3.1 converts gradient descent updates into the dynamics of signal and noise coefficients $\gamma_{j,r}^{(t)}, \overline{\rho}_{j,r,i}^{(t)}, \underline{\rho}_{j,r,i}^{(t)}$, enabling us to analyze the complex, non-convex optimization of neural networks. Since $\ell_i'^{(t)} < 0$ and the upper bounds $\tau_+, \tau_- < 1/2$ hold, the signal coefficient $\gamma_{j,r}^{(t)} \geq 0$ forms an increasing sequence, even in the presence of noisy samples. Similarly, we can easily verify that $\overline{\rho}_{j,r,i}^{(t)} \geq 0$ also increases over time, while $\underline{\rho}_{j,r,i}^{(t)} \leq 0$ decreases but remains bounded below by a small term. Therefore, it suffices to focus on the dynamics of $\overline{\rho}_{j,r,i}^{(t)}$, which is critical for understanding noise memorization in the training process.

## 4 MAIN RESULTS

In this section, we present our main result that a two-stage behaviour emerges in the feature learning process of neural networks in the presence of label noise. For comparison, we also analyze the feature learning process when training without label noise, emphasizing the stark contrasts with the two-stage behavior identified with label noise. A summary of our results is shown in Table 1.

Before presenting our main results, we first state our main condition.

**Additional notations.** We denote $\mathrm{SNR} := \|\boldsymbol{\mu}\|_2 / (\sigma_\xi \sqrt{d})$ to be the signal-to-noise ratio and $T^* = \widetilde{\Theta}(\eta^{-1} \epsilon^{-1} nm \sigma_\xi^{-1} d^{-1})$ to be the maximum iterations for any given $\epsilon > 0$.

**Condition 4.1.** Suppose there exists a sufficiently large constant $C$ such that the following holds.

1. The signal-to-noise ratio and label flipping probability satisfy $n \cdot \mathrm{SNR}^2 = \Theta(1), \tau_+, \tau_- = \Theta(1)$.

2. The data dimension $d$ satisfies $d \geq C \max \left\{ n^2 \log(nm/\delta) \log(T^*)^2, n\|\boldsymbol{\mu}\|_2 \sigma_\xi^{-1} \sqrt{\log(n/\delta)} \right\}$.

3. The size of training sample $n$ and model width $m$ satisfy $m \geq C \log(n/\delta), n \geq C \log(m/\delta)$.

4. The signal strength $\|\boldsymbol{\mu}\|_2$ satisfies $\|\boldsymbol{\mu}\|_2^2 \geq C \sigma_\xi^2 \log(n/\delta)$.

5. The standard deviation $\sigma_0$ of the Gaussian distribution for weights initialization satisfies $\sigma_0 \leq C^{-1} \min \left\{ \sqrt{n} \sigma_\xi^{-1} d^{-1}, \|\boldsymbol{\mu}\|_2^{-1} \log(m/\delta)^{-1/2} \right\}$.

6. The learning rate $\eta$ satisfies $\eta \leq C^{-1} \min \left\{ \sigma_\xi^{-2} d^{-3/2} n^2 m \sqrt{\log(n/\delta)}, \sigma_\xi^{-2} d^{-1} n \right\}$.

**Remarks on Condition 4.1.** Condition 4.1 significantly differs from Kou et al. (2023, Condition 4.1.1) regarding the condition on SNR and label flipping probability $\tau_+, \tau_-$, despite their similarities. Kou et al. (2023) required $d \geq Cn\sigma_\xi^{-2}\|\boldsymbol{\mu}\|_2^2 \log(T^*)$, which translates to $n \cdot \mathrm{SNR}^2 \leq 1/(C \log(T^*)) = o(1)$; however, we require a constant order, i.e., $n \cdot \mathrm{SNR}^2 = \Theta(1)$. Furthermore, Kou et al. (2023) required $\tau_+, \tau_- \leq 1/C$ for sufficiently large constant $C$.[6] However, we require $\tau_+, \tau_-$ to be lower-bounded to emphasize the effects of label noise on the training dynamics. The other requirements regarding the network width $m$, sample size $n$, signal norm $\|\boldsymbol{\mu}\|_2$, initialization standard deviation $\sigma_0$ and learning rate $\eta$ are consistent with those in Kou et al. (2023), which are crucial to ensure the training loss converges under gradient descent.

---

[6]They consider $p = \tau_+ = \tau_-$.

## 4.1 Feature Learning Process With Label Noise

In this subsection, we analyze the feature learning process of neural networks in the presence of label noise. We identify two stages where the learning dynamics exhibits distinct behaviours. This two-stage behaviour only occurs when $n \cdot \mathrm{SNR}^2 = \Theta(1)$ and $\tau_+, \tau_- = \Theta(1)$.

**Stage I. Model fits clean data.** Theorem 4.1 characterizes the learning outcome at the end of Stage I.

**Theorem 4.1.** *Under Condition 4.1, there exists $T_1 = \Theta\left(\eta^{-1}nm\sigma_\xi^{-2}d^{-1}\right)$ such that $\overline{\rho}_{\tilde{y}_i,r,i}^{(T_1)} = \Theta(1)$ for all $i \in [n]$, $r \in [m]$ with $\langle \mathbf{w}_{\tilde{y}_i,r}^{(0)}, \boldsymbol{\xi}_i \rangle \geq 0$ and $\gamma_{j,r}^{(T_1)} = \Theta(1)$ for all $j = \pm 1, r \in [m]$, and*

1. *$\gamma_{j,r}^{(T_1)} > \overline{\rho}_{\tilde{y}_i,r,i}^{(T_1)}$ for all $j = \pm 1, r \in [m], i \in [n]$.*

2. *All clean samples $i \in \mathcal{S}_t$ satisfy that $\tilde{y}_i f(\mathbf{W}^{(T_1)}, \mathbf{x}_i) \geq 0$.*

3. *All noisy samples $i \in \mathcal{S}_f$ satisfy that $\tilde{y}_i f(\mathbf{W}^{(T_1)}, \mathbf{x}_i) \leq 0$.*

Theorem 4.1 implies that at the end of Stage I, the signal coefficients are larger than the noise coefficients, i.e., $\gamma_{j,r}^{(T_1)} > \overline{\rho}_{\tilde{y}_i,r,i}^{(T_1)}$, suggesting signal learning dominates the feature learning process in Stage I. Theorem 4.1 also demonstrates that for all clean samples, the model makes correct predictions, i.e., $\forall i \in \mathcal{S}_t, \tilde{y}_i f(\mathbf{W}^{(T_1)}, \mathbf{x}_i) \geq 0$, while for for all noisy samples, the model makes incorrect predictions, i.e., $\forall i \in \mathcal{S}_f, \tilde{y}_i f(\mathbf{W}^{(T_1)}, \mathbf{x}_i) \leq 0$. This indicates that the model fits all the clean samples while ignoring the noisy samples in Stage I.

**Stage II. Loss converges and model fits noisy data.** Theorem 4.2 formalize the learning behaviour in Stage II as the training loss converges.

**Theorem 4.2.** *Under Condition 4.1, for arbitrary $\epsilon > 0$, there exists $t^* \in [T_1, T^*]$, such that training loss converges, i.e., $L_S(\mathbf{W}^{(t^*)}) \leq \epsilon$ and*

1. *All clean samples, i.e., $i \in \mathcal{S}_t$, it holds that $\tilde{y}_i f(\mathbf{W}^{(t^*)}, \mathbf{x}_i) \geq 0$.*

2. *There exists a constant $0 < \tau' \leq \frac{\tau_+ + \tau_-}{2}$ such that there are $\tau' n$ noisy samples, i.e., $i \in \mathcal{S}_f$ that satisfy $\frac{1}{m}\sum_{r=1}^m \overline{\rho}_{\tilde{y}_i,r,i}^{(t^*)} > \frac{1}{m}\sum_{r=1}^m \gamma_{-\tilde{y}_i,r}^{(t^*)}$. and $\tilde{y}_i f(\mathbf{W}^{(t^*)}, \mathbf{x}_i) \geq 0$.*

3. *Test error $L_D^{0-1}(\mathbf{W}^{(t^*)}) \geq 0.5 \min\{\tau_+, \tau_-\}$.*

Theorem 4.2 states that in Stage II, as the training loss converges, for all clean samples, the model continues to make correct predictions, consistent with Stage I; however, for some noisy samples, the averaged noise coefficient across all neurons surpasses the averaged signal coefficient, i.e., $\frac{1}{m}\sum_{r=1}^m \overline{\rho}_{\tilde{y}_i,r,i}^{(t^*)} > \frac{1}{m}\sum_{r=1}^m \gamma_{-\tilde{y}_i,r}^{(t^*)}$. Consequently, for these noisy samples, the model's predictions align with the noisy observed labels $\tilde{y}$. Theorem 4.2 further shows that if evaluating the model on the test distribution introduced in Section 3, it results in a constant, non-vanishing test error, which is lower bounded by the label flipping probability in the training set, i.e., $L_D^{0-1}(\mathbf{W}^{(t^*)}) \geq 0.5 \min\{\tau_+, \tau_-\}$. We lastly remark that whether $\tau_+, \tau_-$ is identical does not alter the feature learning dynamics.

**Early stopping and sample selection.** Our results naturally explains the effectiveness of two common strategies frequently used in practice to address label noise, i.e., *early stopping* (Liu et al., 2020; Bai et al., 2021) and *sample selection* (Han et al., 2018; 2020). Intuitively, early stopping aims to stop training before the loss converges, preventing over-fitting to noisy samples. On the other hand, sample selection leverages the small-loss criterion Han et al. (2018) to distinguish clean samples from noisy ones, assuming that samples with smaller losses are more likely to be clean. Corollary 4.1 formally supports the effectiveness of these two strategies.

**Corollary 4.1** (Early stopping and sample selection). *Under the same conditions as in Theorem 4.1, the test error satisfies $L_D^{0-1}(\mathbf{W}^{(T_1)}) \leq \exp(-dn^{-1}/C')$ for some constant $C' > 0$. In addition, for all $i \in \mathcal{S}_t$, $\ell_i^{(T_1)} \leq \log(2)$ and for all $i \in \mathcal{S}_f$, $\ell_i^{(T_1)} \geq \log(2)$.*

Corollary 4.1 implies that if training is stopped during Stage I, before the loss converges, the test error can be upper bounded arbitrarily small under the condition that $d = \widetilde{\Omega}(n^2)$. Corollary 4.1 also states that noisy samples tend to have higher loss values compared to clean samples, and there exists a hard threshold $\log(2)$, which allows for a perfect separation of clean samples from noisy ones.

### 4.2 FEATURE LEARNING PROCESS WITHOUT LABEL NOISE

In this subsection, we analyze the feature learning process of neural networks without label noise.

**Loss converges and model fits all data.** The analysis of feature learning when training without label noise follows a similar two-stage framework as the analysis conducted with label noise. Theorem 4.3 formally characterizes the learning outcome without label noise.

**Theorem 4.3.** *Under Condition 4.1 with $\tau_+, \tau_- = 0$, there exists $T_1 = \Theta(\eta^{-1}\epsilon^{-1}nm\sigma_\xi^{-2}d^{-1})$ such that (1) $\overline{\rho}_{y_i,r,i}^{(T_1)} = \Theta(1)$ for all $i \in [n]$ and $r \in [m]$ such that $\langle \mathbf{w}_{\tilde{y}_i,r}^{(0)}, \boldsymbol{\xi}_i \rangle \geq 0$, (2) $\gamma_{j,r}^{(T_1)} = \Theta(1)$ for all $j = \pm 1, r \in [m]$ and (3) $y_i f(\mathbf{W}^{(T_1)}, \mathbf{x}_i) \geq 0$ for all $i \in [n]$. In addition, there exists a time $t^* \in [T_1, T^*]$ such that training loss converges, i.e., $L_S(\mathbf{W}^{(t^*)}) \leq \epsilon$ and*

1. *$y_i f(\mathbf{W}^{(t^*)}, \mathbf{x}_i) \geq 0$ for all $i \in [n]$,*

2. *Test error is bounded as $L_D^{0-1}(\mathbf{W}^{(t^*)}) \leq \exp\left(\frac{d}{n} - \frac{n\|\boldsymbol{\mu}\|_2^4}{C_D\sigma_\xi^4 d}\right)$ for some constant $C_D > 0$.*

Theorem 4.3 demonstrates that without label noise, all samples can be predicted correctly at the end of both Stage I and Stage II, suggesting that the model fits all the samples throughout the training process. Theorem 4.3 also implies that under the test distribution, the test error can be upper bounded by $\exp(\frac{d}{n} - \frac{n\|\boldsymbol{\mu}\|_2^4}{C_D\sigma_\xi^4 d})$. Thus, if $n \cdot \mathrm{SNR}^2 \geq 2C_D = \Theta(1)$, it follows that $L_D^{0-1}(\mathbf{W}^{(t^*)}) \leq \exp(-\frac{n\|\boldsymbol{\mu}\|_2^4}{2C_D\sigma_\xi^4 d})$, which is small given the requirement on $d$ (see Condition 4.1).

## 5 OVERVIEW OF PROOF STRATEGIES

In this section, we provide an overview of our proof techniques for our theoretical results in Section 4.

### 5.1 FEATURE LEARNING PROCESS WITH LABEL NOISE

The analysis of feature learning in the presence of label noise critically relies on Lemma 5.1, which shows the difference in terms of model predictions between clean and noisy samples.

**Lemma 5.1.** *Under Condition 4.1, there exists a sufficiently large constant $C_1$ such that for all $t \in [0, T^*]$, the following are satisfied:*

- *$\frac{1}{m}\sum_{r=1}^{m}\left(\gamma_{\tilde{y}_i,r}^{(t)} + \overline{\rho}_{\tilde{y}_i,r,i}^{(t)}\right) - 1/C_1 \leq \tilde{y}_i f(\mathbf{W}^{(t)}, \mathbf{x}_i) \leq \frac{1}{m}\sum_{r=1}^{m}\left(\gamma_{\tilde{y}_i,r}^{(t)} + \overline{\rho}_{\tilde{y}_i,r,i}^{(t)}\right) + 1/C_1$ for all clean samples, i.e., $i \in \mathcal{S}_t$*

- *$\frac{1}{m}\sum_{r=1}^{m}\left(\overline{\rho}_{\tilde{y}_i,r,i}^{(t)} - \gamma_{-\tilde{y}_i,r}^{(t)}\right) - 1/C_1 \leq \tilde{y}_i f(\mathbf{W}^{(t)}, \mathbf{x}_i) \leq \frac{1}{m}\sum_{r=1}^{m}\left(\overline{\rho}_{\tilde{y}_i,r,i}^{(t)} - \gamma_{-\tilde{y}_i,r}^{(t)}\right) + 1/C_1$ for all noisy samples, i.e., $i \in \mathcal{S}_f$.*

Lemma 5.1 suggests the for clean samples $i \in \mathcal{S}_t$, the mode prediction $\tilde{y}_i f(\mathbf{W}^{(t)}, \mathbf{x}_i)$ is determined by $\frac{1}{m}\sum_{r=1}^{m}\left(\gamma_{\tilde{y}_i,r}^{(t)} + \overline{\rho}_{\tilde{y}_i,r,i}^{(t)}\right)$, while for noisy samples $i \in \mathcal{S}_f$, $\tilde{y}_i f(\mathbf{W}^{(t)}, \mathbf{x}_i)$ is characterized by $\frac{1}{m}\sum_{r=1}^{m}\left(\overline{\rho}_{\tilde{y}_i,r,i}^{(t)} - \gamma_{-\tilde{y}_i,r}^{(t)}\right)$.

Besides Lemma 5.1, we also need to bound the scale of coefficients throughout the training process.

**Proposition 5.1.** *Under Condition 4.1, for any $0 \leq t \leq T^*$, we can bound*

$$0 \leq \overline{\rho}_{j,r,i}^{(t)}, \gamma_{j,r}^{(t)} \leq \Theta(\log(T^*)),$$
$$0 \geq \underline{\rho}_{j,r,i}^{(t)} \geq -\widetilde{O}(\max\{\sigma_0\|\boldsymbol{\mu}\|_2, \sigma_0\sigma_\xi\sqrt{d}, nd^{-1/2}\}).$$

Proposition 5.1 states that $|\underline{\rho}_{j,r,i}^{(t)}|$ is lower bounded by a small term based on Condition 4.1. In addition, both $\overline{\rho}_{j,r,i}^{(t)}, \gamma_{j,r}^{(t)}$ are positive and cannot grow faster than a logarithmic order of $T^*$. Notice that although Kou et al. (2023) demonstrates a similar bound, their analysis is not applicable to our case due to our condition that $n \cdot \mathrm{SNR}^2 = \Theta(1)$. Particularly, a key technique in their analysis is the automatic balance of updates, i.e., the loss derivatives $\ell_i'^{(t)}$ are balanced across all samples (Kou et al., 2023, Key Technique 2). In our case, however, due to the constant order of $n \cdot \mathrm{SNR}^2$, signal

coefficients are on the same scale as noise coefficients. Consequently, the loss derivatives $\ell_i'^{(t)}$ are no longer solely determined by $\frac{1}{m}\sum_{r=1}^m \overline{\rho}_{\tilde{y}_i,r,i}^{(t)}$ as was the case in Kou et al. (2023).

To prove Proposition 5.1 as well as the main theorems in Table 1, we separately consider two stages.

In *Stage I*, before the maximum of the coefficients reaches a constant order, all loss derivatives can be lower bounded by a constant, i.e., $|\ell_i'^{(t)}| \geq C_\ell$ for all $i \in [n]$. This ensures the balance of loss derivatives across all samples as $|\ell_i'^{(t)}| \leq 1$. Such a condition allows both both $\overline{\rho}_{j,r,i}^{(t)}, \gamma_{j,r}^{(t)}$ to increase to a constant order, enabling the establishment of the bound in Proposition 5.1. Furthermore, by applying Lemma 5.1, we can assert that $\tilde{y}_i f(\mathbf{W}^{(t)}, \mathbf{x}_i) \geq 0$ for all $i \in \mathcal{S}_t$. On the other hand, as long as $n \cdot \mathrm{SNR}^2 \geq c'$, for some constant $C' > 0$, we can demonstrate that signal learning slightly surpasses noise memorization, concluding that $\tilde{y}_i f(\mathbf{W}^{(t)}, \mathbf{x}_i) \leq 0$ for noisy samples $i \in \mathcal{S}_f$.

In *Stage II*, after the coefficients reach a constant order, the loss derivatives can no longer be lower-bounded by a constant. To establish that Proposition 5.1 still holds in this stage, we first rewrite the signal learning dynamics in Lemma 3.1 as follows:

$$\gamma_{j,r}^{(t+1)} = \gamma_{j,r}^{(t)} + \frac{\eta}{nm}\Big(\sum_{i\in\mathcal{S}_t} |\ell_i'^{(t)}| \mathbb{1}(\langle \mathbf{w}_{j,r}^{(t)}, y_i\boldsymbol{\mu}\rangle \geq 0) - \sum_{i\in\mathcal{S}_f} |\ell_i'^{(t)}| \mathbb{1}(\langle \mathbf{w}_{j,r}^{(t)}, y_i\boldsymbol{\mu}\rangle \geq 0)\Big)\|\boldsymbol{\mu}\|_2^2.$$

Recall that $|\ell_i'^{(t)}| = \frac{1}{1+\exp(\tilde{y}_i f(\mathbf{W}^{(t)}, \mathbf{x}_i))}$. Based on Lemma 5.1, $|\ell_i'^{(t)}| \mathbb{1}(i \in \mathcal{S}_f)$ can be larger than $|\ell_i'^{(t)}| \mathbb{1}(i \in \mathcal{S}_t)$, which causes $\gamma_{j,r}^{(t)}$ to decrease. However, we show by contradiction that $\gamma_{j,r}^{(t)} \leq 0$ cannot occur. When $\gamma_{j,r}^{(t)}$ decreases, the gap between $|\ell_i'^{(t)}| \mathbb{1}(i \in \mathcal{S}_t)$ and $|\ell_i'^{(t)}| \mathbb{1}(i \in \mathcal{S}_f)$ diminishes, allowing $\gamma_{j,r}^{(t)}$ to eventually increase in subsequent iterations. Therefore, the upper bound for both $\gamma_{j,r}^{(t)}, \overline{\rho}_{j,r,i}^{(t)}$ can be derived by showing that $|\ell_i'^{(t)}|$ converges at a rate of $O(1/t)$ when either $\gamma_{j,r}^{(t)}$ or $\overline{\rho}_{j,r,i}^{(t)}$ grows to a logarithmic order.

Additionally, we demonstrate that training loss converges at some iteration $t^*$ using a similar analysis as in Cao et al. (2022); Kou et al. (2023). Upon convergence, because of the monotonicity of $\overline{\rho}_{\tilde{y}_i,r,i}^{(t)}$ and positivity of $\gamma_{j,r}^{(t)}$, we can show $\tilde{y}_i f(\mathbf{W}^{t^*}, \mathbf{x}_i) \geq 0$ for all $i \in \mathcal{S}_t$. On the other hand, we establish by contradiction that there must exist a constant fraction of noisy samples satisfying $\tilde{y}_i f(\mathbf{W}^{t^*}, \mathbf{x}_i) \geq 0$. Because if not, we would have $\frac{1}{m}\sum_{r=1}^m (\overline{\rho}_{\tilde{y}_i,r,i}^{(t^*)} - \gamma_{-\tilde{y}_i,r}^{(t^*)}) \leq C_\epsilon$ for some constant $C_\epsilon > 0$ over a constant fraction of samples. This suggests the training loss $L_S(\mathbf{W}^{(t^*)})$ can be lower bounded by a strictly positive constant $c_l > 0$, which leads to a contradiction.

Finally, to establish the test error, we show that, based on the test distribution $\mathcal{D}_{\mathrm{test}}$ introduced in Section 3, there exits some sufficiently large constant $C'$ that

$$\mathbb{P}(yf(\mathbf{W}^{(t^*)}, \mathbf{x}) < 0)$$
$$\geq \frac{1}{n}\sum_{i=1}^n \mathbb{P}\Big(\frac{1}{m}\sum_{r=1}^m \sigma(\langle \mathbf{w}_{-y,r}^{(t^*)}, \boldsymbol{\xi}_i + \boldsymbol{\zeta}\rangle) - \frac{1}{m}\sum_{r=1}^m \sigma(\langle \mathbf{w}_{y,r}^{(t^*)}, \boldsymbol{\xi}_i + \boldsymbol{\zeta}\rangle) \geq \frac{1}{m}\sum_{r=1}^m \gamma_{y,r}^{(t^*)} + 1/C'\Big)$$

Next, we show that for any $y = \pm 1$, there exists some sufficiently large constant $C_2$ that for any $i \in \mathcal{S}_f \cap \mathcal{S}_y$, $\langle \mathbf{w}_{-y,r}^{(t^*)}, \boldsymbol{\xi}_i + \boldsymbol{\zeta}\rangle \geq \overline{\rho}_{-y,r,i}^{(t^*)} - 1/C_2$, $\langle \mathbf{w}_{y,r}^{(t^*)}, \boldsymbol{\xi}_i + \boldsymbol{\zeta}\rangle \leq 1/C_2$. Then, based on the scale that $\frac{1}{m}\sum_{r=1}^m (\overline{\rho}_{\tilde{y}_i,r,i}^{(t^*)} - \gamma_{-\tilde{y}_i,r}^{(t^*)}) \geq C_\epsilon$, we can show $\mathbb{P}(yf(\mathbf{W}^{(t^*)}, \mathbf{x}) < 0) \geq 0.5\tau_+$ if $y = 1$ and $\mathbb{P}(yf(\mathbf{W}^{(t^*)}, \mathbf{x}) < 0) \geq 0.5\tau_-$ if $y = -1$ given that $|\mathcal{S}_f \cap \mathcal{S}_1| = \tau_y n/2$.

## 5.2 FEATURE LEARNING PROCESS WITHOUT LABEL NOISE

The analysis of feature learning without label noise follows a similar two-stage framework as the analysis with label noise. Our analysis without label noise relies on in Lemma 5.2, where we show that under Condition 4.1, the loss derivatives are balanced across all samples, and both the noise coefficients and signal coefficients are of the same order.

**Lemma 5.2.** *Under Condition 4.1, for any $0 \leq t \leq T^*$, the following holds*

1. $\frac{1}{m}\sum_{r=1}^m (\overline{\rho}_{y_i,r,i}^{(s)} + \gamma_{y_i,r}^{(s)} - \overline{\rho}_{y_k,r,k}^{(s)} - \gamma_{y_k,r}^{(s)}) \leq \kappa$ *for all $i,k \in [n]$.*

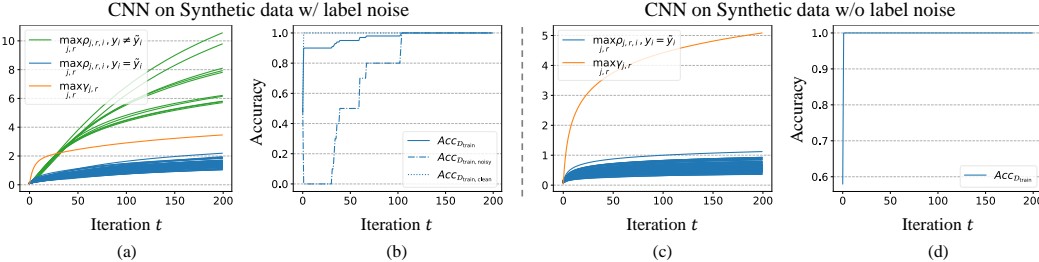

(a)       (b)       (c)       (d)

Figure 1: **Experimental validation under the synthetic setup, with label noise (a, b) and without label noise (c, d).** **(a, c)** The change in $\max_{j,r} \gamma_{j,r}$ (signal learning) and $\max_{j,r} \rho_{j,r,i}$ (noise memorization) on noisy (i.e., when $y_i \neq \tilde{y}_i$) and clean samples (i.e., when $y_i = \tilde{y}_i$) w.r.t the training iteration $t$. **(b, d)** The change in overall training accuracy $Acc_{\mathcal{D}_{\text{train}}}$, as well as the accuracy on clean $Acc_{\mathcal{D}_{\text{train, clean}}}$ and noisy samples $Acc_{\mathcal{D}_{\text{train, noisy}}}$, w.r.t the training iteration $t$ for models under different settings. Note that there are no noisy samples when training without label noise; thus we only plot noise memorization on clean samples and the overall training accuracy.

2. $\ell_i'^{(s)}/\ell_k'^{(s)} \leq \tilde{C}_\ell$ for all $i, k \in [n]$.

Lemma 5.2 allows us to establish Proposition 5.1 for all training iterations. Initially, we can demonstrate that both signal and noise coefficients reach a constant order during Stage I, as in Section 5.1. In Stage II, we can show that the scales of the signal and noise coefficients remain on the same order, i.e., $\gamma_{j,r}^{(t)} / \sum_{i=1}^{n} \bar{\rho}_{j,r,i}^{(t)} = \Theta(\text{SNR}^2)$. Finally, by employing a similar test error analysis as in Kou et al. (2023), we can upper bound the test error in terms of $n \cdot \text{SNR}^2$.

## 6 EXPERIMENTS

In this section, we provide empirical evidence on both synthetic and real-world datasets to support our theoretical analysis in previous sections.

**Validation under the synthetic setup.** First, we conduct experiments under the synthetic setup to verify our theories. Here, we generate the data exactly based on the distribution introduced in Section 3. Specifically, without loss of generality, the fixed signal vector is set to be $\boldsymbol{\mu} = [y\mu, 0, \cdots, 0] \in \mathbb{R}^d$, where $\mu = 20$ and $d = 2000$, and the random noise vector $\boldsymbol{\xi}$ is sampled from $\mathcal{N}(0, \mathbf{I}_d)$. This setting corresponds to $n \cdot \text{SNR}^2 = 20$. We generate $n = 100$ training samples with balanced class labels and flip each sample's label with a probability of $0.1$. Then we train a two-layer CNN (as defined in Section 3) on this synthetic data using gradient descent, with a total of $T = 200$ iterations and a learning rate of $\eta = 0.1$. For comparison, we also train a baseline model under nearly identical settings, but on the dataset without label flipping.

Experiments under this synthetic setup successfully validate our two-stage picture in the feature learning process under the presence of label noise. Specifically, we demonstrate the signal learning process in the two-layer CNN by showing how $\max_{j,r} \gamma_{j,r}$ changes during training. We also present the noise memorization process by illustrating the evolution of $\max_{j,r} \rho_{j,r,i}$. In Figure 1 (a), a clear two-stage pattern emerges in the learning process for the model trained with label noise: i) in the early stage (from initialization to around 25 iterations), the value of $\max_{j,r} \gamma_{j,r}$ is significantly larger than that of $\max_{j,r} \rho_{j,r,i}$, indicating that the signal learning initially dominates; ii) in the later stage (after 25 iterations), the value of $\max_{j,r} \rho_{j,r,i}$ on noisy samples (i.e., when $y_i \neq \tilde{y}_i$) increasingly surpasses that of $\max_{j,r} \gamma_{j,r}$, implying that the noise memorization process, particularly for the noise in label-flipped samples, gradually takes over. In contrast, as shown in Figure 1 (c), when training without label noise, $\max_{j,r} \gamma_{j,r}$ consistently exceeds $\max_{j,r} \rho_{j,r,i}$ throughout the training. Additionally, we provide the classification accuracy for the training samples. In Figure 1 (b), for training with label noise, the accuracy on noisy samples initially drops to $0$ during the early stage and then gradually increases. Conversely, in Figure 1 (c), when training without label noise, the training accuracy remains consistently high once it reaches its maximum value. These results together support our two-stage picture, thereby verifying our theoretical analysis.

**Validation in the real-world scenario.** Taking a step further, we also validate our theoretical analysis in the real-world scenario. We perform experiments on the commonly used image classification dataset CIFAR-10 (Krizhevsky et al., 2009), using standard network architecture VGG-11 (Simonyan

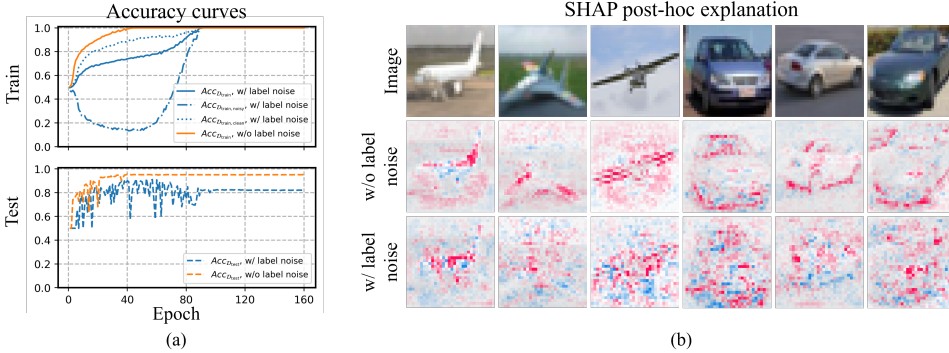

(a)               (b)

Figure 2: **Experimental validation in real-world scenarios.** Two VGG-11 nets are trained on the first two categories of CIFAR-10 under nearly identical settings, differing only in that one is trained with label noise and the other without. **(a)** The accuracy curves for the two models. Here, $Acc_{\mathcal{D}_{\text{train}}}$ and $Acc_{\mathcal{D}_{\text{test}}}$ represent the accuracy on the entire training and test sets, respectively, while $Acc_{\mathcal{D}_{\text{train, clean}}}$ and $Acc_{\mathcal{D}_{\text{train, noisy}}}$ specifically denote the accuracy on clean and noisy samples from the training set. Since there are no noisy samples when training without label noise, we only plot the overall training and test accuracy curves. **(b)** The interpretations of the model predictions using SHAP (Lundberg & Lee, 2017). The top row displays the input images, the middle row presents the interpretations for models trained without label noise, and the bottom row shows the interpretations for models trained with label noise. Red regions indicate positive contributions to model predictions, while blue regions denote negative contributions, with darker regions signifying greater contributions.

& Zisserman, 2015). Specifically, we train the VGG-11 network with stochastic gradient descent on samples from the first two categories of CIFAR-10, where each sample's label is flipped with a probability of 0.2. Similar to the synthetic experiment, for comparison, we also train another VGG-11 net under the same settings but without label flipping.

Experiments in the real-world scenario further *reinforce* the applicability of our theory. First, we provide the accuracy results on both training and test sets, denoted as $Acc_{\mathcal{D}_{\text{train}}}$ and $Acc_{\mathcal{D}_{\text{test}}}$, respectively. In Figure 2(a), when training with label noise, the accuracy on noisy samples follows a similar pattern to the synthetic experiments — an initial drop followed by a gradual increase to 1 — while the test set accuracy remains consistently lower than when training without label noise. However, accuracy results alone are insufficient to demonstrate that the model memorizes the noise from the label-flipped samples by the end of training. Furthermore, we visualize the interpretation of each model prediction using SHAP (Lundberg & Lee, 2017), which provides insights into the contribution of each input variable to the model prediction. In Figure 2(b), the interpretations for models trained with label noise, the interpretations appear messy with no clear pattern observed; yet, for models trained without label noise, the interpretations are more consistent, focusing on generalizable features such as the "wing" for the "airplane" class and the overall "outline" for the "automobile" class. These findings align with our theoretical results, confirming the model's two-stage behavior under noisy conditions.

## 7 CONCLUSION AND LIMITATIONS

In conclusion, our work rigorously characterized the role of label noise in the feature learning process of neural networks, identifying two stages in the learning dynamics. In *Stage I*, the model perfectly fits all the clean samples while ignoring the noisy ones, effectively learning the signal from the data and achieving good generalization. In *Stage II*, as the loss converges, the model inevitably over-fits to the noisy samples, learning noise from data and resulting in poor generalization. This two-stage behaviour firmly explains the effectiveness of early stopping and sample selection. Since the model performs well at the end of Stage I, a natural future direction is to develop methods for accurately identifying the point at which Stage I concludes.

**Limitations.** We note that our current empirical validations mainly focus on image classification tasks, though aligning with most literature on label noise. We also note that our theoretical analysis still relies on several assumptions, such as the simplified data and model setups.

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

# A    COMPARISON OF CONDITIONS TO (KOU ET AL., 2023)

This section compares the Condition 4.1 to conditions required in (Kou et al., 2023). We present the comparisons in Table 2. As observed in Table 2, the differences in the conditions are regarding label noise ratio $\tau$, SNR scale $n \cdot \text{SNR}^2$ and dimensionality.

| Label noise | (Kou et al., 2023) | $\tau \leq 1/C$ |
|---|---|---|
| | Our work | $\tau = \Theta(1)$ |
| SNR | (Kou et al., 2023) | $n \cdot \text{SNR}^2 \leq 1/(C \log(T^*))$ |
| | Our work | $n \cdot \text{SNR}^2 = \Theta(1)$ |
| Dimension | (Kou et al., 2023) | $d \geq C \max\{n\sigma_\xi^{-2}\|\boldsymbol{\mu}\|_2^2 \log(T^*), n^2 \log(nm/\delta)(\log(T^*))^2\}$ |
| | Our work | $d \geq C \max\{n\sigma_\xi^{-1}\|\boldsymbol{\mu}\|_2 \sqrt{\log(n/\delta)}, n^2 \log(nm/\delta)(\log(T^*))^2\}$ |
| Sample size | (Kou et al., 2023) | $n \geq C \log(m/\delta)$ |
| | Our work | $n \geq C \log(m/\delta)$ |
| Network width | (Kou et al., 2023) | $m \geq C \log(n/\delta)$ |
| | Our work | $m \geq C \log(n/\delta)$ |
| Signal norm | (Kou et al., 2023) | $\|\boldsymbol{\mu}\|_2^2 \geq C\sigma_\xi^2 \log(n/\delta)$ |
| | Our work | $\|\boldsymbol{\mu}\|_2^2 \geq C\sigma_\xi^2 \log(n/\delta)$ |
| Learning rate | (Kou et al., 2023) | $\eta \leq C^{-1} \min\left\{\sigma_\xi^{-2}d^{-3/2}n^2m\sqrt{\log(n/\delta)}, \sigma_\xi^{-2}d^{-1}n\right\}$ |
| | Our work | $\eta \leq C^{-1} \min\left\{\sigma_\xi^{-2}d^{-3/2}n^2m\sqrt{\log(n/\delta)}, \sigma_\xi^{-2}d^{-1}n\right\}$ |
| Initialization | (Kou et al., 2023) | $\sigma_0 \leq C^{-1} \min\left\{\sqrt{n}\sigma_\xi^{-1}d^{-1}, \|\boldsymbol{\mu}\|_2^{-1}\log(m/\delta)^{-1/2}\right\}$ |
| | Our work | $\sigma_0 \leq C^{-1} \min\left\{\sqrt{n}\sigma_\xi^{-1}d^{-1}, \|\boldsymbol{\mu}\|_2^{-1}\log(m/\delta)^{-1/2}\right\}$ |

Table 2: Comparisons of required conditions. $T^* = \eta^{-1}\text{poly}(\epsilon^{-1}, d, n, m)$ is the maximum iterations considered. Comparing to (Kou et al., 2023), the only differences in the conditions are regarding the label noise scale $\tau$, SNR and dimension $d$.

# B    COMPARISON OF TECHNICAL QUANTITIES TO (KOU ET AL., 2023)

Among the various differences in conditions compared to (Kou et al., 2023), the most critical distinction lies in the scale of the SNR. Because we aim to characterize the two-stage behaviors induced by label noise, we require the SNR to satisfy $n \cdot \text{SNR}^2 = \Theta(1)$. This enables the signal learning to dominate the noise learning in the first stage while noise learning dominates signal learning in the second stage. Such a distinct two-stage dynamics cannot be captured by (Kou et al., 2023) due to $n \cdot \text{SNR}^2 = o(1)$.

More specifically, in the following, we explicitly compares the key differences in the analysis techniques compared to (Kou et al., 2023):

- **Non-Time-invariant coefficients**: One of the key techniques (Key Technique 1 in (Kou et al., 2023)) is the derivation of time-invariant order of the coefficient ratio: $\gamma_{j,r}^{(t)}/\sum_{i=1}^n \overline{\rho}_{j,r,i}^{(t)} = \Theta(\text{SNR}^2)$, which is critical for their generalization analysis. However, in our case, due to the setting of constant order $n \cdot \text{SNR}^2$, the noisy samples exhibit different behaviors as the clean samples (which is the main goal we wish to show), such time-invariance may not hold for all iterations.

- **Non-balancing of the updates**: Another key technique employed in (Kou et al., 2023) is the automatic balancing of coefficient updates, which requires to show $\ell_i^{\prime(t)}/\ell_k^{\prime(t)} \leq C$ for all $i, k \in [n]$. That is, the loss derivatives across all samples are approximately balanced, which is critical for their convergence analysis. Because in our case $n \cdot \text{SNR}^2$, the loss derivatives of noisy samples may be significantly larger than that of clean samples, we cannot guarantee the balance of updates across all samples.

Without the above two results in our case, the convergence and generalization analysis becomes challenging. To address the challenges, we require to develop novel techniques via refined analysis on clean and noisy samples, which cannot be addressed in the prior works.

To better comprehend the differences to the analysis of (Kou et al., 2023), we present the following tables that compares the different quantities at each training stage. These differences require non-trivial analysis.

| | | First Stage | Second Stage |
|---|---|---|---|
| Monotonicity of signal | (Kou et al., 2023) | Monotonic increase | |
| | Our work | Monotonic increase | No monotonicity |
| Signal-noise magnitude | (Kou et al., 2023) | Noise dominates | |
| | Our work | Signal dominates | Noise dominates |
| Determining factors of $\tilde{y}_i f(\mathbf{W}^{(t)}, \mathbf{x}_i)$ | (Kou et al., 2023) | $\frac{1}{m}\sum_{r=1}^m \overline{\rho}_{\tilde{y}_i,r,i}^{(t)} \pm o(1)$ | |
| | Our work | $\begin{cases} \frac{1}{m}\sum_{r=1}^m (\gamma_{\tilde{y}_i,r}^{(t)} + \overline{\rho}_{\tilde{y}_i,r,i}^{(t)}) \pm o(1), & \text{for } i \in \mathcal{S}_t \\ \frac{1}{m}\sum_{r=1}^m (\overline{\rho}_{\tilde{y}_i,r,i}^{(t)} - \gamma_{-\tilde{y}_i,r}^{(t)}) \pm o(1), & \text{for } i \in \mathcal{S}_f \end{cases}$ | |
| Prediction | (Kou et al., 2023) | $\tilde{y}_i f(\mathbf{W}^{(t)}, \mathbf{x}_i) \geq 0, \forall i \in [n]$ | |
| | Our work | $\begin{cases} \tilde{y}_i f(\mathbf{W}^{(t)}, \mathbf{x}_i) \geq 0, & i \in \mathcal{S}_t \\ \tilde{y}_i f(\mathbf{W}^{(t)}, \mathbf{x}_i) \leq 0, & i \in \mathcal{S}_f \end{cases}$ | $\tilde{y}_i f(\mathbf{W}^{(t)}, \mathbf{x}_i) \geq 0, \forall i \in [n]$ |
| Test error $L_D^{0-1}(\mathbf{W}^{(T_1)})$ | (Kou et al., 2023) | $\begin{cases} o(1), & \text{if } n\|\boldsymbol{\mu}\|_2^4 > C - 1\sigma_\xi^4 d \\ \Omega(1), & \text{if } n\|\boldsymbol{\mu}\|_2^4 \leq C_3\sigma_\xi^4 d \end{cases}$ | |
| | Our work | $o(1)$ | $\Omega(1)$ |

Table 3: Comparisons of key quantities in the analysis at each stage.

## C  PRELIMINARY LEMMAS

This section introduces a few lemmas that are critical to bound the parameters at initialization.

**Lemma C.1** (Cao et al. (2022); Kou et al. (2023)). *Suppose $d = \Omega(\log(6n/\delta))$. Then with probability at least $1 - \delta$,*

$$\sigma_\xi^2 d/2 \leq \|\boldsymbol{\xi}_i\|_2^2 \leq 3\sigma_\xi^2 d/2,$$

$$|\langle \boldsymbol{\xi}_i, \boldsymbol{\xi}_{i'} \rangle| \leq 2\sigma_\xi^2 \sqrt{d\log(6n^2/\delta)},$$

$$|\langle \boldsymbol{\xi}_i, \boldsymbol{\mu} \rangle| \leq \|\boldsymbol{\mu}\|_2 \sigma_\xi \sqrt{2\log(6n/\delta)}.$$

**Lemma C.2** (Cao et al. (2022); Kou et al. (2023)). *Suppose that $d = \Omega(\log(nm/\delta))$, $m = \Omega(\log(1/\delta))$. Then with probability at least $1 - \delta$,*

$$\sigma_0^2 d/2 \leq \|\mathbf{w}_{j,r}^0\|_2^2 \leq 3\sigma_0^2 d/2$$

$$|\langle \mathbf{w}_{j,r}^{(0)}, \boldsymbol{\mu} \rangle| \leq \sqrt{2\log(12m/\delta)} \cdot \sigma_0 \|\boldsymbol{\mu}\|_2,$$

$$|\langle \mathbf{w}_{j,r}^{(0)}, \boldsymbol{\xi}_i \rangle| \leq 2\sqrt{\log(12mn/\delta)} \cdot \sigma_0 \sigma_\xi \sqrt{d}.$$

**Lemma C.3** (Kou et al. (2023)). *Let $\mathcal{S}_i^{(t)} := \{r \in [m] : \langle \mathbf{w}_{\tilde{y}_i,r}^{(t)}, \boldsymbol{\xi}_i \rangle > 0\}$ and $\mathcal{S}_{j,r}^{(t)} := \{i \in [n] : j = \tilde{y}_i, \langle \mathbf{w}_{j,r}^{(t)}, \boldsymbol{\xi}_i \rangle > 0\}$. Then for any $\delta > 0$, and $m \geq 50\log(4n/\delta)$, $n \geq 32\log(8m/\delta)$, we have with probability at least $1 - \delta$,*

$$|\mathcal{S}_i^{(0)}| \geq 0.4m, \quad \forall i \in [n]$$

$$|\mathcal{S}_{j,r}^{(0)}| \geq n/8, \quad \forall j = \pm 1, r \in [m].$$

## D  ANALYSIS WITH LABEL NOISE

Without loss of generality, for the subsequent analysis, we assume $|\mathcal{S}_1 \cap \mathcal{S}_t| = \frac{(1-\tau_+)n}{2}$, $|\mathcal{S}_1 \cap \mathcal{S}_f| = \frac{\tau_+ n}{2}$, $|\mathcal{S}_{-1} \cap \mathcal{S}_t| = \frac{(1-\tau_-)n}{2}$, $|\mathcal{S}_{-1} \cap \mathcal{S}_f| = \frac{\tau_- n}{2}$.

### D.1 COEFFICIENTS DECOMPOSITION ITERATION

*Proof of Lemma 3.1.* By iterating the gradient descent update, we can show

$$\mathbf{w}_{j,r}^{(t)} = \mathbf{w}_{j,r}^{(0)} - \frac{\eta}{nm} \sum_{s=0}^{t-1} \sum_{i=1}^{n} \ell_i'^{(s)} \sigma'(\langle \mathbf{w}_{j,r}^{(s)}, \boldsymbol{\xi}_i \rangle) j \tilde{y}_i \boldsymbol{\xi}_i - \frac{\eta}{nm} \sum_{s=0}^{t-1} \sum_{i=1}^{n} \ell_i'^{(s)} \sigma'(\langle \mathbf{w}_{j,r}^{(s)}, y_i \boldsymbol{\mu} \rangle) j y_i \tilde{y}_i \boldsymbol{\mu}$$

Because $\boldsymbol{\xi}_i, \boldsymbol{\mu}$ are linearly independent almost surely for all $i \in [n]$. Then from the definition:

$$\mathbf{w}_{j,r}^{(t)} = \mathbf{w}_{j,r}^{(0)} + j \gamma_{j,r}^{(t)} \|\boldsymbol{\mu}\|_2^{-2} \boldsymbol{\mu} + \sum_{i=1}^{n} \rho_{j,r,i}^{(t)} \|\boldsymbol{\xi}_i\|_2^{-2} \boldsymbol{\xi}_i$$

there exists a unique decomposition as

$$\gamma_{j,r}^{(t)} = -\frac{\eta}{nm} \sum_{s=0}^{t-1} \sum_{i=1}^{n} \ell_i'^{(s)} \sigma'(\langle \mathbf{w}_{j,r}^{(s)}, y_i \boldsymbol{\mu} \rangle) y_i \tilde{y}_i \|\boldsymbol{\mu}\|_2^2$$

$$\rho_{j,r,i}^{(t)} = -\frac{\eta}{mn} \sum_{s=0}^{t-1} \ell_i'^{(s)} \sigma'(\langle \mathbf{w}_{j,r}^{(s)}, \boldsymbol{\xi}_i \rangle) j \tilde{y}_i \|\boldsymbol{\xi}_i\|_2^2.$$

By definition of $\overline{\rho}_{j,r,i}^{(t)}, \underline{\rho}_{j,r,i}^{(t)}$, and the fact that $\ell_i' \leq 0$,

$$\overline{\rho}_{j,r,i}^{(t)} = -\frac{\eta}{nm} \sum_{s=0}^{t-1} \ell_i'^{(s)} \sigma'(\langle \mathbf{w}_{j,r}^{(s)}, \boldsymbol{\xi}_i \rangle) \|\boldsymbol{\xi}_i\|_2^2 \mathbb{1}(\tilde{y}_i = j)$$

$$\underline{\rho}_{j,r,i}^{(t)} = \frac{\eta}{nm} \sum_{s=0}^{t-1} \ell_i'^{(s)} \sigma'(\langle \mathbf{w}_{j,r}^{(s)}, \boldsymbol{\xi}_i \rangle) \|\boldsymbol{\xi}_i\|_2^2 \mathbb{1}(\tilde{y}_i = -j)$$

Then the iterative updates of the coefficients follow directly. $\square$

### D.2 SCALE OF COEFFICIENTS

Here we start to provide a global bound for the decomposition coefficients. We show for a sufficiently large number of iterations $T^* = \widetilde{\Theta}(\eta^{-1} \epsilon^{-1} nmd^{-1} \sigma_\xi^{-2})$, the scale of the coefficients can be upper bounded up to some logarithmic factors.

We consider the following definition:

$$\beta = 2 \max_{i,j,r} \{|\langle \mathbf{w}_{j,r}^{(0)}, \boldsymbol{\mu} \rangle|, |\langle \mathbf{w}_{j,r}^{(0)}, \boldsymbol{\xi}_i \rangle|\}, \qquad \text{SNR} = \frac{\|\boldsymbol{\mu}\|}{\sigma_\xi \sqrt{d}}, \qquad \alpha = C_t \log(T^*)$$

for some constant $C_t > 0$ to be determined later. Then by Lemma C.2, we can bound as $\beta \leq \sigma_0 \max\left\{\sqrt{2 \log(12m/\delta)} \|\boldsymbol{\mu}\|_2, 2\sqrt{\log(12mn/\delta)} \sigma_\xi \sqrt{d}\right\}$.

We next provide the main proposition that bounds the scale of coefficients.

**Proposition D.1** (Restatement of Proposition 5.1). *Under Condition 4.1, for any $0 \leq t \leq T^*$*

$$0 \leq \overline{\rho}_{j,r,i}^{(t)} \leq \alpha, \tag{4}$$

$$0 \geq \underline{\rho}_{j,r,i}^{(t)} \geq -\beta - 10\sqrt{\frac{\log(6n^2/\delta)}{d}} n\alpha \geq -\alpha, \tag{5}$$

$$0 \leq \gamma_{j,r}^{(t)} \leq C_\gamma \alpha \tag{6}$$

*for some constant $C_\gamma > 0$.*

We aim to prove Proposition D.1 using induction. This requires several intermediate lemmas through the induction process.

**Lemma D.1.** *Under Condition 4.1, suppose (4), (5), (6) hold at iteration $t$. Then for all $r \in [m]$, $j \in \{\pm 1\}, i \in [n]$,*

$$|\langle \mathbf{w}_{j,r}^{(t)} - \mathbf{w}_{j,r}^{(0)}, \boldsymbol{\mu} \rangle - j \cdot \gamma_{j,r}^{(t)}| \le \mathrm{SNR} \sqrt{\frac{8 \log(6n/\delta)}{d}} n\alpha,$$

$$|\langle \mathbf{w}_{j,r}^{(t)} - \mathbf{w}_{j,r}^{(0)}, \boldsymbol{\xi}_i \rangle - \overline{\rho}_{j,r,i}^{(t)}| \le 5 \sqrt{\frac{\log(6n^2/\delta)}{d}} n\alpha, \quad \tilde{y}_i = j$$

$$|\langle \mathbf{w}_{j,r}^{(t)} - \mathbf{w}_{j,r}^{(0)}, \boldsymbol{\xi}_i \rangle - \underline{\rho}_{j,r,i}^{(t)}| \le 5 \sqrt{\frac{\log(6n^2/\delta)}{d}} n\alpha, \quad \tilde{y}_i = -j$$

*Proof of Lemma D.1.* From signal-noise decomposition (3),

$$|\langle \mathbf{w}_{j,r}^{(t)} - \mathbf{w}_{j,r}^{(0)}, \boldsymbol{\mu} \rangle - j \cdot \gamma_{j,r}^{(t)}| = \Big| \sum_{i=1}^{n} \overline{\rho}_{j,r,i}^{(t)} \cdot \|\boldsymbol{\xi}_i\|_2^{-2} \cdot \langle \boldsymbol{\xi}_i, \boldsymbol{\mu} \rangle + \sum_{i=1}^{n} \underline{\rho}_{j,r,i}^{(t)} \cdot \|\boldsymbol{\xi}_i\|_2^{-2} \cdot \langle \boldsymbol{\xi}_i, \boldsymbol{\mu} \rangle \Big|$$

$$\le \sum_{i=1}^{n} (|\overline{\rho}_{j,r,i}^{(t)}| + |\underline{\rho}_{j,r,i}^{(t)}|) \|\boldsymbol{\xi}_i\|_2^{-2} \cdot |\langle \boldsymbol{\xi}_i, \boldsymbol{\mu} \rangle|$$

$$\le \mathrm{SNR} \sqrt{\frac{8 \log(6n/\delta)}{d}} \sum_{i=1}^{n} (|\overline{\rho}_{j,r,i}^{(t)}| + |\underline{\rho}_{j,r,i}^{(t)}|)$$

$$\le \mathrm{SNR} \sqrt{\frac{8 \log(6n/\delta)}{d}} n\alpha$$

where the second inequality is due to Lemma C.1 and the last inequality is by (4), (5). The second inequality follows similarly.

Then, for $\tilde{y}_i = j$, we have $\underline{\rho}_{j,r,i}^{(t)} = 0, \forall t \ge 0$ and hence

$$|\langle \mathbf{w}_{j,r}^{(t)} - \mathbf{w}_{j,r}^{(0)}, \boldsymbol{\xi}_i \rangle - \overline{\rho}_{j,r,i}^{(t)}|$$

$$= \Big| j \cdot \gamma_{j,r}^{(t)} \cdot \|\boldsymbol{\mu}\|_2^{-2} \langle \boldsymbol{\mu}, \boldsymbol{\xi}_i \rangle + \sum_{i' \ne i} \overline{\rho}_{j,r,i'}^{(t)} \cdot \|\boldsymbol{\xi}_{i'}\|_2^{-2} \cdot \langle \boldsymbol{\xi}_i, \boldsymbol{\xi}_{i'} \rangle + \sum_{i' \ne i} \underline{\rho}_{j,r,i'}^{(t)} \cdot \|\boldsymbol{\xi}_{i'}\|_2^{-2} \cdot \langle \boldsymbol{\xi}_i, \boldsymbol{\xi}_{i'} \rangle \Big|$$

$$\le \|\boldsymbol{\mu}\|_2^{-2} \cdot |\langle \boldsymbol{\mu}, \boldsymbol{\xi}_i \rangle| \cdot |\gamma_{j,r}^{(t)}| + \sum_{i' \ne i}^{n} (|\overline{\rho}_{j,r,i'}^{(t)}| + |\underline{\rho}_{j,r,i'}^{(t)}|) \|\boldsymbol{\xi}_{i'}\|_2^{-2} \cdot |\langle \boldsymbol{\xi}_{i'}, \boldsymbol{\xi}_i \rangle|$$

$$\le \mathrm{SNR} \sqrt{\frac{2 \log(6n/\delta)}{d}} C_\gamma n\alpha + 4 \sqrt{\frac{\log(6n^2/\delta)}{d}} n\alpha$$

$$\le (2C_\gamma \mathrm{SNR} + 4) \sqrt{\frac{\log(6n^2/\delta)}{d}} n\alpha$$

$$\le 5 \sqrt{\frac{\log(6n^2/\delta)}{d}} n\alpha$$

where we use Lemma C.1 and (6) in the second inequality. In the third inequality, we use $2 \log(6n/\delta) \le 4 \log(6n^2/\delta)$. In the fourth inequality, we note that the condition on SNR ensures that $\mathrm{SNR} = \Theta(1/\sqrt{n})$.

For $\tilde{y}_i \ne j$, the proof follow exactly the same strategy as for $\tilde{y}_i = j$ and hence is omitted. $\square$

**Lemma D.2.** *Under Condition 4.1 and suppose (4), (5), (6) hold at time $t$, then there exists a sufficiently large constant $C_1 > 0$ such that*

$$\frac{1}{m} \sum_{r=1}^{m} (\gamma_{\tilde{y}_i,r}^{(t)} + \overline{\rho}_{\tilde{y}_i,r,i}^{(t)}) - 1/C_1 \le \tilde{y}_i f(\mathbf{W}^{(t)}, \mathbf{x}_i) \le \frac{1}{m} \sum_{r=1}^{m} (\gamma_{\tilde{y}_i,r}^{(t)} + \overline{\rho}_{\tilde{y}_i,r,i}^{(t)}) + 1/C_1 \quad \text{when } i \in \mathcal{S}_t$$

$$\frac{1}{m} \sum_{r=1}^{m} (\overline{\rho}_{\tilde{y}_i,r,i}^{(t)} - \gamma_{-\tilde{y}_i,r}^{(t)}) - 1/C_1 \le \tilde{y}_i f(\mathbf{W}^{(t)}, \mathbf{x}_i) \le \frac{1}{m} \sum_{r=1}^{m} (\overline{\rho}_{\tilde{y}_i,r,i}^{(t)} - \gamma_{-\tilde{y}_i,r}^{(t)}) + 1/C_1 \quad \text{when } i \in \mathcal{S}_f$$

*Proof of Lemma D.2.* We first see

$$\tilde{y}_i f(\mathbf{W}^{(t)}, \mathbf{x}_i) = \frac{1}{m} \sum_{j,r} \tilde{y}_i \cdot j \cdot \left( \sigma(\langle \mathbf{w}_{j,r}^{(t)}, y_i \boldsymbol{\mu} \rangle) + \sigma(\langle \mathbf{w}_{j,r}^{(t)}, \boldsymbol{\xi}_i \rangle) \right)$$

$$= \frac{1}{m} \sum_{r=1}^{m} \left( \sigma(\langle \mathbf{w}_{\tilde{y}_i,r}^{(t)}, y_i \boldsymbol{\mu} \rangle) + \sigma(\langle \mathbf{w}_{\tilde{y}_i,r}^{(t)}, \boldsymbol{\xi}_i \rangle) \right) - \frac{1}{m} \sum_{r=1}^{m} \left( \sigma(\langle \mathbf{w}_{-\tilde{y}_i,r}^{(t)}, y_i \boldsymbol{\mu} \rangle) + \sigma(\langle \mathbf{w}_{-\tilde{y}_i,r}^{(t)}, \boldsymbol{\xi}_i \rangle) \right).$$

Recall from the gradient descent update and Lemma D.1,

$$\left| \langle \mathbf{w}_{j,r}^{(t)}, \boldsymbol{\mu} \rangle - \langle \mathbf{w}_{j,r}^{(0)}, \boldsymbol{\mu} \rangle - j \cdot \gamma_{j,r}^{(t)} \right| = \text{SNR} \sqrt{\frac{8 \log(6n/\delta)}{d}} n\alpha$$

Then it can be verified that when $\tilde{y}_i = y_i$,

$$\langle \mathbf{w}_{\tilde{y}_i,r}^{(t)}, y_i \boldsymbol{\mu} \rangle \le |\langle \mathbf{w}_{\tilde{y}_i,r}^{(0)}, \boldsymbol{\mu} \rangle| + \gamma_{\tilde{y}_i,r}^{(t)} + \text{SNR} \sqrt{\frac{8 \log(6n/\delta)}{d}} n\alpha$$

$$\langle \mathbf{w}_{\tilde{y}_i,r}^{(t)}, -y_i \boldsymbol{\mu} \rangle \le |\langle \mathbf{w}_{\tilde{y}_i,r}^{(0)}, \boldsymbol{\mu} \rangle| - \gamma_{\tilde{y}_i,r}^{(t)} + \text{SNR} \sqrt{\frac{8 \log(6n/\delta)}{d}} n\alpha$$

$$\le |\langle \mathbf{w}_{\tilde{y}_i,r}^{(0)}, \boldsymbol{\mu} \rangle| + \text{SNR} \sqrt{\frac{8 \log(6n/\delta)}{d}} n\alpha$$

$$\langle \mathbf{w}_{\tilde{y}_i,r}^{(t)}, \boldsymbol{\xi}_i \rangle \le |\langle \mathbf{w}_{\tilde{y}_i,r}^{(0)}, \boldsymbol{\xi}_i \rangle| + \overline{\rho}_{\tilde{y}_i,r,i}^{(t)} + 5 \sqrt{\frac{\log(6n^2/\delta)}{d}} n\alpha$$

$$\langle \mathbf{w}_{-\tilde{y}_i,r}^{(t)}, -y_i \boldsymbol{\mu} \rangle \ge \gamma_{-\tilde{y}_i,r}^{(t)} - |\mathbf{w}_{-\tilde{y}_i,r}^{(0)}, \boldsymbol{\mu}| - \text{SNR} \sqrt{\frac{8 \log(6n/\delta)}{d}} n\alpha$$

Using these inequalities, we can upper bound when $\tilde{y}_i = y_i$, i.e., $i \in \mathcal{S}_t$,

$$\tilde{y}_i f(\mathbf{W}^{(t)}, \mathbf{x}_i) \le \frac{1}{m} \sum_{r=1}^{m} \left( \sigma(\langle \mathbf{w}_{\tilde{y}_i,r}^{(t)}, y_i \boldsymbol{\mu} \rangle) + \sigma(\langle \mathbf{w}_{\tilde{y}_i,r}^{(t)}, \boldsymbol{\xi}_i \rangle) \right)$$

$$\le \frac{1}{m} \sum_{r=1}^{m} \left( \gamma_{\tilde{y}_i,r}^{(t)} + \overline{\rho}_{\tilde{y}_i,r,i}^{(t)} \right) + 2\beta + \widetilde{O}(n\alpha/\sqrt{d})$$

$$\le \frac{1}{m} \sum_{r=1}^{m} \left( \gamma_{\tilde{y}_i,r}^{(t)} + \overline{\rho}_{\tilde{y}_i,r,i}^{(t)} \right) + 1/C_1$$

where we use Lemma D.1 and the Condition 4.1 where we choose a sufficiently large $C_1$.

Similarly, we can lower bound

$$\tilde{y}_i f(\mathbf{W}^{(t)}, \mathbf{x}_i) \ge \frac{1}{m} \sum_{r=1}^{m} \left( \gamma_{\tilde{y}_i,r}^{(t)} + \overline{\rho}_{\tilde{y}_i,r,i}^{(t)} \right) - 1/C_1$$

On the other hand, when $\tilde{y}_i \ne y_i$, it can be shown that

$$\langle \mathbf{w}_{\tilde{y}_i,r}^{(t)}, y_i \boldsymbol{\mu} \rangle \le |\langle \mathbf{w}_{\tilde{y}_i,r}^{(0)}, \boldsymbol{\mu} \rangle| - \gamma_{\tilde{y}_i,r}^{(t)} + \text{SNR} \sqrt{\frac{8 \log(6n/\delta)}{d}} n\alpha \le |\langle \mathbf{w}_{\tilde{y}_i,r}^{(0)}, \boldsymbol{\mu} \rangle| + \text{SNR} \sqrt{\frac{8 \log(6n/\delta)}{d}} n\alpha$$

$$\langle \mathbf{w}_{-\tilde{y}_i,r}^{(t)}, y_i \boldsymbol{\mu} \rangle \le |\langle \mathbf{w}_{-\tilde{y}_i,r}^{(0)}, \boldsymbol{\mu} \rangle| + \gamma_{-\tilde{y}_i,r}^{(t)} + \text{SNR} \sqrt{\frac{8 \log(6n/\delta)}{d}} n\alpha$$

$$\langle \mathbf{w}_{-\tilde{y}_i,r}^{(t)}, y_i \boldsymbol{\mu} \rangle \ge \gamma_{-\tilde{y}_i,r}^{(t)} - |\langle \mathbf{w}_{-\tilde{y}_i,r}^{(0)}, \boldsymbol{\mu} \rangle| - \text{SNR} \sqrt{\frac{8 \log(6n/\delta)}{d}} n\alpha$$

$$\langle \mathbf{w}_{\tilde{y}_i,r}^{(t)}, \boldsymbol{\xi}_i \rangle \le \overline{\rho}_{\tilde{y}_i,r,i}^{(t)} + |\langle \mathbf{w}_{\tilde{y}_i,r}^{(0)}, \boldsymbol{\xi}_i \rangle| + 5 \sqrt{\frac{\log(6n^2/\delta)}{d}} n\alpha$$

where we notice $\gamma_{j,r}^{(t)} \ge 0$.

Then we can upper bound when $\tilde{y}_i \neq y_i$ as

$$
\tilde{y}_i f(\mathbf{W}^{(t)}, \mathbf{x}_i) \leq \frac{1}{m} \sum_{r=1}^{m} \big( \sigma(\langle \mathbf{w}_{\tilde{y}_i,r}^{(t)}, y_i \boldsymbol{\mu} \rangle) + \sigma(\langle \mathbf{w}_{\tilde{y}_i,r}^{(t)}, \boldsymbol{\xi}_i \rangle) \big) - \frac{1}{m} \sum_{r=1}^{m} \sigma(\langle \mathbf{w}_{-\tilde{y}_i,r}^{(t)}, y_i \boldsymbol{\mu} \rangle)
$$

$$
\leq \beta + \mathrm{SNR}\sqrt{\frac{8\log(6n/\delta)}{d}} C_\rho n + \frac{1}{m} \sum_{r=1}^{m} \overline{\rho}_{\tilde{y}_i,r,i}^{(t)} + 5\sqrt{\frac{\log(6n^2/\delta)}{d}} C_\rho n
$$

$$
- \frac{1}{m} \sum_{r=1}^{m} \gamma_{-\tilde{y}_i,r}^{(t)} + \frac{1}{m} \sum_{r=1}^{m} |\langle \mathbf{w}_{-\tilde{y}_i,r}^{(0)}, y_i \boldsymbol{\mu} \rangle| + \mathrm{SNR}\sqrt{\frac{8\log(6n/\delta)}{d}} C_\rho n
$$

$$
\leq \frac{1}{m} \sum_{r=1}^{m} \big( \overline{\rho}_{\tilde{y}_i,r,i}^{(t)} - \gamma_{-\tilde{y}_i,r}^{(t)} \big) + 1/C_1
$$

where the second inequality uses Lemma D.1 and last inequality is by the Condition 4.1.

Similarly, we can lower bound $\tilde{y}_i f(\mathbf{W}^{(t)}, \mathbf{x}_i)$ as

$$
\tilde{y}_i f(\mathbf{W}^{(t)}, \mathbf{x}_i) \geq \frac{1}{m} \sum_{r=1}^{m} \sigma(\langle \mathbf{w}_{\tilde{y}_i,r}^{(t)}, \boldsymbol{\xi}_i \rangle) - \frac{1}{m} \sum_{r=1}^{m} \big( \sigma(\langle \mathbf{w}_{-\tilde{y}_i,r}^{(t)}, y_i \boldsymbol{\mu} \rangle) + \sigma(\langle \mathbf{w}_{-\tilde{y}_i,r}^{(t)}, \boldsymbol{\xi}_i \rangle) \big)
$$

$$
\geq \frac{1}{m} \sum_{r=1}^{m} \big( \overline{\rho}_{\tilde{y}_i,r,i}^{(t)} - \gamma_{\tilde{y}_i,r}^{(t)} \big) - 1/C_1
$$

where we use Lemma D.1. $\qquad\square$

**Lemma D.3.** *Under Condition 4.1 and suppose* (4), (5), (6) *hold at time t. If* $\max_{j,r,i}\{\gamma_{j,r}^{(t)}, \overline{\rho}_{j,r,i}^{(t)}\} = O(1)$, *we have* $\tilde{y}_i f(\mathbf{W}^{(t)}, \mathbf{x}_i) = O(1)$ *and* $\ell_i'^{(t)} = \Omega(1)$ *for all* $i \in [n]$.

*Proof of Lemma D.3.* The proof trivially from Lemma D.2 and the definition of loss. Specifically, we denote the upper bound as $C''$. For $i \in \mathcal{S}_t$, by Lemma D.2,

$$
|\ell_i'^{(t)}| = \frac{1}{1 + \exp(\tilde{y}_i f(\mathbf{W}^{(t)}, \mathbf{x}_i))} \geq \frac{1}{1 + \exp\big(\frac{1}{m}\sum_{r=1}^{m}\big(\gamma_{\tilde{y}_i,r}^{(t)} + \overline{\rho}_{\tilde{y}_i,r,i}^{(t)}\big) + 1/C_1\big)}
$$

$$
\geq \frac{1}{1 + \exp(2C'' + 1/C_1)}
$$

For $i \in \mathcal{S}_f$, by Lemma D.2,

$$
|\ell_i'^{(t)}| = \frac{1}{1 + \exp(\tilde{y}_i f(\mathbf{W}^{(t)}, \mathbf{x}_i))} \geq \frac{1}{1 + \exp\big(\frac{1}{m}\sum_{r=1}^{m}\big(-\gamma_{-\tilde{y}_i,r}^{(t)} + \overline{\rho}_{\tilde{y}_i,r,i}^{(t)}\big) + 1/C_1\big)}
$$

$$
\geq \frac{1}{1 + \exp(C'' + 1/C_1)} > \frac{1}{1 + \exp(2C'' + 1/C_1)}
$$

where the second inequality is by $\gamma_{-\tilde{y}_i,r}^{(t)} \geq 0$. Thus for all $i \in [n]$, we can show that $|\ell_i'^{(t)}| \geq (1 + \exp(2C'' + C_1^{-1}))^{-1}$. $\qquad\square$

Recall $\mathcal{S}_i^{(s)} := \{r \in [m] : \langle \mathbf{w}_{\tilde{y}_i,r}^{(s)}, \boldsymbol{\xi}_i \rangle > 0\}$ and $\mathcal{S}_{j,r}^{(s)} := \{i \in [n] : y_i = j, \langle \mathbf{w}_{j,r}^{(s)}, \boldsymbol{\xi}_i \rangle > 0\}$.

The next lemma shows that in the first stage where the loss derivatives can be lower bounded, the inner product between weights and noise is increasing.

**Lemma D.4.** *Under Condition 4.1 and suppose for any* $t \leq T^*$, (4), (5), (6) *hold for all* $s \leq t$. *Then we can show*

$$
\mathcal{S}_i^{(0)} \subseteq \mathcal{S}_i^{(s)}, \quad \mathcal{S}_{j,r}^{(0)} \subseteq \mathcal{S}_{j,r}^{(s)}.
$$

*for any* $s \leq t$.

*Proof of Lemma D.4.* The proof is by induction where we separately consider two stages. First at $t = 0$, it is trivial to verify that both claims hold. In the first stage where $\max_{j,r,i}\{\gamma_{j,r}^{(t)}, \overline{\rho}_{j,r,i}^{(t)}\} = O(1)$, we can lower bound the loss derivatives by a constant according to Lemma D.3, i.e., $|\ell_i'^{(t)}| \geq C_\ell$ for all $i \in [n]$. Let $T_1$ be the termination time of the first stage. Suppose there exists a time $\tilde{t} \leq T_1$ such that the claims hold for all $s \leq \tilde{t} - 1$, we now prove it also holds at $\tilde{t}$.

By the gradient descent update, for any $r \in \mathcal{S}_i^{(0)}$, we have $r \in \mathcal{S}_i^{(\tilde{t}-1)}$ and thus

$$\langle \mathbf{w}_{\tilde{y}_i,r}^{(\tilde{t})}, \boldsymbol{\xi}_i \rangle = \langle \mathbf{w}_{\tilde{y}_i,r}^{(\tilde{t}-1)}, \boldsymbol{\xi}_i \rangle - \frac{\eta}{nm} \sum_{i'=1}^n \ell_{i'}'^{(\tilde{t}-1)} \cdot \sigma'(\langle \mathbf{w}_{\tilde{y}_i,r}^{(\tilde{t}-1)}, \boldsymbol{\xi}_{i'} \rangle) \cdot \langle \boldsymbol{\xi}_i, \boldsymbol{\xi}_{i'} \rangle$$

$$- \frac{\eta}{nm} \sum_{i'=1}^n \ell_{i'}'^{(\tilde{t}-1)} \cdot \sigma'(\langle \mathbf{w}_{\tilde{y}_i,r}^{(\tilde{t}-1)}, y_{i'}\boldsymbol{\mu} \rangle) \cdot \langle y_{i'}\boldsymbol{\mu}, \boldsymbol{\xi}_i \rangle$$

$$= \langle \mathbf{w}_{\tilde{y}_i,r}^{(\tilde{t}-1)}, \boldsymbol{\xi}_i \rangle \underbrace{- \frac{\eta}{nm} \ell_i'^{(\tilde{t}-1)} \|\boldsymbol{\xi}_i\|_2^2}_{A_1} \underbrace{- \frac{\eta}{nm} \sum_{i' \neq i} \ell_{i'}'^{(\tilde{t}-1)} \sigma'(\langle \mathbf{w}_{\tilde{y}_i,r}^{(\tilde{t}-1)}, \boldsymbol{\xi}_{i'} \rangle) \cdot \langle \boldsymbol{\xi}_i, \boldsymbol{\xi}_{i'} \rangle}_{A_2}$$

$$\underbrace{- \frac{\eta}{nm} \sum_{i'=1}^n \ell_{i'}'^{(\tilde{t}-1)} \cdot \sigma'(\langle \mathbf{w}_{\tilde{y}_i,r}^{(\tilde{t}-1)}, y_{i'}\boldsymbol{\mu} \rangle) \cdot \langle y_{i'}\boldsymbol{\mu}, \boldsymbol{\xi}_i \rangle}_{A_3}.$$

We can respectively bound each term as follows.

$$A_1 \geq \frac{\eta\|\boldsymbol{\xi}_i\|_2^2}{nm} \cdot \min_{i \in [n]} |\ell_i'^{(\tilde{t}-1)}| \geq \frac{\eta\sigma_\xi^2 dC_\ell}{2nm}$$

where the last inequality is by Lemma C.1.

For $A_2$, we can upper bound its magnitude as

$$|A_2| \leq \frac{\eta}{m} \cdot |\langle \boldsymbol{\xi}_i, \boldsymbol{\xi}_{i'} \rangle|$$

$$\leq \frac{2\eta}{m} \cdot \sigma_\xi^2 \sqrt{d\log(6n^2/\delta)}$$

where the first inequality is by $|\ell_i'^{(t)}| \leq 1$ for all $t$ and the second inequality is by Lemma C.1.

For $A_3$, similarly, we can bound

$$|A_3| \leq \frac{\eta}{m} \cdot |\langle \boldsymbol{\mu}, \boldsymbol{\xi}_i \rangle| \leq \frac{\eta\|\boldsymbol{\mu}\|_2\sigma_\xi\sqrt{2\log(6n/\delta)}}{m}$$

where the second inequality is again by Lemma C.1. By requiring $d \geq \max\{32C_\ell^{-2}n^2\log(6n^2/\delta), 4C_\ell^{-1}n\|\boldsymbol{\mu}\|_2\sigma_\xi^{-1}\sqrt{2\log(6n/\delta)}\}$, we can show $A_1 \geq \max\{|A_2|/2, |A_3|/2\}$ and thus

$$\langle \mathbf{w}_{\tilde{y}_i,r}^{(\tilde{t})}, \boldsymbol{\xi}_i \rangle = \langle \mathbf{w}_{\tilde{y}_i,r}^{(\tilde{t}-1)}, \boldsymbol{\xi}_i \rangle \geq \langle \mathbf{w}_{\tilde{y}_i,r}^{(\tilde{t}-1)}, \boldsymbol{\xi}_i \rangle + A_1 - |A_2| - |A_3| > \langle \mathbf{w}_{\tilde{y}_i,r}^{(\tilde{t}-1)}, \boldsymbol{\xi}_i \rangle > 0$$

for all $r \in \mathcal{S}_i^{(\tilde{t}-1)}$. Thus, $r \in \mathcal{S}_i^{(\tilde{t})}$ and $\mathcal{S}_i^{(0)} \subseteq \mathcal{S}_i^{(\tilde{t}-1)} \subseteq \mathcal{S}_i^{(\tilde{t})}$.

For the other claim, we follow a similar strategy as above. For $i \in \mathcal{S}_{j,r}^{(0)}$, we have by induction condition that $i \in \mathcal{S}_{j,r}^{(\tilde{t}-1)}$ and thus for $j = \tilde{y}_i$

$$\langle \mathbf{w}_{j,r}^{(\tilde{t})}, \boldsymbol{\xi}_i \rangle = \langle \mathbf{w}_{j,r}^{(\tilde{t}-1)}, \boldsymbol{\xi}_i \rangle - \frac{\eta}{nm} \sum_{i'=1}^n \ell_{i'}'^{(\tilde{t}-1)} \cdot \sigma'(\langle \mathbf{w}_{j,r}^{(\tilde{t}-1)}, \boldsymbol{\xi}_{i'} \rangle) \cdot \langle \boldsymbol{\xi}_i, \boldsymbol{\xi}_{i'} \rangle$$

$$- \frac{\eta}{nm} \sum_{i'=1}^n \ell_{i'}'^{(\tilde{t}-1)} \cdot \sigma'(\langle \mathbf{w}_{j,r}^{(\tilde{t}-1)}, y_{i'}\boldsymbol{\mu} \rangle) \cdot \langle y_{i'}\boldsymbol{\mu}, \boldsymbol{\xi}_i \rangle$$

Following the same analysis, we can show $\langle \mathbf{w}_{j,r}^{(\tilde{t})}, \boldsymbol{\xi}_i \rangle \geq \langle \mathbf{w}_{j,r}^{(\tilde{t}-1)}, \boldsymbol{\xi}_i \rangle > 0$ and thus $i \in \mathcal{S}_{j,r}^{(\tilde{t})}$ and $\mathcal{S}_{j,r}^{(0)} \subseteq \mathcal{S}_{j,r}^{(\tilde{t}-1)} \subseteq \mathcal{S}_{j,r}^{(\tilde{t})}$.

Now at the end of the first stage where $\overline{\rho}_{j,r,i}^{(T_1)} = \Omega(1)$ for all $j = \tilde{y}_i, r \in \mathcal{S}_i^{(0)}$. Then we continue the proof by induction. Suppose there exists a time $\tilde{t} \geq T_1$ such that for all $T_1 \leq s \leq \tilde{t} - 1, \overline{\rho}_{j,r,i}^{(s)} \geq C_\rho$ for some constant $C_\rho > 0$. Then by the update of $\overline{\rho}_{j,r,i}^{(t)}$, we can show for $j = \tilde{y}_i, r \in \mathcal{S}_i^{(0)}$,

$$\overline{\rho}_{j,r,i}^{(\tilde{t})} = \overline{\rho}_{j,r,i}^{(\tilde{t}-1)} - \frac{\eta}{nm} \ell_i'^{(t)} \sigma'(\langle \mathbf{w}_{j,r}^{(t)}, \boldsymbol{\xi}_i \rangle) \|\boldsymbol{\xi}_i\|_2^2 \geq \overline{\rho}_{j,r,i}^{(\tilde{t}-1)} \geq C_\rho$$

where we notice that $-\ell_i'^{(t)} \geq 0$. Then we can show from Lemma D.1

$$\langle \mathbf{w}_{j,r}^{(\tilde{t})}, \boldsymbol{\xi}_i \rangle \geq \overline{\rho}_{j,r,i}^{(\tilde{t})} - |\langle \mathbf{w}_{j,r}^{(0)}, \boldsymbol{\xi}_i \rangle| - 5\sqrt{\frac{\log(6n^2/\delta)}{d}} n\alpha \geq C_\rho - 1/C' > 0$$

where we use the condition on $d$ to be sufficiently large and choose $C' > 1/C_\rho$. Thus we have for $r \in \mathcal{S}_i^{(\tilde{t})}$ and thus $\mathcal{S}_i^{(0)} \subseteq \mathcal{S}_i^{(\tilde{t}-1)} \subseteq \mathcal{S}_i^{(\tilde{t})}$. For the other claim, we can use the same argument. $\qquad \square$

Next, we proceed to prove Proposition D.1.

*Proof of Proposition D.1.* We prove the claims by induction. It is clear that at $t = 0$, all the claims are satisfied trivially given $\gamma_{j,r}^{(0)}, \overline{\rho}_{j,r,i}^{(0)}, \underline{\rho}_{-j,r,i}^{(0)} = 0$ for all $j, r, i$. Suppose there exists $\tilde{T} \leq T^*$ such that the results in Proposition D.1 hold for all time $t \leq \tilde{T} - 1$. Then we have Lemma D.1, D.2, Lemma D.4 hold for all $t \leq \tilde{T} - 1$.

Now we show that the results in Proposition D.1 also hold for $t = \tilde{T}$.

(1) We first show $\underline{\rho}_{-j,r,i}^{(t)} \geq -\beta - 10\sqrt{\frac{\log(6n^2/\delta)}{d}} n\alpha$. When $\underline{\rho}_{-j,r,i}^{(\tilde{T}-1)} \leq -0.5\beta - 5\sqrt{\frac{\log(6n^2/\delta)}{d}} n\alpha$, by Lemma D.1, we have

$$\langle \mathbf{w}_{j,r}^{(\tilde{T}-1)}, \boldsymbol{\xi}_i \rangle \leq \underline{\rho}_{-j,r,i}^{(\tilde{T}-1)} + |\langle \mathbf{w}_{j,r}^{(0)}, \boldsymbol{\xi}_i \rangle| + 5\sqrt{\frac{\log(6n^2/\delta)}{d}} n\alpha < 0$$

and this suggests

$$\begin{aligned}
\underline{\rho}_{-j,r,i}^{(\tilde{T})} &= \underline{\rho}_{-j,r,i}^{(\tilde{T}-1)} + \frac{\eta}{nm} \ell_i'^{(\tilde{T}-1)} \sigma'(\langle \mathbf{w}_{j,r}^{(\tilde{T}-1)}, \boldsymbol{\xi}_i \rangle) \|\boldsymbol{\xi}_i\|_2^2 \\
&= \underline{\rho}_{-j,r,i}^{(\tilde{T}-1)} \\
&\geq -\beta - 10\sqrt{\frac{\log(6n^2/\delta)}{d}} n\alpha
\end{aligned}$$

On the other hand, when $\underline{\rho}_{-j,r,i}^{(\tilde{T}-1)} \geq -0.5\beta - 5\sqrt{\frac{\log(6n^2/\delta)}{d}} n\alpha$,

$$\begin{aligned}
\underline{\rho}_{-j,r,i}^{(\tilde{T})} &= \underline{\rho}_{-j,r,i}^{(\tilde{T}-1)} + \frac{\eta}{nm} \ell_i'^{(\tilde{T}-1)} \sigma'(\langle \mathbf{w}_{j,r}^{(\tilde{T}-1)}, \boldsymbol{\xi}_i \rangle) \|\boldsymbol{\xi}_i\|_2^2 \\
&\geq -0.5\beta - 5\sqrt{\frac{\log(6n^2/\delta)}{d}} C_\rho n - \frac{3\eta \sigma_\xi^2 d}{2nm} \\
&\geq -0.5\beta - 10\sqrt{\frac{\log(6n^2/\delta)}{d}} C_\rho n \\
&\geq -\beta - 10\sqrt{\frac{\log(6n^2/\delta)}{d}} C_\rho n
\end{aligned}$$

where we use Lemma C.1 in the first inequality. The second inequality is by the condition on $\eta$ such that $5\sqrt{\frac{\log(6n^2/\delta)}{d}} C_\rho n \geq 3\eta \sigma_\xi^2 d/(2nm)$. This completes the induction for the result on $\underline{\rho}_{-j,r,i}^{(t)}$.

(2) We next prove $\gamma_{j,r}^{(\bar{T})} \geq 0$. Towards this end, we separate the analysis in two stages. In the first stage, the loss derivatives can be lower bounded by a constant, i.e., $|\ell_i^{\prime(t)}| \geq C_\ell$ for all $i \in [n]$. Recall the update rule for $\gamma_{j,r}^{(t)}$ is

$$\gamma_{j,r}^{(\bar{T})} = \gamma_{j,r}^{(\bar{T}-1)} - \frac{\eta}{nm} \sum_{i=1}^{n} \ell_i^{\prime(\bar{T}-1)} \sigma'(\langle \mathbf{w}_{j,r}^{(\bar{T}-1)}, y_i\boldsymbol{\mu}\rangle) y_i \tilde{y}_i \|\boldsymbol{\mu}\|_2^2.$$

When $\langle \mathbf{w}_{j,r}^{(\bar{T}-1)}, \boldsymbol{\mu}\rangle \geq 0$, we can show

$$\gamma_{j,r}^{(\bar{T})} = \gamma_{j,r}^{(\bar{T}-1)} - \frac{\eta}{nm}\Big(\sum_{i \in \mathcal{S}_t \cap \mathcal{S}_1} \ell_i^{\prime(\bar{T}-1)} - \sum_{i \in \mathcal{S}_f \cap \mathcal{S}_1} \ell_i^{\prime(\bar{T}-1)}\Big)\|\boldsymbol{\mu}\|_2^2$$

$$\geq \gamma_{j,r}^{(\bar{T}-1)} + \frac{\eta}{nm}\Big(\frac{1-\tau_+}{2}C_\ell - \frac{\tau_+}{2}\Big)\|\boldsymbol{\mu}\|_2^2$$

$$\geq \gamma_{j,r}^{(\bar{T}-1)}$$

$$\geq 0$$

where in the first inequality, we uses $C_\ell \leq |\ell_i^{\prime(t)}| \leq 1$. The second inequality is by the choice $\tau_+ \leq \frac{C_\ell}{C_\ell+1}$.

Similarly, when $\langle \mathbf{w}_{j,r}^{(\bar{T}-1)}, \boldsymbol{\mu}\rangle \leq 0$, we have

$$\gamma_{j,r}^{(\bar{T})} = \gamma_{j,r}^{(\bar{T}-1)} - \frac{\eta}{nm}\Big(\sum_{i \in \mathcal{S}_t \cap \mathcal{S}_{-1}} \ell_i^{\prime(\bar{T}-1)} - \sum_{i \in \mathcal{S}_f \cap \mathcal{S}_{-1}} \ell_i^{\prime(\bar{T}-1)}\Big)\|\boldsymbol{\mu}\|_2^2$$

$$\geq \gamma_{j,r}^{(\bar{T}-1)} + \frac{\eta}{nm}\Big(\frac{1-\tau_-}{2}C_\ell - \frac{\tau_-}{2}\Big)\|\boldsymbol{\mu}\|_2^2$$

$$\geq \gamma_{j,r}^{(\bar{T}-1)}$$

$$\geq 0$$

where we choose $\tau_- \leq \frac{C_\ell}{C_\ell+1}$.

In the second stage, we prove the claim by contradiction. First, without loss of generality that $\langle \mathbf{w}_{j,r}^{(t)}, \boldsymbol{\mu}\rangle \geq 0$, and we write the update as

$$\gamma_{j,r}^{(t+1)} = \gamma_{j,r}^{(t)} - \frac{\eta}{nm}\Big(\sum_{i \in \mathcal{S}_t \cap \mathcal{S}_1} \ell_i^{\prime(t)} - \sum_{i \in \mathcal{S}_f \cap \mathcal{S}_1} \ell_i^{\prime(t)}\Big)\|\boldsymbol{\mu}\|_2^2$$

$$= \gamma_{j,r}^{(t)} + \frac{\eta}{nm}\Big(\underbrace{\sum_{i \in \mathcal{S}_t \cap \mathcal{S}_1} \frac{1}{1 + \exp\big(\tilde{y}_i f(\mathbf{W}^{(t)}, \mathbf{x}_i)\big)} - \sum_{i \in \mathcal{S}_f \cap \mathcal{S}_1} \frac{1}{1 + \exp\big(\tilde{y}_i f(\mathbf{W}^{(t)}, \mathbf{x}_i)\big)}}_{A_4}\Big)\|\boldsymbol{\mu}\|_2^2$$

Suppose at an iteration $t$, $A_4 < 0$, which leads to a decrease in the $\gamma_{j,r}^{(t)}$. Then by Lemma D.2

$$A_4 \geq \sum_{i \in \mathcal{S}_t \cap \mathcal{S}_1} \frac{1}{1 + \exp\big(\frac{1}{m}\sum_{r=1}^{m}(\overline{\rho}_{\tilde{y}_i,r,i}^{(t)} + \gamma_{\tilde{y}_i,r}^{(t)}) + 1/C_1\big)}$$

$$- \sum_{i \in \mathcal{S}_f \cap \mathcal{S}_1} \frac{1}{1 + \exp\big(\frac{1}{m}\sum_{r=1}^{m}(\overline{\rho}_{\tilde{y}_i,r,i}^{(t)} - \gamma_{-\tilde{y}_i,r}^{(t)}) - 1/C_1\big)}$$

Then we can see the gap between loss derivatives of $\mathcal{S}_t$ and $\mathcal{S}_f$ becomes progressively smaller such that for a given $\tau_+$ (or $\tau_-$ when $\langle \mathbf{w}_{j,r}^{(t)}, \boldsymbol{\mu}\rangle \leq 0$) which is sufficiently small, $A_4 > 0$ and $\gamma_{j,r}^{(t)}$ starts to increase.

(3) Next we show upper bound for $\overline{\rho}_{\tilde{y}_i,r,i}^{(t)}$. Recall the update rule for $\overline{\rho}_{j,r,i}^{(t)}$ is

$$\overline{\rho}_{\tilde{y}_i,r,i}^{(t+1)} = \overline{\rho}_{\tilde{y}_i,r,i}^{(t)} - \frac{\eta}{nm}\ell_i^{\prime(t)}\sigma'(\langle \mathbf{w}_{j,r}^{(t)}, \boldsymbol{\xi}_i\rangle)\|\boldsymbol{\xi}_i\|_2^2$$

Now suppose $t_{r,i}$ be the last time $t < T^*$ such that $\overline{\rho}_{\tilde{y}_i,r,i}^{(t)} \leq 0.5\alpha$. Then

$$\overline{\rho}_{\tilde{y}_i,r,i}^{(\widetilde{T})} = \overline{\rho}_{\tilde{y}_i,r,i}^{(t_{r,i})} - \frac{\eta}{nm}\ell_i'^{(t)}\sigma'(\langle \mathbf{w}_{\tilde{y}_i,r}^{(t_{r,i})}, \boldsymbol{\xi}_i\rangle)\|\boldsymbol{\xi}_i\|_2^2 - \frac{\eta}{nm}\sum_{t_{r,i}<t<\widetilde{T}}\ell_i'^{(t)}\sigma'(\langle \mathbf{w}_{\tilde{y}_i,r}^{(t)}, \boldsymbol{\xi}_i\rangle)\|\boldsymbol{\xi}_i\|_2^2$$

$$\leq \overline{\rho}_{\tilde{y}_i,r,i}^{(t_{r,i})} + \frac{3\eta\sigma_\xi^2 d}{2nm} - \frac{\eta}{nm}\sum_{t_{r,i}<t<\widetilde{T}}\ell_i'^{(t)}\sigma'(\langle \mathbf{w}_{\tilde{y}_i,r}^{(t)}, \boldsymbol{\xi}_i\rangle)\|\boldsymbol{\xi}_i\|_2^2$$

$$\leq 0.5\alpha + 0.25\alpha - \frac{\eta}{nm}\sum_{t_{r,i}<t<\widetilde{T}}\ell_i'^{(t)}\sigma'(\langle \mathbf{w}_{\tilde{y}_i,r}^{(t)}, \boldsymbol{\xi}_i\rangle)\|\boldsymbol{\xi}_i\|_2^2 \qquad (7)$$

where we apply Lemma C.1 for the first inequality and choose $\eta \leq C^{-1}n\sigma_\xi^{-2}d^{-1}$ for the last inequality. Then we bound the last term for $t \in (t_{r,i}, \widetilde{T})$ as

$$-\ell_i'^{(t)} = \frac{1}{1 + \exp(\tilde{y}_i f(\mathbf{W}^{(t)}, \mathbf{x}_i))} \leq \exp(-\tilde{y}_i f(\mathbf{W}^{(t)}, \mathbf{x}_i))$$

Next we consider two cases depending on whether $i \in \mathcal{S}_t$ or $i \in \mathcal{S}_f$.

- When $i \in \mathcal{S}_t$, we can bound by Lemma D.2

$$\tilde{y}_i f(\mathbf{W}^{(t)}, \mathbf{x}_i) \geq \frac{1}{m}\sum_{r=1}^m(\gamma_{\tilde{y}_i,r}^{(t)} + \overline{\rho}_{\tilde{y}_i,r,i}^{(t)}) - 1/C_1 \geq \frac{1}{m}\sum_{r=1}^m\overline{\rho}_{\tilde{y}_i,r,i}^{(t)} - 1/C_1 \geq 0.5\alpha - 0.1.$$

   where the second inequality is by $\gamma_{\tilde{y}_i,r}^{(t)} \geq 0$ and the last inequality is by choosing $C_1 \geq 10$. Then this suggests

$$-\ell_i'^{(t)} \leq \exp(-\tilde{y}_i f(\mathbf{W}^{(t)}, \mathbf{x}_i)) \leq 2\exp(-0.5\alpha) \leq 2/T^*$$

   where the last inequality is by choosing $C_t \geq 2$.

- When $i \in \mathcal{S}_f$, we can bound by Lemma D.2

$$\tilde{y}_i f(\mathbf{W}^{(t)}, \mathbf{x}_i) \geq \frac{1}{m}\sum_{r=1}^m(\overline{\rho}_{\tilde{y}_i,r,i}^{(t)} - \gamma_{-\tilde{y}_i,r}^{(t)}) - 1/C_1 \geq \frac{1}{m}\sum_{r=1}^m\overline{\rho}_{\tilde{y}_i,r,i}^{(t)} - 1/C_1 \geq 0.5\alpha - 0.1.$$

   Here we only consider the case when $\frac{1}{m}\sum_{r=1}^m\overline{\rho}_{\tilde{y}_i,r,i}^{(t)} > \frac{1}{m}\sum_{r=1}^m\gamma_{-\tilde{y}_i,r}^{(t)}$ when deriving the upper bound for $\frac{1}{m}\sum_{r=1}^m\overline{\rho}_{\tilde{y}_i,r,i}^{(t)}$ because otherwise, the loss cannot converge to arbitrarily small as we show later. To see this, we suppose $\frac{1}{m}\sum_{r=1}^m\overline{\rho}_{\tilde{y}_i,r,i}^{(T^*)} \leq \frac{1}{m}\sum_{r=1}^m\gamma_{-\tilde{y}_i,r}^{(T^*)}$ at termination time. Then for such sample, $\tilde{y}_i f(\mathbf{W}^{(t)}, \mathbf{x}_i) \leq \frac{1}{m}\sum_{r=1}^m(\overline{\rho}_{\tilde{y}_i,r,i}^{(T^*)} - \gamma_{-\tilde{y}_i,r}^{(T^*)}) + 1/C_1 \leq 0.1$ the loss can be lower bounded as $\ell_i^{(T^*)} = \log(1 + \exp(-\tilde{y}_i f(\mathbf{W}^{(T^*)}, \mathbf{x}_i))) \geq \log(1 + \exp(-0.1)) \geq 0.6$.

   Hence we let $c' := (\frac{1}{m}\sum_{r=1}^m\overline{\rho}_{\tilde{y}_i,r,i}^{(t)})/(\frac{1}{m}\sum_{r=1}^m\gamma_{-\tilde{y}_i,r}^{(t)}) > 1$. Then

$$\tilde{y}_i f(\mathbf{W}^{(t)}, \mathbf{x}_i) \geq \frac{1}{m}\sum_{r=1}^m(1 - 1/c')\overline{\rho}_{\tilde{y}_i,r,i}^{(t)} - 1/C_1 \geq (1 - 1/c')0.5\alpha - 0.1.$$

   Then we have

$$-\ell_i'^{(t)} \leq \exp(-\tilde{y}_i f(\mathbf{W}^{(t)}, \mathbf{x}_i)) \leq 2\exp\left(-(1 - 1/c')0.5\alpha\right) \leq 2/T^*$$

   where the last inequality is by choosing $C_t$ sufficiently large.

In both cases, (7) can be further bounded as

$$\overline{\rho}_{\tilde{y}_i,r,i}^{(\widetilde{T})} \leq 0.75\alpha + \frac{3\eta\sigma_\xi^2 dT^*}{2nm} \cdot \frac{2}{T^*} \leq \alpha$$

where the first inequality is by upper bound on the loss derivatives and the last inequality is by the condition on Condition 4.1 where $\eta \leq C^{-1}n\sigma_\xi^{-2}d^{-1} \leq nm\sigma_\xi^{-2}d^{-1}/3$.

(4) Finally for the upper bound on $\gamma_{j,r}^{(t)}$, we can verify that by the update of $\gamma_{j,r}^{(t)}$

$$
\gamma_{j,r}^{(t+1)} = \gamma_{j,r}^{(t)} - \frac{\eta}{nm}\Big(\sum_{i\in\mathcal{S}_t}\ell_i'^{(t)}\mathbb{1}(\langle\mathbf{w}_{j,r}^{(t)}, y_i\boldsymbol{\mu}\rangle \geq 0) - \sum_{i\in\mathcal{S}_f}\ell_i'^{(t)}\mathbb{1}(\langle\mathbf{w}_{j,r}^{(t)}, y_i\boldsymbol{\mu}\rangle \geq 0)\Big)\|\boldsymbol{\mu}\|_2^2
$$

$$
\leq \gamma_{j,r}^{(t)} + \frac{\eta}{nm}\sum_{i\in\mathcal{S}_t}|\ell_i'^{(t)}|\|\boldsymbol{\mu}\|_2^2
$$

$$
\leq \gamma_{j,r}^{(t)} + \frac{\eta\|\boldsymbol{\mu}\|_2^2}{nm}\sum_{i\in\mathcal{S}_t}\frac{1}{1+\exp\left(\frac{1}{m}\sum_{r=1}^m(\overline{\rho}_{\tilde{y}_i,r,i}^{(t)} + \gamma_{\tilde{y}_i,r}^{(t)}) - 1/C_1\right)}
$$

Suppose $t_{j,r}$ be the last time $t < T^*$ such that $\gamma_{j,r}^{(t)} \leq 0.5C_\gamma\alpha$. Then

$$
\gamma_{j,r}^{(\tilde{T})} \leq \gamma_{j,r}^{(t_{j,r})} + \frac{\eta(2-\tau_+-\tau_-)\|\boldsymbol{\mu}\|_2^2}{2m}T^*\frac{1}{1+\exp(0.5C_\gamma C_t\log(T^*)-0.1)}
$$

$$
\leq \gamma_{j,r}^{(t_{j,r})} + \frac{\eta(2-\tau_+-\tau_-)\|\boldsymbol{\mu}\|_2^2}{2m}T^*\cdot 2\exp(-0.5C_\gamma C_t\log(T^*))
$$

$$
\leq \gamma_{j,r}^{(t_{j,r})} + \frac{\eta(2-\tau_+-\tau_-)\|\boldsymbol{\mu}\|_2^2}{m}
$$

$$
\leq C_\gamma\alpha
$$

where we notice $\overline{\rho}_{j,r,i}^{(t)} \geq 0$ and we let $C_1 \geq 10, C_\gamma \geq 1$. The last inequality follows from Condition 4.1 where $\eta \leq C^{-1}n\sigma_\xi^{-2}d^{-1} \leq 0.5(2-\tau_+-\tau_-)^{-1}m\|\boldsymbol{\mu}\|_2^{-2}$, where the last inequality is by condition on $\|\boldsymbol{\mu}\|_2^2$ and $d$. Thus the proof is now complete. $\square$

### D.3 FIRST STAGE

Next we consider first stage of the training dynamics. In this stage, before the coefficients $\gamma_{j,r}^{(t)}, \overline{\rho}_{j,r,i}^{(t)}$ reach a constant order, we can both lower and upper bound the loss derivatives by an absolute constant, i.e., $\underline{C}_\ell \leq |\ell_i'^{(t)}| \leq \overline{C}_\ell$. Here we suggests $\underline{C}_\ell = 0.49$ and $\overline{C}_\ell = 0.51$ is sufficient to show the desired result.

Before we proceed to prove Theorem 4.1, we provide a tighter bound on $\|\boldsymbol{\xi}_i\|_2^2$ and $|\mathcal{S}_i^{(0)}|$ compared to Lemma C.1 and Lemma C.3 respectively.

**Lemma D.5.** *With probability at least $1 - \delta$, we can bound*

$$
\sigma_\xi^2 d(1 - \widetilde{O}(1/\sqrt{d})) \leq \|\boldsymbol{\xi}_i\|_2^2 \leq \sigma_\xi^2 d(1 + \widetilde{O}(1/\sqrt{d}))
$$

*Proof.* By Bernstein inequality, with probability at least $1 - \delta/n$, we have $\big|\|\boldsymbol{\xi}_i\|_2^2 - \sigma_p^2 d\big| = O\left(\sigma_p^2\cdot\sqrt{d\log(6n/\delta)}\right)$. Then taking the union bound gives the desired result. $\square$

**Lemma D.6.** *With probability at least $1 - \delta$, we can bound*

$$
\frac{m}{2}\big(1 - \widetilde{O}(1/\sqrt{m})\big) \leq |\mathcal{S}_i^{(0)}| \leq \frac{m}{2}\big(1 + \widetilde{O}(1/\sqrt{m})\big)
$$

*Proof.* Because $\mathbb{P}(\langle\mathbf{w}_{\tilde{y}_i,r}^{(0)}, \boldsymbol{\xi}_i\rangle > 0) = 0.5$, by Hoeffding inequality, with probability at least $1 - \delta/n$, we can bound $\big|\frac{|\mathcal{S}_i^{(0)}|}{m} - \frac{1}{2}\big| \leq \sqrt{\frac{\log(2n/\delta)}{2m}}$. Then taking union bound gives the desired result. $\square$

**Theorem D.1** (Restatement of Theorem 4.1). *Under Condition 4.1, there exists $T_1 = \Theta\big(\eta^{-1}nm\sigma_\xi^{-2}d^{-1}\big)$ such that*

1. $\overline{\rho}_{\tilde{y}_i,r,i}^{(T_1)} = \Theta(1)$ *for all $i \in [n]$, $r \in [m]$ such that $\langle\mathbf{w}_{\tilde{y}_i,r}^{(0)}, \mathbf{x}_i\rangle \geq 0$.*

2. $\gamma_{j,r}^{(T_1)} = \Theta(1)$ *for all $j = \pm 1, r \in [m]$.*

3. $\gamma_{j,r}^{(T_1)} > \bar{\rho}_{\tilde{y}_i,r,i}^{(T_1)}$ for all $j = \pm 1, r \in [m], i \in [n]$.

4. All clean samples $i \in \mathcal{S}_t$ satisfy that $\tilde{y}_i f(\mathbf{W}^{(T_1)}, \mathbf{x}_i) \geq 0$.

5. All noisy samples $i \in \mathcal{S}_f$ satisfy that $\tilde{y}_i f(\mathbf{W}^{(T_1)}, \mathbf{x}_i) \leq 0$.

*Proof of Theorem D.1.* We first show the lower and upper bound for noise dynamics. For any $i \in [n]$ and $r \in \mathcal{S}_i^{(0)}$, by Lemma D.4, we know that $r \in \mathcal{S}_i^{(t)}$ for all $t \leq T_1$. Hence, by the update of noise coefficients,

$$\bar{\rho}_{\tilde{y}_i,r,i}^{(t)} = \bar{\rho}_{\tilde{y}_i,r,i}^{(t-1)} - \frac{\eta}{nm}\ell_i'^{(t-1)}\|\boldsymbol{\xi}_i\|_2^2 \geq \bar{\rho}_{\tilde{y}_i,r,i}^{(t-1)} + 0.99 \cdot \frac{\eta C_\ell \sigma_\xi^2 d}{nm} = 0.99 \cdot \frac{\eta C_\ell \sigma_\xi^2 d}{nm}t$$

where the first inequality is by Lemma D.5 where we choose $d = \Omega(\log(n/\delta))$ sufficiently large and the loss derivative lower bound in this stage. The second inequality is by iterating the first inequality to $t = 0$ and by noticing $\bar{\rho}_{\tilde{y}_i,r,i}^{(0)} = 0$.

For the upper bound, for $i \in [n]$,

$$\bar{\rho}_{\tilde{y}_i,r,i}^{(t)} = \bar{\rho}_{\tilde{y}_i,r,i}^{(t-1)} + \frac{\eta}{nm}|\ell_i'^{(t-1)}|\|\boldsymbol{\xi}_i\|_2^2 \leq \bar{\rho}_{\tilde{y}_i,r,i}^{(t-1)} + 1.01 \cdot \frac{\eta \sigma_\xi^2 d}{nm} \leq 1.01 \cdot \frac{\eta \sigma_\xi^2 d}{nm}t$$

where we use Lemma D.5 and $|\ell_i'^{(t)}| \leq 1$.

Next, we lower and upper bound the signal dynamics. Recall the update rule for $\gamma_{j,r}^{(t)}$ as

$$\gamma_{j,r}^{(t)} = \gamma_{j,r}^{(t-1)} - \frac{\eta}{nm}\sum_{i=1}^n \ell_i'^{(t)}\sigma'(\langle \mathbf{w}_{j,r}^{(t-1)}, y_i\boldsymbol{\mu}\rangle)y_i\tilde{y}_i\|\boldsymbol{\mu}\|_2^2$$

$$= \gamma_{j,r}^{(t-1)} - \frac{\eta}{nm}\Big(\sum_{i\in\mathcal{S}_t}\ell_i'^{(t)}\sigma'(\langle \mathbf{w}_{j,r}^{(t-1)}, y_i\boldsymbol{\mu}\rangle) - \sum_{i\in\mathcal{S}_f}\ell_i'^{(t)}\sigma'(\langle \mathbf{w}_{j,r}^{(t-1)}, y_i\boldsymbol{\mu}\rangle)\Big)\|\boldsymbol{\mu}\|_2^2$$

When $\langle \mathbf{w}_{j,r}^{(t-1)}, \boldsymbol{\mu}\rangle \geq 0$,

$$\gamma_{j,r}^{(t)} = \gamma_{j,r}^{(t-1)} - \frac{\eta}{nm}\Big(\sum_{i\in\mathcal{S}_t\cap\mathcal{S}_1}\ell_i'^{(t-1)} - \sum_{i\in\mathcal{S}_f\cap\mathcal{S}_1}\ell_i'^{(t-1)}\Big)\|\boldsymbol{\mu}\|_2^2$$

$$\geq \gamma_{j,r}^{(t-1)} + \frac{\eta}{nm}\Big(\frac{(1-\tau_+)nC_\ell}{2} - \frac{\tau_+ n}{2}\Big)\|\boldsymbol{\mu}\|_2^2$$

$$\geq \gamma_{j,r}^{(t-1)} + 0.49 \cdot \frac{\eta\|\boldsymbol{\mu}\|_2^2 C_\ell}{m}$$

where the first inequality uses the lower bound and upper bound on loss derivatives, i.e., $C_\ell \leq |\ell_i'^{(t)}| \leq 1$. The second inequality follows by letting $\tau_+ \leq \frac{0.02C_\ell}{1+C_\ell}$.

Similarly, when $\langle \mathbf{w}_{j,r}^{(t-1)}, \boldsymbol{\mu}\rangle < 0$,

$$\gamma_{j,r}^{(t)} = \gamma_{j,r}^{(t-1)} - \frac{\eta}{nm}\Big(\sum_{i\in\mathcal{S}_t\cap\mathcal{S}_{-1}}\ell_i'^{(t-1)} - \sum_{i\in\mathcal{S}_f\cap\mathcal{S}_{-1}}\ell_i'^{(t-1)}\Big)\|\boldsymbol{\mu}\|_2^2$$

$$\geq \gamma_{j,r}^{(t-1)} + \frac{\eta}{nm}\Big(\frac{(1-\tau_-)nC_\ell}{2} - \frac{\tau_- n}{2}\Big)\|\boldsymbol{\mu}\|_2^2$$

$$\geq \gamma_{j,r}^{(t-1)} + 0.49 \cdot \frac{\eta\|\boldsymbol{\mu}\|_2^2 C_\ell}{m}$$

where we let $\tau_- \leq \frac{0.02C_\ell}{1+C_\ell}$. Combining both cases, we can iterate the inequality, which gives

$$\gamma_{j,r}^{(t)} \geq \gamma_{j,r}^{(0)} + \frac{\eta\|\boldsymbol{\mu}\|_2^2 C_\ell}{4m}t = 0.49 \cdot \frac{\eta\|\boldsymbol{\mu}\|_2^2 C_\ell}{m}t$$

We first show the claim that $\tilde{y}_i f(\mathbf{W}^{(T_1)}, \mathbf{x}_i) \geq 0$ for $i \in \mathcal{S}_t$. By the update of signal and noise coefficients, we have for all any $i \in \mathcal{S}_t$, and $r \in \mathcal{S}_i^{(0)}$, by , we have $r \in \mathcal{S}_i^{(t)}$ and thus for all $t \leq T_1$,

For the upper bound, we obtain from the signal dynamics that

$$\gamma_{j,r}^{(t)} = \gamma_{j,r}^{(t-1)} - \frac{\eta}{nm}\Big(\sum_{i\in\mathcal{S}_t}\ell_i'^{(t)}\sigma'(\langle\mathbf{w}_{j,r}^{(t-1)}, y_i\boldsymbol{\mu}\rangle) - \sum_{i\in\mathcal{S}_f}\ell_i'^{(t)}\sigma'(\langle\mathbf{w}_{j,r}^{(t-1)}, y_i\boldsymbol{\mu}\rangle)\Big)\|\boldsymbol{\mu}\|_2^2$$

$$\leq \gamma_{j,r}^{(t-1)} + \frac{\eta\|\boldsymbol{\mu}\|_2^2}{m}\frac{1-\tau_\pm}{2}$$

$$\leq \gamma_{j,r}^{(t-1)} + \frac{\eta\|\boldsymbol{\mu}\|_2^2}{2m}$$

$$\leq \frac{\eta\|\boldsymbol{\mu}\|_2^2}{2m}t$$

where we use the upper bound on loss derivative in the first inequality.

Now we verify the conditions such that the claims are satisfied. First, we set a termination time for the first stage as $T_1$, where

$$T_1 = C_2\eta^{-1}C_\ell^{-1}nm\sigma_\xi^{-2}d^{-1}$$

for some constant $C_2 > 0$ to be chosen later. This suggests at the end of first stage, we have

- $\overline{\rho}_{\tilde{y}_i,r,i}^{(T_1)} \geq 0.99 \cdot C_2$, for all $i \in [n]$ and $r \in \mathcal{S}_i^{(0)}$

- $\overline{\rho}_{\tilde{y}_i,r,i}^{(T_1)} \leq 1.01 \cdot C_2 C_\ell^{-1}$ for all $i \in [n]$.

- $\gamma_{j,r}^{(T_1)} \geq 0.49 C_2 n \cdot \text{SNR}^2$ for all $j = \pm 1, r \in [m]$.

- $\gamma_{j,r}^{(T_1)} \leq 0.5 C_2 n \cdot \text{SNR}^2$ for all $j = \pm 1, r \in [m]$.

Then for all $i \in \mathcal{S}_t$, we have by Lemma D.2,

$$\tilde{y}_i f(\mathbf{W}^{(t)}, \mathbf{x}_i) \geq \frac{1}{m}\sum_{r=1}^m \big(\gamma_{\tilde{y}_i,r}^{(t)} + \overline{\rho}_{\tilde{y}_i,r,i}^{(t)}\big) - 1/C_1 \geq 0.49 C_2 n \cdot \text{SNR}^2 + 0.99 C_2 - 1/C_1 > 0$$

where the last inequality is by choosing $C_1$ sufficiently large, e.g., $C_1 \geq (0.99 C_2)^{-1}$. This verifies the first claim of Theorem 4.1.

Second, for all $i \in \mathcal{S}_f$, we have by Lemma D.2,

$$\tilde{y}_i f(\mathbf{W}^{(t)}, \mathbf{x}_i) \leq \frac{1}{m}\sum_{r=1}^m \big(-\gamma_{-\tilde{y}_i,r}^{(t)} + \overline{\rho}_{\tilde{y}_i,r,i}^{(t)}\big) + 1/C_1 \leq -0.49 C_2 n \cdot \text{SNR}^2 + 1.01 C_2 C_\ell^{-1} + 1/C_1$$

$$< 0$$

where the last inequality is by the choice that $C_1 \geq 1000$ and

$$n \cdot \text{SNR}^2 \geq 2.07 C_\ell^{-1} + 0.002 C_2^{-1} \tag{8}$$

Under such condition, we also verify the third claim of Theorem D.1.

It remains to analyze the condition under which the loss derivative is lower bounded by $C_\ell$. In particular, we require $\min_{i\in[n],t\leq T_1}|\ell_i'^{(t)}| \geq C_\ell$, where

$$\min_{i\in[n],t\leq T_1}|\ell_i'^{(t)}| = \min_{i\in\mathcal{S}_t}|\ell_i'^{(T_1)}| = \frac{1}{1 + \exp(\max_{i\in\mathcal{S}_t}\tilde{y}_i f(\mathbf{W}^{(T_1)}, \mathbf{x}_i))}$$

$$\geq \frac{1}{1 + \exp\big(\frac{1}{m}\sum_{r=1}^m(\gamma_{\tilde{y}_{i^*},r}^{(T_1)} + \overline{\rho}_{\tilde{y}_{i^*},r,i^*}^{(T_1)}) + 1/C_1\big)}$$

$$\geq \frac{1}{1 + \exp\big(1.01 C_2 C_\ell^{-1} + 0.5 C_2 n \cdot \text{SNR}^2 + 1/C_1\big)}$$

where we denote $i^* = \arg\max_{i \in \mathcal{S}_t} \tilde{y}_i f(\mathbf{W}^{(T_1)}, \mathbf{x}_i)$ and apply Lemma D.2.

Thus to ensure $\min_{i \in [n], t \leq T_1} |\ell_i'^{(t)}| \geq C_\ell$, we require $1.01 C_2 C_\ell^{-1} + 0.5 C_2 n \cdot \mathrm{SNR}^2 + 1/C_1 \leq \log(C_\ell^{-1} - 1)$, which translates to

$$n \cdot \mathrm{SNR}^2 \leq 2 \log(C_\ell^{-1} - 1) C_2^{-1} - 2.02 C_\ell^{-1} - 0.002 C_2^{-1} \tag{9}$$

where we choose $C_1 \geq 1000$.

The final step is to show there exists a combination of $C_2$, $n \cdot \mathrm{SNR}^2$ and $C_\ell$ such that conditions (8) and (9) are satisfied. For this, we can fix $C_\ell = 0.4$ for example and thus

$$5.175 + 0.002 C_2^{-1} \leq n \cdot \mathrm{SNR}^2 \leq 0.34 C_2^{-1} - 5.05$$

Then let $C_2 = 1/31$, we have $5.237 \leq n \cdot \mathrm{SNR}^2 \leq 5.49$. Thus the proof is complete. $\qquad\square$

### D.4 SECOND STAGE

The second stage aims to show convergence of the training dynamics. By the end of the first stage, without loss of generality, we set $C_2 = 2.1$ and we can see

- For all $i \in [n]$, $r \in \mathcal{S}_i^{(0)}$, we have $\overline{\rho}_{\tilde{y}_i, r, i}^{(T_1)} \geq 2$;
- For all $j = \pm 1$, $r \in [m]$, we have $\gamma_{j,r}^{(t)} = \Omega(n \cdot \mathrm{SNR}^2)$, where $n \cdot \mathrm{SNR}^2 = \Theta(1)$.
- $\max_{j,r,i} |\underline{\rho}_{j,r,i}^{(T_1)}| \leq 1/C$ for some sufficiently large constant $C > 0$.

In the second stage, we show that in order to achieve convergence in loss to arbitrary tolerance, noise coefficients for noisy samples would first surpass signal coefficients by a large margin. To this end, we first show convergence in the loss function. The proof mainly follows from the analysis of Kou et al. (2023). Nevertheless, we again highlight a critical difference is that in our case, $n \cdot \mathrm{SNR}^2 = \Theta(1)$.

First, we let $\mathbf{w}_{j,r}^* = \mathbf{w}_{j,r}^{(0)} + 5 \log(2/\epsilon) \sum_{i=1}^n \frac{\boldsymbol{\xi}_i}{\|\boldsymbol{\xi}_i\|_2^2} \mathbb{1}(\tilde{y}_i = j)$.

**Lemma D.7.** *Under Condition 4.1, we can show* $\|\mathbf{W}^{(T_1)} - \mathbf{W}^*\|_F \leq \widetilde{O}(m^{1/2} n^{1/2} \sigma_\xi^{-1} d^{-1/2})$.

*Proof of Lemma D.7.* From the decomposition at $T_1$, we can show

$$\|\mathbf{W}^{(T_1)} - \mathbf{W}^*\|_F \leq \|\mathbf{W}^{(T_1)} - \mathbf{W}^{(0)}\|_F + \|\mathbf{W}^* - \mathbf{W}^{(0)}\|_F$$

$$\leq O(\sqrt{m}) \max_{j,r} \gamma_{j,r}^{(T_1)} \|\boldsymbol{\mu}\|_2^{-1} + O(\sqrt{m}) \max_{j,r} \| \sum_{i=1}^n \overline{\rho}_{j,r,i}^{(T_1)} \cdot \frac{\boldsymbol{\xi}_i}{\|\boldsymbol{\xi}_i\|_2^2} + \sum_{i=1}^n \underline{\rho}_{j,r,i}^{(T_1)} \cdot \frac{\boldsymbol{\xi}_i}{\|\boldsymbol{\xi}_i\|_2^2} \|_2$$

$$+ O(m^{1/2} n^{1/2} \log(1/\epsilon) \sigma_\xi^{-1} d^{-1/2})$$

$$= O(m^{1/2} n \cdot \mathrm{SNR}^2 \|\boldsymbol{\mu}\|_2^{-1}) + \widetilde{O}(m^{1/2} n^{1/2} \sigma_\xi^{-1} d^{-1/2})$$

$$= \widetilde{O}(m^{1/2} n^{1/2} \sigma_\xi^{-1} d^{-1/2})$$

where the last inequality is by $n \cdot \mathrm{SNR}^2 = \Theta(1)$. $\qquad\square$

**Lemma D.8.** *Under Condition 4.1, we can show that for all $T_1 \leq t \leq T^*$,*

$$\tilde{y}_i \langle \nabla f(\mathbf{W}^{(t)}), \mathbf{W}^* \rangle \geq \log(2/\epsilon)$$

*Proof of Lemma D.8.* By the gradient decomposition, we can write

$$\tilde{y}_i \langle \nabla f(\mathbf{W}^{(t)}, \mathbf{x}_i), \mathbf{W}^* \rangle$$

$$= \frac{1}{m} \sum_{j,r} \sigma'(\langle \mathbf{w}_{j,r}^{(t)}, y_i \boldsymbol{\mu} \rangle) \langle \boldsymbol{\mu}, j \cdot \mathbf{w}_{j,r}^* \rangle + \frac{1}{m} \sum_{j,r} \sigma'(\langle \mathbf{w}_{j,r}^{(t)}, \boldsymbol{\xi}_i \rangle) \langle y_i \boldsymbol{\xi}_i, j \cdot \mathbf{w}_{j,r}^* \rangle$$

$$\geq \frac{1}{m} \sum_{j = \tilde{y}_i, r} \sigma'(\langle \mathbf{w}_{j,r}^{(t)}, \boldsymbol{\xi}_i \rangle) 5 \log(2/\epsilon) - \frac{1}{m} \sum_{j,r} \sum_{i' \neq i} \sigma'(\langle \mathbf{w}_{j,r}^{(t)}, \boldsymbol{\xi}_i \rangle) 5 \log(2/\epsilon) \widetilde{O}(d^{-1/2})$$

$$- \frac{1}{m} \sum_{j,r} \sum_{i'=1}^{n} \sigma'(\langle \mathbf{w}_{j,r}^{(t)}, y_i \boldsymbol{\mu} \rangle) 5 \log(2/\epsilon) \widetilde{O}(n^{-1} \|\boldsymbol{\mu}\|_2^{-1}) - \frac{1}{m} \sum_{j,r} \sigma'(\langle \mathbf{w}_{j,r}^{(t)}, y_i \boldsymbol{\mu} \rangle) \widetilde{O}(\sigma_0 \|\boldsymbol{\mu}\|_2)$$

$$- \frac{1}{m} \sum_{j,r} \sigma'(\langle \mathbf{w}_{j,r}^{(t)}, \boldsymbol{\xi}_i \rangle) \widetilde{O}\left(\sigma_0 \sigma_\xi \sqrt{d}\right)$$

$$\geq 2 \log(2/\epsilon) - \log(2/\epsilon)$$
$$= \log(2/\epsilon)$$

where in the first inequality, we use the expression of $\mathbf{w}_{j,r}^*$ and Lemma C.1. The second inequality is by

$$\frac{1}{m} \sum_{j=\tilde{y}_i, r} \sigma'(\langle \mathbf{w}_{j,r}^{(t)}, \boldsymbol{\xi}_i \rangle) 5 \log(2/\epsilon) \geq \frac{1}{m} |\mathcal{S}_i^{(t)}| 5 \log(2/\epsilon) \geq 2 \log(2/\epsilon)$$

where we use Lemma D.4 and Lemma D.6 that $|\mathcal{S}_i^{(t)}| \geq 0.4m$. Further the other terms can be bounded by arbitrarily small constant. This completes the proof. $\square$

**Theorem D.2** (Restatement of Theorem 4.2). *Under Condition 4.1, for arbitrary $\epsilon > 0$, there exists $t^* \in [T_1, T^*]$, where $T^* = \widetilde{\Theta}(\eta^{-1}\epsilon^{-1}nm\sigma_\xi^{-2}d^{-1})$, such that*

1. *Training loss converges, i.e., $L_S(\mathbf{W}^{(t^*)}) \leq \epsilon$*

2. *All clean samples, i.e., $i \in \mathcal{S}_t$, it holds that $\tilde{y}_i f(\mathbf{W}^{(t^*)}, \mathbf{x}_i) \geq 0$*

3. *There exists a constant $0 < \tau' \leq \frac{\tau_+ + \tau_-}{2}$ such that there are $\tau'n$ noisy samples, i.e., $i \in \mathcal{S}_f$ satisfy $\tilde{y}_i f(\mathbf{W}^{(t^*)}, \mathbf{x}_i) \geq 0$.*

4. *The test error $L_D^{0-1}(\mathbf{W}^{(t^*)}) \geq 0.5 \min\{\tau_+, \tau_-\}$.*

*Proof of Theorem D.2.* (1) First, we prove that the loss converges to arbitrarily small tolerance. Specifically, we use Lemma D.4 of Kou et al. (2023) to bound for all $t \leq T^*$, we have

$$\|\nabla L_S(\mathbf{W}^{(t)})\|_F^2 = O(\max\{\|\boldsymbol{\mu}\|_2^2, \sigma_\xi^2 d\}) L_S(\mathbf{W}^{(t)}) \tag{10}$$

Then we bound the difference in the distance to optimal solution as

$$\|\mathbf{W}^{(t)} - \mathbf{W}^*\|_F^2 - \|\mathbf{W}^{(t+1)} - \mathbf{W}^*\|_F^2$$
$$= 2\eta \langle \nabla L_S(\mathbf{W}^{(t)}), \mathbf{W}^{(t)} - \mathbf{W}^* \rangle - \eta^2 \|\nabla L_S(\mathbf{W}^{(t)})\|_F^2$$
$$= \frac{2\eta}{n} \sum_{i=1}^{n} \ell_i'^{(t)} \left[ \tilde{y}_i f(\mathbf{W}^{(t)}, \mathbf{x}_i) - \langle \nabla f(\mathbf{W}^{(t)}, \mathbf{x}_i), \mathbf{W}^* \rangle \right] - \eta^2 \|\nabla L_S(\mathbf{W}^{(t)})\|_F^2$$
$$\geq \frac{2\eta}{n} \sum_{i=1}^{n} \ell_i'^{(t)} \left[ \tilde{y}_i f(\mathbf{W}^{(t)}, \mathbf{x}_i) - \log(2/\epsilon) \right] - \eta^2 \|\nabla L_S(\mathbf{W}^{(t)})\|_F^2$$
$$\geq \frac{2\eta}{n} \sum_{i=1}^{n} \left[ \ell\left(f(\mathbf{W}^{(t)}, \mathbf{x}_i), \tilde{y}_i\right) - \epsilon/2 \right] - \eta^2 \|\nabla L_S(\mathbf{W}^{(t)})\|_F^2$$
$$\geq 2\eta L_S(\mathbf{W}^{(t)}) - \eta\epsilon - \eta^2 O(\max\{\|\boldsymbol{\mu}\|_2^2, \sigma_\xi^2 d\}) L_S(\mathbf{W}^{(t)})$$
$$\geq \eta L_S(\mathbf{W}^{(t)}) - \eta\epsilon$$

where the first inequality is due to Lemma D.8 and the second inequality is by convexity of cross entropy function. The third inequality is by (10) and the last inequality is by choosing $\eta \leq C^{-1} \max\{\|\boldsymbol{\mu}\|_2^2, \sigma_\xi^2 d\}^{-1}$ to be sufficiently small.

Finally, we telescope the inequality from $t = T_1$ to $t = T^*$, which yields

$$\frac{1}{T^* - T_1 + 1} \sum_{s=T_1}^{T^*} L_S(\mathbf{W}^{(s)}) \leq \frac{\|\mathbf{W}^{(T_1)} - \mathbf{W}^*\|_F^2}{\eta(T^* - T_1 + 1)} + \epsilon \leq 2\epsilon$$

where the last inequality is due to the choice of $T^* = T_1 + \lfloor \eta^{-1}\epsilon^{-1}\|\mathbf{W}^{(T_1)} - \mathbf{W}^*\|_F^2 \rfloor = T_1 + \widetilde{O}(\eta^{-1}\epsilon^{-1}mnd^{-1}\sigma_\xi^{-2})$.

This suggests there exists an iteration $t^* \in [T_1, T^*]$ where $L_S(\mathbf{W}^{(t^*)}) \leq 2\epsilon$ for any $\epsilon < 0$. By setting $\epsilon \leftarrow 2\epsilon$, we verify the first claim.

(2) For the second claim, it is easy to see by Lemma D.2, for all $i \in \mathcal{S}_t$

$$\tilde{y}_i f(\mathbf{W}^{(t^*)}, \mathbf{x}_i) \geq \frac{1}{m} \sum_{r=1}^{m} \left( \overline{\rho}_{\tilde{y}_i,r,i}^{(t^*)} + \gamma_{\tilde{y}_i,r}^{(t^*)} \right) - 1/C_1 \geq 0$$

where the second inequality is by $\gamma_{\tilde{y}_i,r} \geq 0$ and $\overline{\rho}_{\tilde{y}_i,r,i}^{(t^*)} \geq \overline{\rho}_{\tilde{y}_i,r,i}^{(T_1)} = \Omega(1)$ for $i \in [n], r \in \mathcal{S}_i^{(0)}$.

(3) For the third claim, we prove by contradiction that there exists a sufficiently large gap $C_\epsilon > 0$ such that for noisy samples $i \in \mathcal{S}_f$, there exists a constant fraction that satisfies $\frac{1}{m} \sum_{r=1}^{m} \left( \overline{\rho}_{\tilde{y}_i,r,i}^{(t^*)} - \gamma_{-\tilde{y}_i,r}^{(t^*)} \right) \geq C_\epsilon$.

We prove this claim by contradiction. Suppose the claim does not hold. Then there must exist a constant fraction of samples such that $\frac{1}{m} \sum_{r=1}^{m} \left( \overline{\rho}_{\tilde{y}_i,r,i}^{(t^*)} - \gamma_{-\tilde{y}_i,r}^{(t^*)} \right) \leq C_\epsilon$. Formally, We denote the set of such samples as

$$\mathcal{I}' := \left\{ i \in \mathcal{S}_f : \frac{1}{m} \sum_{r=1}^{m} \left( \overline{\rho}_{\tilde{y}_i,r,i}^{(t^*)} - \gamma_{-\tilde{y}_i,r}^{(t^*)} \right) \leq C_\epsilon \right\}$$

with $|\mathcal{I}'| = \tau' n$ for some constant $\tau' > 0$ that satisfies $\tau' \leq \frac{\tau_+ + \tau_-}{2}$, i.e., upper bounded by the number of noisy samples in the dataset. Then we have

$$L_S(\mathbf{W}^{(t^*)}) = \frac{1}{n} \sum_{i=1}^{n} \ell(f(\mathbf{W}^{(t^*)}, \mathbf{x}_i), \tilde{y}_i) \geq \frac{1}{n} \sum_{i \in \mathcal{I}'} \log(1 + \exp(-\tilde{y}_i f(\mathbf{W}^{(t^*)}, \mathbf{x}_i)))$$

$$\geq \frac{1}{n} \sum_{i \in \mathcal{I}'} \log \left( 1 + \exp \left( \frac{1}{m} \sum_{r=1}^{m} \left( \gamma_{-\tilde{y}_i,r}^{(t^*)} - \overline{\rho}_{\tilde{y}_i,r,i}^{(t^*)} \right) - 1/C_1 \right) \right)$$

$$\geq \tau' \log(1 + \exp(-C_\epsilon - 0.001)) > \tau' \log(2)$$

where we use Lemma D.2 in the second inequality. The third inequality is by the definition of $\mathcal{I}'$ and $C_1 \geq 1000$. Thus this raises a contradiction given that $L_S(\mathbf{W}^{(t^*)}) \leq 2\epsilon$ for any $\epsilon > 0$. This suggests there exists a constant fraction of noisy samples satisfy $\frac{1}{m} \sum_{r=1}^{m} \left( \overline{\rho}_{\tilde{y}_i,r,i}^{(t^*)} - \gamma_{-\tilde{y}_i,r}^{(t^*)} \right) \geq C_\epsilon$. This further indicates by Lemma D.2, for these samples

$$\tilde{y}_i f(\mathbf{W}^{(t^*)}, \mathbf{x}_i) \geq C_\epsilon - 1/C_1 > 0.$$

(4) For the test error, we first derive the probability $\mathbb{P}(yf(\mathbf{W}^{(t^*)}, \mathbf{x}) < 0)$ as

$\mathbb{P}(yf(\mathbf{W}^{(T^*)}, \mathbf{x}) < 0)$

$$= \mathbb{P}\left( \sum_{r=1}^{m} \sigma(\langle \mathbf{w}_{-y,r}^{(t^*)}, \boldsymbol{\xi} + \boldsymbol{\zeta} \rangle) - \sum_{r=1}^{m} \sigma(\langle \mathbf{w}_{y,r}^{(t^*)}, \boldsymbol{\xi} + \boldsymbol{\zeta} \rangle) \geq \sum_{r=1}^{m} \sigma(\langle \mathbf{w}_{y,r}^{(t^*)}, y\boldsymbol{\mu} \rangle) - \sum_{r=1}^{m} \sigma(\langle \mathbf{w}_{-y,r}^{(t^*)}, y\boldsymbol{\mu} \rangle) \right)$$

$$\geq \mathbb{P}\left( \frac{1}{m} \sum_{r=1}^{m} \sigma(\langle \mathbf{w}_{-y,r}^{(t^*)}, \boldsymbol{\xi} + \boldsymbol{\zeta} \rangle) - \frac{1}{m} \sum_{r=1}^{m} \sigma(\langle \mathbf{w}_{y,r}^{(t^*)}, \boldsymbol{\xi} + \boldsymbol{\zeta} \rangle) \geq \frac{1}{m} \sum_{r=1}^{m} \gamma_{y,r}^{(t^*)} + 1/C' \right)$$

$$= \frac{1}{n} \sum_{i=1}^{n} \mathbb{P}\left( \frac{1}{m} \sum_{r=1}^{m} \sigma(\langle \mathbf{w}_{-y,r}^{(t^*)}, \boldsymbol{\xi}_i + \boldsymbol{\zeta} \rangle) - \frac{1}{m} \sum_{r=1}^{m} \sigma(\langle \mathbf{w}_{y,r}^{(t^*)}, \boldsymbol{\xi}_i + \boldsymbol{\zeta} \rangle) \geq \frac{1}{m} \sum_{r=1}^{m} \gamma_{y,r}^{(t^*)} + 1/C' \right) \tag{11}$$

for some sufficiently large constant $C' > 0$ and the second equality is by uniform distribution of $\boldsymbol{\xi}$.

Next, we consider the following two cases separately, i.e., (a) When $y = 1$ and (b) when $y = -1$. When $y = 1$, (11) can be further bounded as

$\mathbb{P}(yf(\mathbf{W}^{(t^*)}, \mathbf{x}) < 0)$

$$\geq 0.5\tau_+ \mathbb{P}\Big(\frac{1}{m}\sum_{r=1}^{m}\sigma(\langle \mathbf{w}_{-1,r}^{(t^*)}, \boldsymbol{\xi}_{i:i\in\mathcal{S}_f\cap\mathcal{S}_1} + \boldsymbol{\zeta}\rangle) - \frac{1}{m}\sum_{r=1}^{m}\sigma(\langle \mathbf{w}_{1,r}^{(t^*)}, \boldsymbol{\xi}_{i:i\in\mathcal{S}_f\cap\mathcal{S}_1} + \boldsymbol{\zeta}\rangle) \geq \frac{1}{m}\sum_{r=1}^{m}\gamma_{1,r}^{(t^*)} + 1/C'\Big)$$

where we use the sample size of $|\mathcal{S}_f \cap \mathcal{S}_1| = \frac{\tau_+ n}{2}$.

Now we analyze the the magnitude of each term. Based on the decomposition, we obtain for any $i \in \mathcal{S}_f$, $j = \pm 1$ and any $r \in [m]$

$$\langle \mathbf{w}_{j,r}^{(t^*)}, \boldsymbol{\xi}_i + \boldsymbol{\zeta} \rangle = \Big\langle \mathbf{w}_{j,r}^{(0)} - \gamma_{j,r}^{(t^*)}\|\boldsymbol{\mu}\|_2^{-2}\boldsymbol{\mu} + \sum_{i'=1}^{n}\overline{\rho}_{j,r,i}^{(t^*)}\|\boldsymbol{\xi}_{i'}\|_2^{-2}\boldsymbol{\xi}_{i'} + \sum_{i'=1}^{n}\underline{\rho}_{j,r,i}^{(t^*)}\|\boldsymbol{\xi}_{i'}\|_2^{-2}\boldsymbol{\xi}_{i'}, \boldsymbol{\xi}_i + \boldsymbol{\zeta}\Big\rangle$$

$$= \overline{\rho}_{j,r,i}^{(t^*)} + \underline{\rho}_{j,r,i}^{(t^*)} + \langle \mathbf{w}_{j,r}^{(0)}, \boldsymbol{\xi}_i + \boldsymbol{\zeta}\rangle - \langle \gamma_{j,r}^{(t^*)}\|\boldsymbol{\mu}\|_2^{-2}\boldsymbol{\mu}, \boldsymbol{\xi}_i + \boldsymbol{\zeta}\rangle$$

$$+ \sum_{i'\neq i}\overline{\rho}_{j,r,i'}^{(t^*)}\frac{\langle \boldsymbol{\xi}_{i'}, \boldsymbol{\xi}_i + \boldsymbol{\zeta}\rangle}{\|\boldsymbol{\xi}_{i'}\|_2^2} + \sum_{i'\neq i}\underline{\rho}_{j,r,i'}^{(t^*)}\frac{\langle \boldsymbol{\xi}_{i'}, \boldsymbol{\xi}_i + \boldsymbol{\zeta}\rangle}{\|\boldsymbol{\xi}_{i'}\|_2^2}$$

Then we can bound particularly for $i \in \mathcal{S}_f \cap \mathcal{S}_1$, i.e., $\tilde{y}_i = -1$

$$\langle \mathbf{w}_{-1,r}^{(t^*)}, \boldsymbol{\xi}_i + \boldsymbol{\zeta}\rangle \geq \overline{\rho}_{-1,r,i}^{(t^*)} - \widetilde{O}(\sigma_0\sigma_\xi\sqrt{d}) - \widetilde{O}(\|\boldsymbol{\mu}\|_2^{-1}\sigma_\xi) - \widetilde{O}(nd^{-1/2})$$

$$\geq \overline{\rho}_{-1,r,i}^{(t^*)} - 1/C_3$$

where we use Lemma C.1, C.2 and the upper bound on $\overline{\rho}_{j,r,i}^{(t^*)}, \gamma_{j,r}^{(t^*)} = \widetilde{O}(1)$ for the first inequality. The second inequality is by Condition 4.1 on $\|\boldsymbol{\mu}\|_2$ and $d$ for some sufficiently large constant $C_3$.

In addition, we can similarly show

$$\langle \mathbf{w}_{1,r}^{(t^*)}, \boldsymbol{\xi}_i + \boldsymbol{\zeta}\rangle \leq \underline{\rho}_{1,r,i}^{(t^*)} + \widetilde{O}(\sigma_0\sigma_\xi\sqrt{d}) + \widetilde{O}(\|\boldsymbol{\mu}\|_2^{-1}\sigma_\xi) + \widetilde{O}(nd^{-1/2}) \leq 1/C_3$$

Then can show for any $i \in \mathcal{S}_f \cap \mathcal{S}_1$, i.e., $\tilde{y}_i = -1$

$$\frac{1}{m}\sum_{r=1}^{m}\sigma(\langle \mathbf{w}_{-1,r}^{(t^*)}, \boldsymbol{\xi}_i + \boldsymbol{\zeta}\rangle) - \frac{1}{m}\sum_{r=1}^{m}\sigma(\langle \mathbf{w}_{1,r}^{(t^*)}, \boldsymbol{\xi}_i + \boldsymbol{\zeta}\rangle) \geq \frac{1}{m}\sum_{r=1}^{m}\overline{\rho}_{-1,r,i}^{(t^*)} - 2/C_3$$

$$\geq \frac{1}{m}\sum_{r=1}^{m}\gamma_{1,r}^{(t^*)} + C_\epsilon - 2/C_3$$

$$> \frac{1}{m}\sum_{r=1}^{m}\gamma_{1,r}^{(t^*)} + 1/C'$$

where we choose $C_3, C'$ such that $C_\epsilon - 2/C_3 > 1/C'$. This suggests when $y = 1$, we have

$$\mathbb{P}(yf(\mathbf{W}^{(T^*)}, \mathbf{x}) < 0) \geq 0.5\tau_+$$

Similarly, we use the same argument to show when $y = -1$,

$$\mathbb{P}(yf(\mathbf{W}^{(T^*)}, \mathbf{x}) < 0) \geq 0.5\tau_-.$$

This completes the proof that $\mathbb{P}(yf(\mathbf{W}^{(T^*)}, \mathbf{x}) < 0) \geq 0.5\min\{\tau_-, \tau_+\}$. □

# E   ANALYSIS WITHOUT LABEL NOISE

For the case of no label noise, i.e., $\tau_+, \tau_- = 0$. We still require the same assumption as in Condition 4.1. We reiterate the assumption for completeness here.

**Condition E.1.** We let $T^* = \widetilde{\Theta}(\eta^{-1}\epsilon^{-1}nm\sigma_\xi^{-2}d^{-1})$ to be the maximum number of iterations considered. Suppose that there exists a sufficiently large constant $C$ such that the following hold:

1. The signal-to-noise ratio is bounded by constants $n \cdot \text{SNR}^2 = \Theta(1)$.

2. The dimension $d$ is sufficiently large, $d \geq C \max \left\{ n^2 \log(nm/\delta) \log(T^*)^2, n\|\boldsymbol{\mu}\|_2 \sigma_\xi^{-1} \sqrt{\log(n/\delta)} \right\}$.

3. The standard deviation of the Gaussian initialization $\sigma_0$ is chosen such that $\sigma_0 \leq C^{-1} \min \left\{ \sqrt{n} \sigma_\xi^{-1} d^{-1}, \|\boldsymbol{\mu}\|_2^{-1} \log(m/\delta)^{-1/2} \right\}$.

4. The size of training sample $n$ and width $m$ adhere to $m \geq C \log(n/\delta), n \geq C \log(m/\delta)$.

5. The signal strength satisfies $\|\boldsymbol{\mu}\|_2^2 \geq C \sigma_\xi^2 \log(n/\delta)$.

6. The learning rate $\eta$ satisfies $\eta \leq C^{-1} \min \left\{ \sigma_\xi^{-2} d^{-3/2} n^2 m \sqrt{\log(n/\delta)}, \sigma_\xi^{-2} d^{-1} n \right\}$.

With the label noise, the coefficient update equations are given by

$$\gamma_{j,r}^{(0)}, \overline{\rho}_{j,r,i}^{(0)}, \underline{\rho}_{j,r,i}^{(0)} = 0,$$

$$\gamma_{j,r}^{(t+1)} = \gamma_{j,r}^{(t)} - \frac{\eta}{nm} \sum_{i=1}^n \ell_i'^{(t)} \sigma'(\langle \mathbf{w}_{j,r}^{(t)}, y_i \boldsymbol{\mu} \rangle) \|\boldsymbol{\mu}\|_2^2,$$

$$\overline{\rho}_{j,r,i}^{(t+1)} = \overline{\rho}_{j,r,i}^{(t)} - \frac{\eta}{nm} \ell_i'^{(t)} \sigma'(\langle \mathbf{w}_{j,r}^{(t)}, \boldsymbol{\xi}_i \rangle) \|\boldsymbol{\xi}_i\|_2^2 \mathbb{1}(y_i = j),$$

$$\underline{\rho}_{j,r,i}^{(t+1)} = \underline{\rho}_{j,r,i}^{(t)} + \frac{\eta}{nm} \ell_i'^{(t)} \sigma'(\langle \mathbf{w}_{j,r}^{(t)}, \boldsymbol{\xi}_i \rangle) \|\boldsymbol{\xi}_i\|_2^2 \mathbb{1}(y_i = -j).$$

where we highlight that for all $i \in [n]$, $\tilde{y}_i = y_i$.

**Proposition E.1.** *Under Assumption 4.1 and the same definition as for the label noise case, for $0 \leq t \leq T^*$, we have*

$$0 \leq \overline{\rho}_{j,r,i}^{(t)} \leq \alpha, \tag{12}$$

$$0 \geq \underline{\rho}_{j,r,i}^{(t)} \geq -\beta - 10\sqrt{\frac{\log(6n^2/\delta)}{d}} n\alpha \geq -\alpha, \tag{13}$$

$$0 \leq \gamma_{j,r}^{(t)} \leq C_\gamma \alpha \tag{14}$$

In order to prove such results, we use the same induction strategy as for the label noise case. We first notice that if (12), (13), (14) hold at iteration $t$, then bounds in Lemma D.1 and Lemma D.2 hold at iteration $t$. We include the results here for the purpose of completeness.

**Lemma E.1.** *Under Condition E.1, suppose* (12), (13), (14) *hold at iteration $t$,*

$$|\langle \mathbf{w}_{j,r}^{(t)} - \mathbf{w}_{j,r}^{(0)}, \boldsymbol{\mu} \rangle - j \cdot \gamma_{j,r}^{(t)}| \leq \mathrm{SNR} \sqrt{\frac{8 \log(6n/\delta)}{d}} n\alpha,$$

$$|\langle \mathbf{w}_{j,r}^{(t)} - \mathbf{w}_{j,r}^{(0)}, \boldsymbol{\xi}_i \rangle - \overline{\rho}_{j,r,i}^{(t)}| \leq 5\sqrt{\frac{\log(6n^2/\delta)}{d}} n\alpha, \quad y_i = j$$

$$|\langle \mathbf{w}_{j,r}^{(t)} - \mathbf{w}_{j,r}^{(0)}, \boldsymbol{\xi}_i \rangle - \underline{\rho}_{j,r,i}^{(t)}| \leq 5\sqrt{\frac{\log(6n^2/\delta)}{d}} n\alpha, \quad y_i = -j$$

*for all $r \in [m], j = \pm 1, i \in [n]$. Further, there exists a sufficiently large constant $C_1$ such that*

$$\frac{1}{m} \sum_{r=1}^m \left( \gamma_{y_i,r}^{(t)} + \overline{\rho}_{y_i,r,i}^{(t)} \right) - 1/C_1 \leq y_i f(\mathbf{W}^{(t)}, \mathbf{x}_i) \leq \frac{1}{m} \sum_{r=1}^m \left( \gamma_{y_i,r}^{(t)} + \overline{\rho}_{y_i,r,i}^{(t)} \right) + 1/C_1$$

*for all $i \in [n]$.*

*Proof of Lemma E.1.* The proof follows directly from Lemma D.1 and Lemma D.2. □

Next we prove a stronger lemma that only holds under the condition $n \cdot \mathrm{SNR}^2 = \Theta(1)$ and without the presence of label noise. Nevertheless, we remark that the following result holds under small SNR, i.e., $n \cdot \mathrm{SNR}^2 = o(1)$ and has been proved in Kou et al. (2023).

First we require an lemma that allows to bound the loss derivative ratios.

**Lemma E.2** (Kou et al. (2023)). *Let $g(z) = -1/(1 + \exp(z))$, then for all $z_2 - c \geq z_1 \geq -1$, for $c \geq 0$, we have*

$$\frac{\exp(c)}{4} \leq \frac{g(z_1)}{g(z_2)} \leq \exp(c)$$

**Lemma E.3.** *Under Condition E.1, and for any given $t \leq T^*$, suppose (12), (13), (14) hold for all iterations $s \leq t$. Then we can prove for some constant $\kappa \geq 0$*

(1) $\frac{1}{m}\sum_{r=1}^{m}(\overline{\rho}_{y_i,r,i}^{(s)} + \gamma_{y_i,r}^{(s)} - \overline{\rho}_{y_k,r,k}^{(s)} - \gamma_{y_k,r}^{(s)}) \leq \kappa$ *for all $i, k \in [n]$.*

(2) $\ell_i'^{(s)}/\ell_k'^{(s)} \leq \tilde{C}_\ell$ *for all $i, k \in [n]$.*

(3) $\mathcal{S}_i^{(0)} \subseteq \mathcal{S}_i^{(s)}, \mathcal{S}_{j,r}^{(0)} \subseteq \mathcal{S}_{j,r}^{(s)}$, *for all $i \in [n]$ and $j = \pm 1, r \in [m]$.*

*Proof of Lemma E.3.* We prove the results by induction. It is clear at $s = 0$, claim (1), (3) are satisfied trivially. Then for claim (2), we use Lemma E.2 to bound

$$\frac{\ell_i'^{(0)}}{\ell_k'^{(0)}} \leq \exp(y_k f(\mathbf{W}^{(0)}, \mathbf{x}_k) - y_i f(\mathbf{W}^{(0)}, \mathbf{x}_i))$$

$$\leq \exp\left(\frac{1}{m}\sum_{r=1}^{m}\left(\overline{\rho}_{y_k,r,k}^{(0)} + \gamma_{y_k,r}^{(0)}\right) - \frac{1}{m}\sum_{r=1}^{m}\left(\overline{\rho}_{y_i,r,i}^{(0)} + \gamma_{y_i,r}^{(0)}\right) + 2/C_1\right)$$

$$= \exp(2/C_1),$$

which shows a constant upper bound.

Next suppose at $t = \tilde{t}$, (1)-(3) hold for all $s \leq \tilde{t} - 1$, then we show they also hold at $\tilde{t}$. For (1), according to the update rule of the coefficients,

$$\frac{1}{m}\sum_{r=1}^{m}\left(\overline{\rho}_{y_i,r,i}^{(\tilde{t})} - \overline{\rho}_{y_k,r,k}^{(\tilde{t})}\right) = \frac{1}{m}\sum_{r=1}^{m}\left(\overline{\rho}_{y_i,r,i}^{(\tilde{t}-1)} - \overline{\rho}_{y_k,r,k}^{(\tilde{t}-1)}\right)$$

$$- \frac{\eta}{nm^2}\left(\sum_{r\in\mathcal{S}_i^{(\tilde{t}-1)}} \ell_i'^{(\tilde{t}-1)}\|\boldsymbol{\xi}_i\|_2^2 - \sum_{r\in\mathcal{S}_k^{(\tilde{t}-1)}} \ell_k'^{(\tilde{t}-1)}\|\boldsymbol{\xi}_k\|_2^2\right) \quad (15)$$

$$\frac{1}{m}\sum_{r=1}^{m}\left(\gamma_{y_i,r}^{(\tilde{t})} - \gamma_{y_k,r}^{(\tilde{t})}\right) = \frac{1}{m}\sum_{r=1}^{m}\left(\gamma_{y_i,r}^{(\tilde{t}-1)} - \gamma_{y_k,r}^{(\tilde{t}-1)}\right)$$

$$- \frac{\eta\|\boldsymbol{\mu}\|_2^2}{nm^2}\sum_{r=1}^{m}\underbrace{\left(\sum_{i'=1}^{n} \ell_{i'}'^{(\tilde{t}-1)}\mathbb{1}(\langle\mathbf{w}_{y_i,r}^{(\tilde{t}-1)}, y_{i'}\boldsymbol{\mu}\rangle) - \sum_{i'=1}^{n} \ell_{i'}'^{(\tilde{t}-1)}\mathbb{1}(\langle\mathbf{w}_{y_k,r}^{(\tilde{t}-1)}, y_{i'}\boldsymbol{\mu}\rangle)\right)}_{A_5}$$

$$(16)$$

We first analyze $A_5$ depending on the following four cases.

- When $\langle\mathbf{w}_{y_i,r}^{(\tilde{t}-1)}, \boldsymbol{\mu}\rangle \geq 0, \langle\mathbf{w}_{y_k,r}^{(\tilde{t}-1)}, \boldsymbol{\mu}\rangle \geq 0$, we have $A_5 = \sum_{i'\in\mathcal{S}_1} \ell_{i'}'^{(t-1)} - \sum_{i'\in\mathcal{S}_1} \ell_{i'}'^{(t-1)} = 0$.

- When $\langle\mathbf{w}_{y_i,r}^{(\tilde{t}-1)}, \boldsymbol{\mu}\rangle \leq 0, \langle\mathbf{w}_{y_k,r}^{(\tilde{t}-1)}, \boldsymbol{\mu}\rangle \leq 0$, we have $A_5 = \sum_{i'\in\mathcal{S}_{-1}} \ell_{i'}'^{(t-1)} - \sum_{i'\in\mathcal{S}_{-1}} \ell_{i'}'^{(t-1)} = 0$.

- When $\langle\mathbf{w}_{y_i,r}^{(\tilde{t}-1)}, \boldsymbol{\mu}\rangle \geq 0, \langle\mathbf{w}_{y_k,r}^{(\tilde{t}-1)}, \boldsymbol{\mu}\rangle \leq 0$, we have $A_5 = \sum_{i'\in\mathcal{S}_1} \ell_{i'}'^{(t-1)} - \sum_{i'\in\mathcal{S}_{-1}} \ell_{i'}'^{(t-1)}$.

- When $\langle\mathbf{w}_{y_i,r}^{(\tilde{t}-1)}, \boldsymbol{\mu}\rangle \leq 0, \langle\mathbf{w}_{y_k,r}^{(\tilde{t}-1)}, \boldsymbol{\mu}\rangle \geq 0$, we have $A_5 = \sum_{i'\in\mathcal{S}_{-1}} \ell_{i'}'^{(t-1)} - \sum_{i'\in\mathcal{S}_1} \ell_{i'}'^{(t-1)}$.

Now we would like to bound the combination of (15) and (16).

When $\frac{1}{m}\sum_{r=1}^m \big(\overline{\rho}_{y_i,r,i}^{(\tilde{t}-1)} + \gamma_{y_i,r}^{(\tilde{t}-1)} - \overline{\rho}_{y_k,r,k}^{(\tilde{t}-1)} - \gamma_{y_k,r}^{(\tilde{t}-1)}\big) \le 0.5\kappa$, then (15) can be bounded as

$$\frac{1}{m}\sum_{r=1}^m \big(\overline{\rho}_{y_i,r,i}^{(\tilde{t})} + \gamma_{y_i,r}^{(\tilde{t})} - \overline{\rho}_{y_k,r,k}^{(\tilde{t})} - \gamma_{y_k,r}^{(\tilde{t})}\big)$$

$$\le \frac{1}{m}\sum_{r=1}^m \big(\overline{\rho}_{y_i,r,i}^{(\tilde{t}-1)} + \gamma_{y_i,r}^{(\tilde{t}-1)} - \overline{\rho}_{y_k,r,k}^{(\tilde{t}-1)} - \gamma_{y_k,r}^{(\tilde{t}-1)}\big) - \frac{\eta}{nm^2}\left(\sum_{r\in\mathcal{S}_i^{(\tilde{t}-1)}} \ell_i'^{(\tilde{t}-1)}\|\boldsymbol{\xi}_i\|_2^2 - \sum_{r\in\mathcal{S}_k^{(\tilde{t}-1)}} \ell_k'^{(\tilde{t}-1)}\|\boldsymbol{\xi}_k\|_2^2\right)$$

$$- \frac{\eta\|\boldsymbol{\mu}\|_2^2}{nm^2}\sum_{r=1}^m\left(\sum_{i'=1}^n \ell_{i'}'^{(t-1)}\mathbb{1}(\langle \mathbf{w}_{y_i,r}^{(\tilde{t}-1)}, y_{i'}\boldsymbol{\mu}\rangle) - \sum_{i'=1}^n \ell_{i'}'^{(t-1)}\mathbb{1}(\langle \mathbf{w}_{y_k,r}^{(\tilde{t}-1)}, y_{i'}\boldsymbol{\mu}\rangle)\right)$$

$$\le 0.5\kappa - \frac{\eta}{nm}|\mathcal{S}_i^{(\tilde{t}-1)}|\ell_i'^{(\tilde{t}-1)}\|\boldsymbol{\xi}_i\|_2^2 - \frac{\eta}{nm}\sum_{i'=1}^n \ell_i'^{(\tilde{t})}\|\boldsymbol{\mu}\|_2^2$$

$$\le 0.5\kappa + 1.01\frac{\eta\sigma_\xi^2 d}{n} + \frac{\eta\|\boldsymbol{\mu}\|_2^2}{m}$$

$$\le \kappa$$

where the second inequality is by $\ell_i'^{(t)} \le 0$ for all $i, t$. The third inequality is by $|\mathcal{S}_i^{(\tilde{t}-1)}| \le m$, $|\ell_i'^{(\tilde{t}-1)}| \le 1$ and Lemma D.5. The last inequality us by Condition E.1 for sufficiently small stepsize $\eta$.

When $\frac{1}{m}\sum_{r=1}^m \big(\overline{\rho}_{y_i,r,i}^{(\tilde{t}-1)} + \gamma_{y_i,r}^{(\tilde{t}-1)} - \overline{\rho}_{y_k,r,k}^{(\tilde{t}-1)} - \gamma_{y_k,r}^{(\tilde{t}-1)}\big) \ge 0.5\kappa$, then by Lemma E.1,

$$y_i f(\mathbf{W}^{(\tilde{t}-1)}, \mathbf{x}_i) - y_k f(\mathbf{W}^{(\tilde{t}-1)}, \mathbf{x}_k) \ge \frac{1}{m}\sum_{r=1}^m \big(\gamma_{y_i,r}^{(\tilde{t}-1)} + \overline{\rho}_{y_i,r,i}^{(\tilde{t}-1)} - \gamma_{y_k,r}^{(\tilde{t}-1)} - \overline{\rho}_{y_k,r,k}^{(\tilde{t}-1)}\big) - 2/C_1$$

$$\ge 0.5\kappa - 2/C_1$$

$$\ge 0.4\kappa$$

where we choose $C_1 \ge 20/\kappa$. Then by Lemma E.2

$$\frac{\ell_i'^{(\tilde{t}-1)}}{\ell_k'^{(\tilde{t}-1)}} \le \exp(y_k f(\mathbf{W}^{(\tilde{t}-1)}, \mathbf{x}_k) - y_i f(\mathbf{W}^{(\tilde{t}-1)}, \mathbf{x}_i)) \le \exp(-0.4\kappa).$$

Then we can show

$$\frac{|\mathcal{S}_i^{(\tilde{t}-1)}| \cdot |\ell_i'^{(\tilde{t}-1)}| \cdot \|\boldsymbol{\xi}_i\|_2^2}{|\mathcal{S}_k^{(\tilde{t}-1)}| \cdot |\ell_k'^{(\tilde{t}-1)}| \cdot \|\boldsymbol{\xi}_k\|_2^2} \le 1.01 \cdot \exp(-0.4\kappa) \tag{17}$$

where we use Lemma D.5 and D.6 by choosing sufficiently large $d$ and $m$.

Then we obtain

$$\frac{1}{m}\sum_{r=1}^m \big(\overline{\rho}_{y_i,r,i}^{(\tilde{t})} + \gamma_{y_i,r}^{(\tilde{t})} - \overline{\rho}_{y_k,r,k}^{(\tilde{t})} - \gamma_{y_k,r}^{(\tilde{t})}\big)$$

$$\le \frac{1}{m}\sum_{r=1}^m \big(\overline{\rho}_{y_i,r,i}^{(\tilde{t}-1)} + \gamma_{y_i,r}^{(\tilde{t}-1)} - \overline{\rho}_{y_k,r,k}^{(\tilde{t}-1)} - \gamma_{y_k,r}^{(\tilde{t}-1)}\big) - \frac{\eta}{nm^2}\big(|\mathcal{S}_i^{(\tilde{t}-1)}|\ell_i'^{(\tilde{t}-1)}\|\boldsymbol{\xi}_i\|_2^2 - |\mathcal{S}_k^{(\tilde{t}-1)}|\ell_k'^{(\tilde{t}-1)}\|\boldsymbol{\xi}_k\|_2^2\big)$$

$$- \frac{\eta}{nm}\Big(\sum_{i'\in\mathcal{S}_{\pm 1}} \ell_{i'}'^{(\tilde{t}-1)} - \sum_{i'\in\mathcal{S}_{\mp 1}} \ell_{i'}'^{(\tilde{t}-1)}\Big)\|\boldsymbol{\mu}\|_2^2$$

$$\le \frac{1}{m}\sum_{r=1}^m \big(\overline{\rho}_{y_i,r,i}^{(\tilde{t}-1)} + \gamma_{y_i,r}^{(\tilde{t}-1)} - \overline{\rho}_{y_k,r,k}^{(\tilde{t}-1)} - \gamma_{y_k,r}^{(\tilde{t}-1)}\big) + \frac{\eta}{nm^2}\big(1.01\exp(-0.4\kappa) - 1\big)|\mathcal{S}_k^{(\tilde{t}-1)}||\ell_k'^{(\tilde{t}-1)}| \cdot \|\boldsymbol{\xi}_i\|_2^2$$

$$+ \frac{\eta}{2m}(\tilde{C}_\ell - 1)\min_{i\in[n]}|\ell_i'^{(\tilde{t}-1)}|\|\boldsymbol{\mu}\|_2^2$$

$$\leq \frac{1}{m} \sum_{r=1}^{m} \left(\overline{\rho}_{y_i,r,i}^{(\tilde{t}-1)} + \gamma_{y_i,r}^{(\tilde{t}-1)} - \overline{\rho}_{y_k,r,k}^{(\tilde{t}-1)} - \gamma_{y_k,r}^{(\tilde{t}-1)}\right) + \frac{\eta}{nm}\left((0.49\exp(-0.4\kappa)-0.48)|\ell_k'^{(\tilde{t}-1)}|\sigma_\xi^2 d\right.$$

$$\left. + 0.5n(\tilde{C}_\ell - 1)\tilde{C}_\ell|\ell_k'^{(\tilde{t}-1)}|\|\boldsymbol{\mu}\|_2^2\right)$$

$$\leq \frac{1}{m} \sum_{r=1}^{m} \left(\overline{\rho}_{y_i,r,i}^{(\tilde{t}-1)} + \gamma_{y_i,r}^{(\tilde{t}-1)} - \overline{\rho}_{y_k,r,k}^{(\tilde{t}-1)} - \gamma_{y_k,r}^{(\tilde{t}-1)}\right)$$

$$\leq \kappa$$

where the second inequality is by applying (17) and also $\sum_{i'\in\mathcal{S}_{\pm 1}} \ell_{i'}'^{(\tilde{t}-1)} - \sum_{i'\in\mathcal{S}_{\mp 1}} \ell_{i'}'^{(\tilde{t}-1)} \leq \frac{n}{2}(\max_{i\in[n]}|\ell_i'^{(\tilde{t}-1)}| - \min_{i\in[n]}|\ell_i'^{(\tilde{t}-1)}|) \leq \frac{n}{2}(\tilde{C}_\ell - 1)\min_{i\in[n]}|\ell_i'^{(\tilde{t}-1)}|)$ by induction. The third inequality is by $\kappa \geq 1$ and Lemma D.5, Lemma D.6. The fourth inequality follows from the conditions on $n \cdot \text{SNR}^2 \leq \frac{2(0.48-0.49\exp(-0.4\kappa))}{(\tilde{C}_\ell-1)\tilde{C}_\ell} = O(1)$. This verifies the claim (1) for $t = \tilde{t}$.

Now by Lemma E.2 and Lemma E.1, we can show

$$\frac{\ell_i'^{(\tilde{t})}}{\ell_k'^{(\tilde{t})}} \leq \exp(y_k f(\mathbf{W}^{(\tilde{t})}, \mathbf{x}_k) - y_i f(\mathbf{W}^{(\tilde{t})}, \mathbf{x}_i))$$

$$\leq \exp\left(\frac{1}{m}\sum_{r=1}^{m}\left(\overline{\rho}_{y_k,r,i}^{(\tilde{t})} + \gamma_{y_k,r}^{(\tilde{t})} - \overline{\rho}_{y_i,r,k}^{(\tilde{t})} - \gamma_{y_i,r}^{(\tilde{t})}\right) + 2/C_1\right)$$

$$\leq \exp(\kappa + 2/C_1)$$

Hence for a given $\kappa$, we can take $\tilde{C}_\ell = \exp(\kappa + 2/C_1)$. This verifies that claim (2) is satisfied.

To verify the claim (3), we can show

$$\langle \mathbf{w}_{\tilde{y}_i,r}^{(\tilde{t})}, \boldsymbol{\xi}_i \rangle = \langle \mathbf{w}_{\tilde{y}_i,r}^{(\tilde{t}-1)}, \boldsymbol{\xi}_i \rangle - \frac{\eta}{nm}\sum_{i'=1}^{n}\ell_{i'}'^{(\tilde{t}-1)} \cdot \sigma'(\langle \mathbf{w}_{\tilde{y}_i,r}^{(\tilde{t}-1)}, \boldsymbol{\xi}_{i'}\rangle) \cdot \langle \boldsymbol{\xi}_i, \boldsymbol{\xi}_{i'}\rangle$$

$$- \frac{\eta}{nm}\sum_{i'=1}^{n}\ell_{i'}'^{(\tilde{t}-1)} \cdot \sigma'(\langle \mathbf{w}_{\tilde{y}_i,r}^{(\tilde{t}-1)}, y_{i'}\boldsymbol{\mu}\rangle) \cdot \langle y_{i'}\boldsymbol{\mu}, \boldsymbol{\xi}_i\rangle$$

$$= \langle \mathbf{w}_{\tilde{y}_i,r}^{(\tilde{t}-1)}, \boldsymbol{\xi}_i \rangle \underbrace{- \frac{\eta}{nm}\ell_i'^{(\tilde{t}-1)}\|\boldsymbol{\xi}_i\|_2^2}_{A_6} \underbrace{- \frac{\eta}{nm}\sum_{i'\neq i}\ell_{i'}'^{(\tilde{t}-1)}\sigma'(\langle \mathbf{w}_{\tilde{y}_i,r}^{(\tilde{t}-1)}, \boldsymbol{\xi}_{i'}\rangle) \cdot \langle \boldsymbol{\xi}_i, \boldsymbol{\xi}_{i'}\rangle}_{A_7}$$

$$\underbrace{- \frac{\eta}{nm}\sum_{i'=1}^{n}\ell_{i'}'^{(\tilde{t}-1)} \cdot \sigma'(\langle \mathbf{w}_{\tilde{y}_i,r}^{(\tilde{t}-1)}, y_{i'}\boldsymbol{\mu}\rangle) \cdot \langle y_{i'}\boldsymbol{\mu}, \boldsymbol{\xi}_i\rangle}_{A_8}.$$

We respectively bound the three terms as

$$A_6 \geq 0.99\frac{\sigma_\xi^2 d\eta}{nm}|\ell_i'^{(\tilde{t}-1)}|,$$

where we use D.5. For $A_7$, we can bound

$$|A_7| \leq 2n\tilde{C}_\ell|\ell_i'^{(\tilde{t}-1)}|\sigma_\xi^2\sqrt{d\log(6n^2/\delta)},$$

where we use Lemma C.1 and claim (2). Similarly,

$$|A_8| \leq n\tilde{C}_\ell|\ell_i'^{(\tilde{t}-1)}|\|\boldsymbol{\mu}\|_2\sigma_\xi\sqrt{2\log(6n/\delta)}$$

where we use Lemma C.1 and claim (2). By the Condition E.1 where $d \geq C\max\{\tilde{C}_\ell^2 n^2\log(6n^2/\delta), \tilde{C}_\ell n\|\boldsymbol{\mu}\|_2\sigma_\xi^{-1}\sqrt{2\log(6n/\delta)}\}$ for sufficiently large $C$. This ensures $A_6 \geq |A_7| + |A_8|$, which leads to $\langle \mathbf{w}_{\tilde{y}_i,r}^{(\tilde{t})}, \boldsymbol{\xi}_i\rangle \geq \langle \mathbf{w}_{\tilde{y}_i,r}^{(\tilde{t}-1)}, \boldsymbol{\xi}_i\rangle > 0$ and thus $\mathcal{S}_i^{(0)} \subseteq \mathcal{S}_i^{(\tilde{t}-1)} \subseteq \mathcal{S}_i^{(\tilde{t})}$. Similarly, we can use the same argument to prove $\mathcal{S}_{j,r}^{(0)} \subseteq \mathcal{S}_{j,r}^{(\tilde{t}-1)} \subseteq \mathcal{S}_{j,r}^{(\tilde{t})}$. $\qquad\square$

*Proof of Proposition E.1.* We prove the results by induction. At $t = 0$, it is clear that the claims hold trivially. Suppose there exists $\widetilde{T} \leq T^*$ such that the the claims hold for all $t \leq \widetilde{T} - 1$. We aim to show they also hold at $t = \widetilde{T}$. By Lemma E.3, we know for all $t \leq \widetilde{T} - 1$, we have $\ell_i'^{(t)}/\ell_k'^{(t)} \leq \tilde{C}_\ell$ for all $i, k \in [n]$ and $\mathcal{S}_i^{(0)} \subseteq \mathcal{S}_i^{(t)}, \mathcal{S}_{j,r}^{(0)} \subseteq \mathcal{S}_{j,r}^{(t)}$.

First, we follow the same proof strategy to show $0 \geq \rho_{-j,r,i}^{(\widetilde{T})} \geq -\beta - 10\sqrt{\log(6n^2/\delta)/d}$.

Next we show the upper bound for $\overline{\rho}_{j,r,i}^{(\widetilde{T})}$. Let $t_{r,i}$ be the last time $t < T^*$ such that $\overline{\rho}_{j,r,i}^{(t)} \leq 0.5\alpha$. Then

$$
\overline{\rho}_{y_i,r,i}^{(\widetilde{T})} = \overline{\rho}_{y_i,r,i}^{(t_{r,i})} - \frac{\eta}{nm}\ell_i'^{(t_{r,i})}\mathbb{1}(\langle \mathbf{w}_{y_i,r}^{(t_{r,i})}, \boldsymbol{\xi}_i \rangle \geq 0)\|\boldsymbol{\xi}_i\|_2^2 - \sum_{t \in (t_{r,i}, \widetilde{T})} \frac{\eta}{nm}\ell_i'^{(t)}\mathbb{1}(\langle \mathbf{w}_{y_i,r}^{(t)}, \boldsymbol{\xi}_i \rangle \geq 0)\|\boldsymbol{\xi}_i\|_2^2
$$

$$
\leq 0.5\alpha + 1.01\frac{\eta\sigma_\xi^2 d}{nm} + 1.01\frac{\eta\sigma_\xi^2 d}{nm}\sum_{t \in (t_{r,i}, \widetilde{T})} \frac{1}{1 + \exp(\frac{1}{m}\sum_{r=1}^m (\overline{\rho}_{y_i,r,i}^{(t)} + \gamma_{y_i,r}^{(t)}) - 1/C_1)}
$$

$$
\leq 0.75\alpha + 2.02\frac{\eta\sigma_\xi^2 d}{nm}
$$

$$
\leq \alpha
$$

where the first inequality is by Lemma D.5, Lemma E.1 and $|\ell_i'^{(t_{r,i})}| \leq 1$. The second and third inequality is by choosing $\eta \leq C^{-1}nm\sigma_\xi^{-2}d^{-1}$ for sufficiently large $C$.

Then we proceed to show upper bound for $\gamma_{j,r}^{(\widetilde{T})}$. From the update rule, it is clear that $\gamma_{j,r}^{(\widetilde{T})} \geq \gamma_{j,r}^{(\widetilde{T}-1)} \geq 0$. To prove the upper bound on $\gamma_{j,r}^{(\widetilde{T})}$, we aim to show there exists $i^* \in [n]$ such that for all $t \leq T^*$ that

$$
\frac{\gamma_{j,r}^{(t)}}{\overline{\rho}_{y_{i^*},r,i^*}^{(t)}} \leq C_\gamma n \cdot \text{SNR}^2.
$$

where we take $C_\gamma = 1.1\tilde{C}_\ell$. We prove the claim by induction. We first lower bound $\overline{\rho}_{y_i,r,i}^{(\widetilde{T})}$. In particular, we can lower bound for any $i^* \in [n], r \in \mathcal{S}_i^{(0)}$ that

$$
\overline{\rho}_{y_{i^*},r,i^*}^{(\widetilde{T})} = \overline{\rho}_{y_{i^*},r,i^*}^{(\widetilde{T}-1)} - \frac{\eta}{nm}\ell_{i^*}'^{(\widetilde{T}-1)}\|\boldsymbol{\xi}_{i^*}\|_2^2 \geq \overline{\rho}_{y_{i^*},r,i^*}^{(\widetilde{T}-1)} + 0.49\frac{\eta\sigma_\xi^2 d}{nm}|\ell_{i^*}'^{(\widetilde{T}-1)}|,
$$

where we use Lemma D.5 and Lemma E.1. Then for $\gamma_{j,r}^{(\widetilde{T})}$, we have

$$
\gamma_{j,r}^{(\widetilde{T})} = \gamma_{j,r}^{(\widetilde{T}-1)} - \frac{\eta}{nm}\sum_{i=1}^n \ell_i'^{(\widetilde{T}-1)}\sigma'(\langle \mathbf{w}_{j,r}^{(\widetilde{T}-1)}, y_i\boldsymbol{\mu}\rangle)\|\boldsymbol{\mu}\|_2^2 \leq \gamma_{j,r}^{(\widetilde{T}-1)} + \frac{\eta\|\boldsymbol{\mu}\|_2^2\tilde{C}_\ell|\ell_{i^*}'^{(\widetilde{T}-1)}|}{m}
$$

where we use the second claim of Lemma E.3.

Then at iteration $t = 1$, we can show

$$
\frac{\gamma_{j,r}^{(1)}}{\overline{\rho}_{y_{i^*},r,i^*}^{(1)}} \leq \frac{n\|\boldsymbol{\mu}\|_2^2\tilde{C}_\ell}{0.49\sigma_\xi^2 d} \leq 1.1\tilde{C}_\ell n \cdot \text{SNR}^2 = C_\gamma n \cdot \text{SNR}^2.
$$

Now suppose for all $t \leq \widetilde{T} - 1$, we have $\gamma_{j,r}^{(t)}/\overline{\rho}_{y_{i^*},r,i^*}^{(t)} \leq C_\gamma n \cdot \text{SNR}^2$, then we

Then we can bound

$$
\frac{\gamma_{j,r}^{(\widetilde{T})}}{\overline{\rho}_{y_{i^*},r,i^*}^{(\widetilde{T})}} \leq \max\left\{\frac{\gamma_{j,r}^{(\widetilde{T}-1)}}{\overline{\rho}_{y_{i^*},r,i^*}^{(\widetilde{T}-1)}}, \frac{n\|\boldsymbol{\mu}\|_2^2\tilde{C}_\ell}{0.49\sigma_\xi^2 d}\right\} \leq C_\gamma n \cdot \text{SNR}^2.
$$

This shows $\gamma_{j,r}^{(\widetilde{T})} \leq C_\gamma n \cdot \text{SNR}^2\overline{\rho}_{y_{i^*},r,i^*}^{(\widetilde{T}-1)} \leq C_\gamma\alpha$. $\qquad\square$

### E.1 FIRST STAGE

In the first stage, we can lower bound the loss derivatives by a constant $C_\ell$ (by Lemma D.3) and we show both $\rho_{y_i,r,i}^{(t)}$, $r \in \mathcal{S}_i^{(0)}$ and $\gamma_{j,r}^{(t)}$ can grow to a constant order.

**Theorem E.1.** *Under Condition E.1, there exists $T_1 = \Theta(\eta^{-1}nm\sigma_\xi^{-2}d^{-1})$ such that*

- $\overline{\rho}_{y_i,r,i}^{(T_1)} = \Theta(1)$ *for all $i \in [n]$ and $r \in \mathcal{S}_i^{(0)}$.*

- $\gamma_{j,r}^{(T_1)} = \Theta(n \cdot \mathrm{SNR}^2) = \Theta(1)$ *for all $j = \pm 1, r \in [m]$.*

- $y_i f(\mathbf{W}^{(T_1)}, \mathbf{x}_i) \geq 0$, *for all $i \in [n]$.*

*Proof.* By the update rule of the coefficients, for $r \in \mathcal{S}_i^{(0)}$,

$$\overline{\rho}_{y_i,r,i}^{(t)} = \overline{\rho}_{y_i,r,i}^{(t-1)} - \frac{\eta}{nm}\ell_i'^{(t-1)}\|\boldsymbol{\xi}_i\|_2^2 \geq \overline{\rho}_{y_i,r,i}^{(t-1)} + 0.99\frac{\eta C_\ell \sigma_\xi^2 d}{nm} \geq 0.99\frac{\eta C_\ell \sigma_\xi^2 d}{nm}t,$$

where we use the lower bound on loss derivatives and Lemma D.5.

Then with

$$T_1 = 2.1\eta^{-1}nmC_\ell^{-1}\sigma_\xi^{-2}d^{-1}$$

we can show $\overline{\rho}_{y_i,r,i}^{(t)} \geq 2$. Further we can obtain the upper bound as for all $i \in [n], r \in [m], j = \pm 1$

$$\overline{\rho}_{j,r,i}^{(t)} = \overline{\rho}_{j,r,i}^{(t-1)} - \frac{\eta}{nm}\ell_i'^{(t-1)}\|\boldsymbol{\xi}_i\|_2^2 \leq \overline{\rho}_{j,r,i}^{(t-1)} + 1.01\frac{\eta\sigma_\xi^2 d}{nm} \leq 1.01\frac{\eta\sigma_\xi^2 d}{nm}t$$

where we use the upper bound on loss derivatives and Lemma D.5. Under the definition of $T_1$, for all $i, r, j$, we upper bound $\overline{\rho}_{j,r,i}^{(t)} \leq 3C_\ell^{-1}$.

Next for lower and upper bound for $\gamma_{j,r}^{(t)}$, we first recall the update as for any $j = \pm 1, r \in [m]$,

$$\gamma_{j,r}^{(t)} = \gamma_{j,r}^{(t)} - \frac{\eta}{nm}\sum_{i=1}^{n}\ell_i'^{(t-1)}\sigma'(\langle\mathbf{w}_{j,r}^{(t-1)}, y_i\boldsymbol{\mu}\rangle)\|\boldsymbol{\mu}\|_2^2.$$

When $\langle\mathbf{w}_{j,r}^{(t-1)}, \boldsymbol{\mu}\rangle \geq 0$,

$$\gamma_{j,r}^{(t)} = \gamma_{j,r}^{(t-1)} + \frac{\eta}{nm}\sum_{i\in\mathcal{S}_1}|\ell_i'^{(t-1)}|\|\boldsymbol{\mu}\|_2^2 \geq \gamma_{j,r}^{(t-1)} + \frac{\eta C_\ell\|\boldsymbol{\mu}\|_2^2}{2m} \geq \frac{\eta C_\ell\|\boldsymbol{\mu}\|_2^2}{2m}t$$

where we use the lower bound on $|\ell_i'^{(t-1)}|$. Similarly, when $\langle\mathbf{w}_{j,r}^{(t-1)}, \boldsymbol{\mu}\rangle \leq 0$, we can obtain the same lower bound as $\gamma_{j,r}^{(t)} \geq \frac{\eta C_\ell\|\boldsymbol{\mu}\|_2^2}{2m}$.

Then at $t = T_1$, we can bound for all $j = \pm 1, r \in [m]$ as

$$\gamma_{j,r}^{(t)} \geq \frac{n\|\boldsymbol{\mu}\|_2^2}{\sigma_\xi^2 d} = n \cdot \mathrm{SNR}^2 = \Omega(1)$$

The upper bound follows from

$$\gamma_{j,r}^{(t)} \leq \gamma_{j,r}^{(t-1)} + \frac{\eta\|\boldsymbol{\mu}\|_2^2}{2m} \leq \frac{\eta\|\boldsymbol{\mu}\|_2^2}{2m}t$$

where we apply the upper bound on $|\gamma_{j,r}^{(t-1)}| \leq 1$. This verifies that at $t = T_1$,

$$\gamma_{j,r}^{(t)} \leq 1.1C_\ell^{-1}n \cdot \mathrm{SNR}^2 = O(1),$$

which shows at $t = T_1$, both $\overline{\rho}_{y_i,r,i}^{(T_1)}, \gamma_{j,r}^{(T_1)} = \Theta(1)$.

For all samples, by Lemma E.1,

$$y_i f(\mathbf{W}^{(T_1)}, \mathbf{x}_i) \geq \frac{1}{m}\sum_{r=1}^{m}\left(\overline{\rho}_{y_i,r,i}^{(T_1)} + \gamma_{j,r}^{(T_1)}\right) - 1/C_1 \geq 0,$$

where we let $C_1$ to be sufficiently large. $\qquad\square$

## E.2 SECOND STAGE

In the second stage, we show the loss converges and under constant signal-to-noise ratio, we can show the test error can be arbitrarily small at convergence, while for all $T_1 \leq t \leq T^*$, $y_i f(\mathbf{W}^{(t)}, \mathbf{x}_i) \geq 0$, for all $i \in [n]$.

**Theorem E.2.** *Under Condition E.1, there exists a time $t^* \in [T_1, T^*]$ where $T^* = \widetilde{\Theta}(\eta^{-1}\epsilon^{-1}nm\sigma_\xi^{-2}d^{-1})$ such that*

- *Training loss converges, i.e., $L_S(\mathbf{W}^{(t^*)}) \leq \epsilon$.*

- *For all samples, i.e., $i \in [n]$, it satisfies $y_i f(\mathbf{W}^{(t)}, \mathbf{x}_i) \geq 0$.*

- *Test error is small, i.e., $L_D^{0-1}(\mathbf{W}^{(t^*)}) \leq \exp\left(\frac{d}{n} - \frac{n\|\boldsymbol{\mu}\|_2^4}{C_D\sigma_\xi^4 d}\right)$.*

*Proof of Theorem E.2.* The proof of convergence is exactly the same as for the case with label noise, and thus we omit it here. The second claim is also easy to verify given that both $\gamma_{j,r}^{(t)}, \overline{\rho}_{y_i,r,i}^{(t)}$ are monotonically increasing. By Lemma E.1, we can obtain the desired result.

Now we prove the third claim regarding the test error. To this end, we first show for all $T_1 \leq t \leq T^*$ $\gamma_{j,r}^{(t)}/\sum_{i=1}^n \overline{\rho}_{j,r,i}^{(t)} = \Theta(\text{SNR}^2)$ for all $j = \pm 1, r \in [m]$. We prove such a claim by induction. It is clear at $t = T_1$, we have $\sum_{i=1}^n \overline{\rho}_{j,r,i}^{(T_1)} = \Theta(n)$ and $\gamma_{j,r}^{(T_1)} = \Theta(n \cdot \text{SNR}^2)$ for all $j = \pm 1, r \in [m]$. Thus, we can verify the $\gamma_{j,r}^{(T_1)}/\sum_{i=1}^n \overline{\rho}_{j,r,i}^{(T_1)} = \Theta(\text{SNR}^2)$. Now suppose for a given $\widetilde{T} \in [T_1, T^*]$ such that $\gamma_{j,r}^{(t)}/\sum_{i=1}^n \overline{\rho}_{j,r,i}^{(t)} = \Theta(\text{SNR}^2)$ holds for all $T_1 \leq t \leq \widetilde{T} - 1$. Then according tor the update,

$$\sum_{i=1}^n \overline{\rho}_{j,r,i}^{(\widetilde{T})} = \sum_{i:y_i=j}^n \overline{\rho}_{j,r,i}^{(\widetilde{T})} = \sum_{i:y_i=j} \overline{\rho}_{j,r,i}^{(\widetilde{T}-1)} - \frac{\eta}{nm} \sum_{i \in \mathcal{S}_{j,r}^{(\widetilde{T}-1)}} \ell_i'^{(\widetilde{T}-1)} \|\boldsymbol{\xi}_i\|_2^2$$

$$\geq \sum_{i=1}^n \overline{\rho}_{j,r,i}^{(\widetilde{T}-1)} + 0.12\frac{\eta\sigma_\xi^2 d}{m} \min_{i\in[n]} |\ell_i'^{(\widetilde{T}-1)}|$$

where the second inequality is by $\mathcal{S}_{j,r}^{(0)} \subseteq \mathcal{S}_{j,r}^{(\widetilde{T}-1)}$ and Lemma C.3, Lemma D.5. Similarly, we can upper bound

$$\sum_{i=1}^n \overline{\rho}_{j,r,i}^{(\widetilde{T})} \leq \sum_{i=1}^n \overline{\rho}_{j,r,i}^{(\widetilde{T}-1)} + 1.01\frac{\eta\sigma_\xi^2 d}{m} \max_{i\in[n]} |\ell_i'^{(\widetilde{T}-1)}|$$

where we Lemma D.5.

On the other hand, we can lower and upper bound

$$\gamma_{j,r}^{(\widetilde{T})} \geq \gamma_{j,r}^{(\widetilde{T}-1)} + \frac{\eta}{2m} \min_{i\in\mathcal{S}_n} |\ell_i'^{(\widetilde{T}-1)}| \|\boldsymbol{\mu}\|_2^2$$

$$\gamma_{j,r}^{(\widetilde{T})} \leq \gamma_{j,r}^{(\widetilde{T}-1)} + \frac{\eta}{2m} \max_{i\in[n]} |\ell_i'^{(\widetilde{T}-1)}| \|\boldsymbol{\mu}\|_2^2$$

This suggests

$$\frac{\gamma_{j,r}^{(\widetilde{T})}}{\sum_{i=1}^n \overline{\rho}_{j,r,i}^{(\widetilde{T})}} \geq \min\left\{\frac{\gamma_{j,r}^{(\widetilde{T}-1)}}{\sum_{i=1}^n \overline{\rho}_{j,r,i}^{(\widetilde{T}-1)}}, \frac{\min_{i\in[n]} |\ell_i'^{(\widetilde{T}-1)}| \|\boldsymbol{\mu}\|_2^2}{2.02\max_{i\in[n]} |\ell_i'^{(\widetilde{T}-1)}|\sigma_\xi^2 d}\right\} \geq \min\left\{\frac{\gamma_{j,r}^{(\widetilde{T}-1)}}{\sum_{i=1}^n \overline{\rho}_{j,r,i}^{(\widetilde{T}-1)}}, \frac{\text{SNR}^2}{2.02\tilde{C}_\ell}\right\}$$

$$= \Omega(\text{SNR}^2)$$

where we use $\max_{i\in[n]} |\ell_i'^{(\widetilde{T}-1)}| \leq \tilde{C}_\ell \min_{i\in[n]} |\ell_i'^{(\widetilde{T}-1)}|$ by second claim of Lemma E.3. Similarly,

$$\frac{\gamma_{j,r}^{(\widetilde{T})}}{\sum_{i=1}^n \overline{\rho}_{j,r,i}^{(\widetilde{T})}} \leq \max\left\{\frac{\gamma_{j,r}^{(\widetilde{T}-1)}}{\sum_{i=1}^n \overline{\rho}_{j,r,i}^{(\widetilde{T}-1)}}, \frac{\tilde{C}_\ell\text{SNR}^2}{0.24}\right\} = O(\text{SNR}^2).$$

This verifies for all $T_1 \leq t \leq T^*$,

$$\gamma_{j,r}^{(t)} / \sum_{i=1}^{n} \overline{\rho}_{j,r,i}^{(t)} = \Theta(\text{SNR}^2). \tag{18}$$

Finally, we prove the test error can be upper bounded. We first write for a test sample $(\mathbf{x}, y) \sim \mathcal{D}_{\text{test}}$,

$$yf(\mathbf{W}^{(t^*)}, \mathbf{x}) = \frac{1}{m} \sum_{r=1}^{m} \left( \sigma(\langle \mathbf{w}_{y,r}^{(t^*)}, y\boldsymbol{\mu} \rangle) + \sigma(\langle \mathbf{w}_{y,r}^{(t^*)}, \boldsymbol{\xi} + \boldsymbol{\zeta} \rangle) \right)$$

$$- \frac{1}{m} \sum_{r=1}^{m} \left( \sigma(\mathbf{w}_{-y,r}^{(t^*)}, y\boldsymbol{\mu}) + \sigma(\langle \mathbf{w}_{-y,r}^{(t^*)}, \boldsymbol{\xi} + \boldsymbol{\zeta} \rangle) \right).$$

For $\langle \mathbf{w}_{y,r}^{(t^*)}, y\boldsymbol{\mu} \rangle$, we can bound

$$\langle \mathbf{w}_{y,r}^{(t^*)}, y\boldsymbol{\mu} \rangle = \gamma_{y,r}^{(t^*)} + \langle \mathbf{w}_{y,r}^{(0)}, y\boldsymbol{\mu} \rangle + \sum_{i=1}^{n} \frac{\langle \boldsymbol{\xi}_i, y\boldsymbol{\mu} \rangle}{\|\boldsymbol{\xi}_i\|_2^2} \overline{\rho}_{y,r,i}^{(t^*)} + \sum_{i=1}^{n} \frac{\langle \boldsymbol{\xi}_i, y\boldsymbol{\mu} \rangle}{\|\boldsymbol{\xi}_i\|_2^2} \underline{\rho}_{y,r,i}^{(t^*)}$$

$$\geq \gamma_{y,r}^{(t^*)} - \sqrt{2\log(12m/\delta)}\sigma_0 \|\boldsymbol{\mu}\|_2 - (\|\boldsymbol{\mu}\|_2 \sqrt{2\log(6n/\delta)}\sigma_\xi^{-1} d^{-1})\Theta(\text{SNR}^{-2})\gamma_{y,r}^{(t^*)}$$

$$\geq 0.99\gamma_{y,r}^{(t^*)}$$

where we use Lemma C.1 and Lemma C.2 and (18) in the second inequality. The last inequality follows from Condition E.1. With an similar argument, we can show

$$\langle \mathbf{w}_{-y,r}^{(t^*)}, y\boldsymbol{\mu} \rangle \leq -0.99\gamma_{-y,r}^{(t^*)}.$$

Further, we let $g(\boldsymbol{\zeta}) = \sum_{r=1}^{m} \sigma(\langle \mathbf{w}_{-y,r}^{(t^*)}, \boldsymbol{\zeta} + \boldsymbol{\xi} \rangle)$ and by (Vershynin, 2018, Theorem 5.2.2), we have for any $a \geq 0$,

$$\mathbb{P}\big( g(\boldsymbol{\zeta}) - \mathbb{E}g(\boldsymbol{\zeta}) > a \big) \leq \exp(-ca^2 \sigma_\xi^{-2} \|g\|_{\text{Lip}}^{-2})$$

where expectation is taken with respect to $\boldsymbol{\zeta} \sim \mathcal{N}(0, \sigma_\xi^2 \mathbf{I})$ and $c > 0$ is a constant. To compute the Lipschitz constant, we compute

$$|g(\boldsymbol{\zeta}) - g(\boldsymbol{\zeta}')| = \left| \sum_{r=1}^{m} \sigma(\langle \mathbf{w}_{-y,r}^{(t^*)}, \boldsymbol{\zeta} + \boldsymbol{\xi} \rangle) - \sum_{r=1}^{m} \sigma(\langle \mathbf{w}_{-y,r}^{(t^*)}, \boldsymbol{\zeta}' + \boldsymbol{\xi} \rangle) \right|$$

$$\leq \sum_{r=1}^{m} \left| \sigma(\langle \mathbf{w}_{-y,r}^{(t^*)}, \boldsymbol{\zeta} + \boldsymbol{\xi} \rangle) - \sigma(\langle \mathbf{w}_{-y,r}^{(t^*)}, \boldsymbol{\zeta}' + \boldsymbol{\xi} \rangle) \right|$$

$$\leq \sum_{r=1}^{m} |\langle \mathbf{w}_{-y,r}^{(t^*)}, \boldsymbol{\zeta} - \boldsymbol{\zeta}' \rangle| \leq \sum_{r=1}^{m} \|\mathbf{w}_{-y,r}^{(t^*)}\|_2 \|\boldsymbol{\zeta} - \boldsymbol{\zeta}'\|_2$$

where the first inequality is by triangle inequality and the second is by the property of ReLU function. This suggests $\|g\|_{\text{Lip}} \leq \sum_{r=1}^{m} \|\mathbf{w}_{-y,r}^{(t^*)}\|_2$.

In addition, because conditioned on $\boldsymbol{\xi}$, $\langle \mathbf{w}_{-y,r}^{(t^*)}, \boldsymbol{\zeta} + \boldsymbol{\xi} \rangle \sim \mathcal{N}\big( \langle \mathbf{w}_{-y,r}^{(t^*)}, \boldsymbol{\xi} \rangle, \|\mathbf{w}_{-y,r}^{(t^*)}\|_2^2 \sigma_\xi^2 \big)$.

$$\mathbb{E}g(\boldsymbol{\zeta}) = \sum_{r=1}^{m} \mathbb{E}\sigma(\langle \mathbf{w}_{-y,r}^{(t^*)}, \boldsymbol{\zeta} + \boldsymbol{\xi} \rangle)$$

$$= \sum_{r=1}^{m} \left( \langle \mathbf{w}_{-y,r}^{(t^*)}, \boldsymbol{\xi} \rangle \Big( 1 - \Phi\big( -\frac{\langle \mathbf{w}_{-y,r}^{(t^*)}, \boldsymbol{\xi} \rangle}{\|\mathbf{w}_{-y,r}^{(t^*)}\|_2 \sigma_\xi} \big) \Big) + \frac{\|\mathbf{w}_{-y,r}^{(t^*)}\|_2 \sigma_\xi}{\sqrt{2\pi}} \exp\big( -\frac{\langle \mathbf{w}_{-y,r}^{(t^*)}, \boldsymbol{\xi} \rangle^2}{2\|\mathbf{w}_{-y,r}^{(t^*)}\|_2^2 \sigma_\xi^2} \big) \right)$$

$$\leq \sum_{r=1}^{m} \left( \overline{\rho}_{-y,r,i^*}^{(t^*)} + \frac{\|\mathbf{w}_{-y,r}^{(t^*)}\|_2 \sigma_\xi}{\sqrt{2\pi}} \right)$$

where the expectation is taken with respect to $\boldsymbol{\zeta}$ and the last equality is due to Beauchamp (2018) on expectation of truncated Gaussian. The first inequality is by taking $\boldsymbol{\xi} = \boldsymbol{\xi}_{i^*}$ where $i^* = \arg\max_{i \in [n], r} \overline{\rho}_{-y,r,i}^{(t^*)}$ and use Condition E.1 to remove the leading constant.

Further, we require to bound $\|\mathbf{w}_{-y,r}^{(t^*)}\|_2$ by first bounding

$$\left\|\sum_{i=1}^{n} \rho_{j,r,i}^{(t^*)}\|\boldsymbol{\xi}_i\|_2^{-2} \cdot \boldsymbol{\xi}_i\right\|_2^2 = \sum_{i=1}^{n} \rho_{j,r,i}^{(t^*)2}\|\boldsymbol{\xi}_i\|_2^{-2} + 2\sum_{1\leq i<i'\leq n} \rho_{j,r,i}^{(t^*)}\rho_{j,r,i'}^{(t^*)} \frac{\langle\boldsymbol{\xi}_i, \boldsymbol{\xi}_{i'}\rangle}{\|\boldsymbol{\xi}_i\|_2^2\|\boldsymbol{\xi}_{i'}\|_2^2}$$

$$\leq 1.01\sigma_\xi^{-2}d^{-1}\sum_{i=1}^{n}\rho_{j,r,i}^{(t^*)2} + 4.08\frac{\sqrt{\log(6n^2/\delta)}}{\sigma_\xi^2 d^{3/2}}\sum_{1\leq i<i'\leq n}|\rho_{j,r,i}^{(t^*)}\rho_{j,r,i'}^{(t^*)}|$$

$$= 1.01\sigma_\xi^{-2}d^{-1}\sum_{i=1}^{n}\rho_{j,r,i}^{(t^*)2} + 4.08\frac{\sqrt{\log(6n^2/\delta)}}{\sigma_\xi^2 d^{3/2}}\left((\sum_{i=1}^{n}\rho_{j,r,i}^{(t^*)})^2 - \sum_{i=1}^{n}\rho_{j,r,i}^{(t^*)2}\right)$$

$$= \Theta(\sigma_\xi^{-2}d^{-1})\sum_{i=1}^{n}\rho_{j,r,i}^{(t^*)2} + \widetilde{\Theta}(\sigma_\xi^{-2}d^{-3/2})(\sum_{i=1}^{n}\rho_{j,r,i}^{(t^*)})^2$$

$$\leq \Theta(\sigma_\xi^{-2}d^{-1}n^{-1})(\sum_{i=1}^{n}\overline{\rho}_{j,r,i}^{(t^*)})^2$$

where the first inequality is by Lemma D.5 and Lemma C.1 and the last inequality is by the coefficient orders at $t^*$. Thus, we can bound

$$\|\mathbf{w}_{j,r}^{(t^*)}\|_2 \leq \|\mathbf{w}_{j,r}^{(0)}\|_2 + \gamma_{j,r}^{(t^*)}\|\boldsymbol{\mu}\|_2^{-1} + \Theta(\sigma_\xi^{-1}d^{-1/2}n^{-1/2})\sum_{i=1}^{n}\overline{\rho}_{j,r,i}^{(t^*)}$$

$$= \Theta(\sigma_0\sqrt{d}) + \Theta(\mathrm{SNR}^2\|\boldsymbol{\mu}\|_2^{-1} + \sigma_\xi^{-1}d^{-1/2}n^{-1/2})\sum_{i=1}^{n}\overline{\rho}_{j,r,i}^{(t^*)}$$

$$\leq \Theta(\sigma_\xi^{-1}d^{-1/2}n^{-1/2})\sum_{i=1}^{n}\overline{\rho}_{j,r,i}^{(t^*)},$$

where the first equality uses Lemma C.2 and (18). The second equality follows from $\sum_{i=1}^{n}\overline{\rho}_{j,r,i}^{(t^*)} = \Omega(n)$. The last inequality is by Condition E.1 where $\mathrm{SNR}^2\|\boldsymbol{\mu}\|_2^{-1}/(\sigma_\xi^{-1}d^{-1/2}n^{-1/2}) = \sqrt{n}\cdot\mathrm{SNR} = \Theta(1)$ and $\sigma_0\sqrt{d}/(\sigma_\xi^{-1}d^{-1/2}n^{-1/2}\sum_{i=1}^{n}\overline{\rho}_{j,r,i}^{(t^*)}) = O(\sigma_0\sigma_\xi dn^{-1/2}) = O(1)$ by condition on $\sigma_0$ in Condition E.1.

This gives

$$\frac{\sum_{r=1}^{m}\sigma(\langle\mathbf{w}_{y,r}^{(t^*)}, y\boldsymbol{\mu}\rangle)}{\mathbb{E}g(\boldsymbol{\zeta})} \geq \frac{\Theta(\sum_{r=1}^{m}\gamma_{y,r}^{(t^*)})}{\Theta(d^{-1/2}n^{-1/2})\sum_{r,i}\overline{\rho}_{-y,r,i}^{(t^*)} + \sum_{r=1}^{m}\overline{\rho}_{-y,r^*,i^*}^{(t^*)}} = \Theta(n\cdot\mathrm{SNR}^2)$$

Suppose $n\cdot\mathrm{SNR}^2 \geq C_+$ for $C_+ > 0$ sufficiently large. Then we have $\sum_{r=1}^{m}\sigma(\langle\mathbf{w}_{y,r}^{(t^*)}, y\boldsymbol{\mu}\rangle) - \mathbb{E}g(\boldsymbol{\zeta}) > 0$.

We bound the test error as

$$\mathbb{P}(yf(\mathbf{W}^{(t^*)}, \mathbf{x}) < 0)$$

$$\leq \mathbb{P}\left(\sum_{r=1}^{m}\sigma(\langle\mathbf{w}_{-y,r}^{(t^*)}, \boldsymbol{\xi} + \boldsymbol{\zeta}\rangle) \geq \sum_{r=1}^{m}\sigma(\langle\mathbf{w}_{y,r}^{(t^*)}, y\boldsymbol{\mu}\rangle)\right)$$

$$= \frac{1}{n}\sum_{i=1}^{n}\mathbb{P}\left(\sum_{r=1}^{m}\sigma(\langle\mathbf{w}_{-y,r}^{(t^*)}, \boldsymbol{\xi}_i + \boldsymbol{\zeta}\rangle) \geq \sum_{r=1}^{m}\sigma(\langle\mathbf{w}_{y,r}^{(t^*)}, y\boldsymbol{\mu}\rangle)\right)$$

$$= \frac{1}{n}\sum_{i=1}^{n}\mathbb{P}\left(g_i(\boldsymbol{\zeta}) - \mathbb{E}g_i(\boldsymbol{\zeta}) \geq \sum_{r=1}^{m}\sigma(\langle\mathbf{w}_{y,r}^{(t^*)}, y\boldsymbol{\mu}\rangle) - \mathbb{E}g_i(\boldsymbol{\zeta})\right)$$

$$\leq \frac{1}{n}\sum_{i=1}^{n}\exp\left(-\frac{c\left(\sum_{r=1}^{m}\sigma(\langle\mathbf{w}_{y,r}^{(t^*)}, y\boldsymbol{\mu}\rangle) - \sum_{r=1}^{m}\overline{\rho}_{-y,r,i}^{(t^*)} - \sigma_\xi/(2\pi)\sum_{r=1}^{m}\|\mathbf{w}_{-y,r}^{(t^*)}\|_2\right)^2}{\sigma_\xi^2(\sum_{r=1}^{m}\|\mathbf{w}_{y,r}^{(t^*)}\|_2)^2}\right)$$

$$\leq \frac{1}{n} \sum_{i=1}^{n} \exp\left(\frac{c}{2\pi} - 0.5c\left(\frac{\sum_{r=1}^{m} \sigma(\langle \mathbf{w}_{y,r}^{(t^*)}, y\boldsymbol{\mu}\rangle) - \sum_{r=1}^{m} \overline{\rho}_{-y,r,i}^{(t^*)}}{\sigma_\xi \sum_{r=1}^{m} \|\mathbf{w}_{y,r}^{(t^*)}\|_2}\right)^2\right)$$

$$= \frac{1}{n} \sum_{i=1}^{n} \exp\left(\frac{c}{2\pi} - 0.5c\left(\frac{\sum_{r=1}^{m} \Theta(\gamma_{y,r}^{(t^*)}) - \sum_{r=1}^{m} \overline{\rho}_{-y,r,i}^{(t^*)}}{\Theta(d^{-1/2}n^{-1/2}) \sum_{r=1}^{m} \sum_{i=1}^{n} \overline{\rho}_{j,r,i}^{(t^*)}}\right)^2\right)$$

$$\leq \frac{1}{n} \sum_{i=1}^{n} \exp\left(\frac{c}{2\pi} + \left(\frac{\sum_{r=1}^{m} \Theta(n^{-1}\mathrm{SNR}^{-2}\gamma_{y,r}^{(t^*)})}{\Theta(d^{-1/2}n^{-1/2}) \sum_{r=1}^{m} \sum_{i=1}^{n} \overline{\rho}_{j,r,i}^{(t^*)}}\right)^2 - 0.25c\left(\frac{\sum_{r=1}^{m} \Theta(\gamma_{y,r}^{(t^*)})}{\Theta(d^{-1/2}n^{-1/2}) \sum_{r=1}^{m} \sum_{i=1}^{n} \overline{\rho}_{j,r,i}^{(t^*)}}\right)^2\right)$$

$$= \exp\left(\frac{d}{n} - \frac{n\|\boldsymbol{\mu}\|_2^4}{C_D \sigma_\xi^4 d}\right)$$

where we denote $g_i(\boldsymbol{\zeta}) = \sum_{r=1}^{m} \sigma(\langle \mathbf{w}_{-y,r}^{(t^*)}, \boldsymbol{\zeta} + \boldsymbol{\xi}_i\rangle)$. The third and fourth inequalities are by $(s-t)^2 \geq s^2/2 - t^2$. $\qquad \square$

## F  EARLY STOPPING

*Proof of Proposition 4.1.* The proof follows the same idea as for Theorem E.2, with the difference that both $\sum_{r\in[m]} \gamma_{j,r}^{(T_1)} = \Theta(m)$ and $\sum_{r\in[m]} \overline{\rho}_{\tilde{y}_i,r,i}^{(T_1)} = \Theta(m)$ and $\sum_{r\in[m]} \gamma_{j,r}^{(T_1)} > \sum_{r\in[m]} \overline{\rho}_{\tilde{y}_i,r,i}^{(T_1)}$ for all $j = \pm 1$ and $i \in [n]$ (from the results of Theorem D.1). Then we can bound the test error directly as

$$\mathbb{P}(yf(\mathbf{W}^{(T_1)}, \mathbf{x}) < 0) \leq \frac{1}{n} \sum_{i=1}^{n} \exp\left(\frac{c}{2\pi} - 0.5c\left(\frac{\sum_{r=1}^{m} \sigma(\langle \mathbf{w}_{y,r}^{(T_1)}, y\boldsymbol{\mu}\rangle) - \sum_{r=1}^{m} \overline{\rho}_{-y,r,i}^{(T_1)}}{\sigma_\xi \sum_{r=1}^{m} \|\mathbf{w}_{y,r}^{(T_1)}\|_2}\right)^2\right)$$

$$\leq \frac{1}{n} \sum_{i=1}^{n} \exp\left(\frac{c}{2\pi} - 0.5c\left(\frac{1}{\Theta(d^{-1/2}n^{1/2})}\right)^2\right)$$

$$= \exp\left(\frac{c}{2\pi} - \Theta\left(\frac{d}{n}\right)\right)$$

$$\leq \exp(-\frac{d}{nC_e})$$

where the second inequality follows from $\sum_{r\in[m]} \gamma_{j,r}^{(T_1)} > \sum_{r\in[m]} \overline{\rho}_{\tilde{y}_i,r,i}^{(T_1)} = \Theta(m)$ and $\|\mathbf{w}_{y,r}^{(T_1)}\|_2 \leq \Theta(\sigma_\xi^{-1}d^{-1/2}n^{1/2})$ where $\sum_{i=1}^{n} \overline{\rho}_{j,r,i}^{(T_1)} \leq \Theta(n)$. The last inequality is by choosing a sufficiently large constant $C_e > 0$. $\qquad \square$

## G  ADDITIONAL EXPERIMENTS

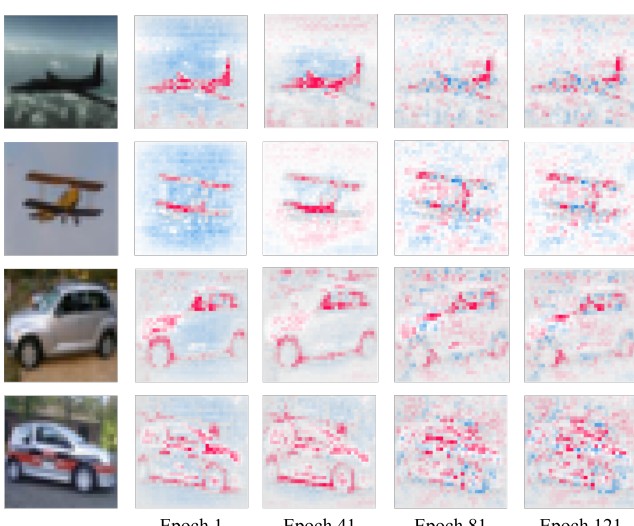

Epoch 1     Epoch 41     Epoch 81     Epoch 121

Figure 3: Visualization of model predictions (via SHAP) for noisy samples across multiple epochs. Red regions indicate positive contributions to model predictions, while blue regions denote negative contributions, with darker regions signifying greater contributions. It is evident that in the first stage (reflected by Epoch 1 and 41), model learns the generalizable features, such as the wings of "airplane" class and contours of "automobile" class. However, in the second stage (reflected by Epoch 81 and 121), the model overfits to partial features where the pattern of prediction is no longer visible.

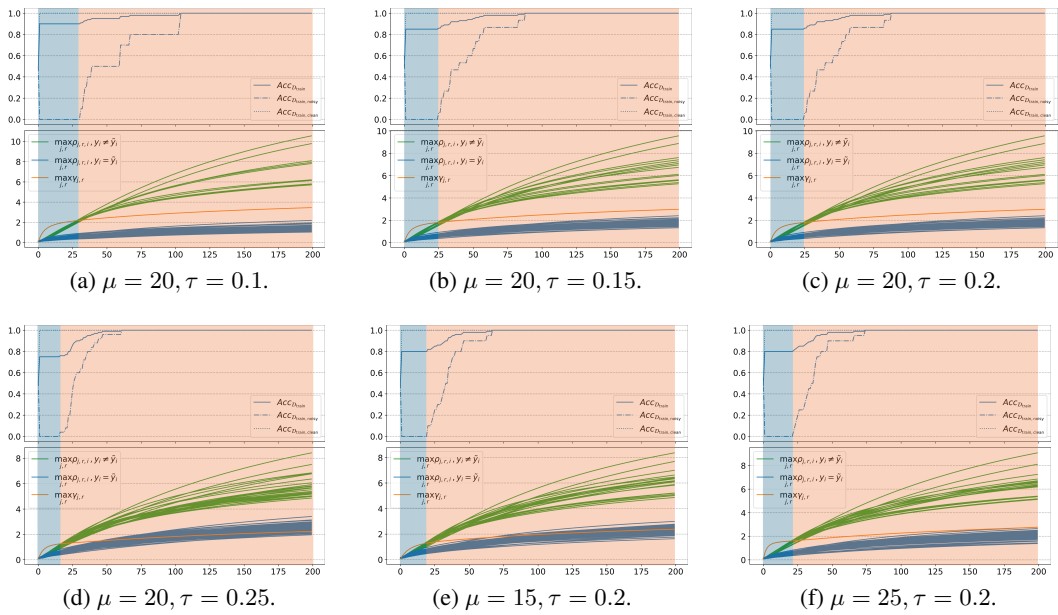

(a) $\mu = 20, \tau = 0.1$.     (b) $\mu = 20, \tau = 0.15$.     (c) $\mu = 20, \tau = 0.2$.

(d) $\mu = 20, \tau = 0.25$.     (e) $\mu = 15, \tau = 0.2$.     (f) $\mu = 25, \tau = 0.2$.

Figure 4: Experiments on synthetic data with varying problem settings, including varying signal strength $\mu$ and label noise ratio $\tau$. We shade the area before noise learning overtakes signal learning of noisy samples in blue. This corresponds to the Stage I in our analysis, where early stopping is beneficial. We shade the area where signal learning exceeds noise learning for noisy samples in orange, which corresponds to Stage II in our analysis.

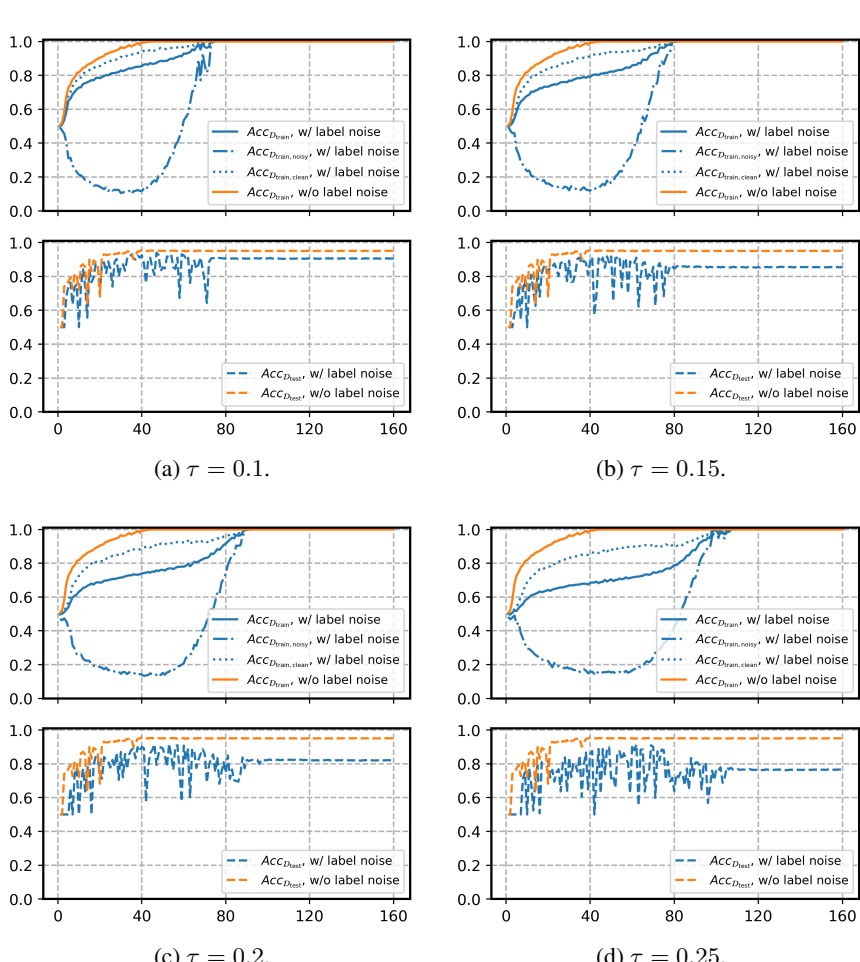

Figure 5: Experiments on CIFAR-10 dataset with varying label noise ratio $\tau$. Across different label noise ratios, we observe a similar pattern that there exist an initial decrease in the training accuracy on noisy samples before an increase to perfect classification. This validates our theoretical findings in real-world settings under various label noise ratios.