# OpenReview forum: "The Role of Label Noise in the Feature Learning Process"
_ICLR.cc/2025/Conference — Submitted to ICLR 2025_

### Official Review · Reviewer_AvJg · 2024-10-26

**Soundness:** 3
**Presentation:** 3
**Contribution:** 2
**Rating:** 5
**Confidence:** 4

**Summary:**

This paper proposes to theoretically analyze the learning behavior on data sets with label noise. Theoretical results on the case with label noise generally match existing empirical observations, and the authors have also conducted several experiments to verify their theoretical analysis.

**Strengths:**

- Theoretical results are generally sound and match empirical observations
- The paper is clearly written and easy to understand

**Weaknesses:**

The novelty seems limited: the analysis mostly follows the framework in (Kou et al. 2023) with modifications to settings with label noise

**Questions:**

- While the model setup in section 3 should generally follow (Kou et al. 2023), I am a bit curious on how this model can be realized, as it needs to distinguish signal and noise into different parts for activation in advance? I have checked (Kou et al. 2023) but did not find sufficient explanation, and the authors are encouraged to add some explanation for this point.
- Also in section 3, the authors mentioned that their analysis is applicable to class-conditional noise. However, I cannot find how this type of noise leads to different learning behaviors, and most quantities related to noise rates $\tau_+, \tau_-$ are symmetric on them. Is this really the case? If so, the authors can clarify this point to avoid possible misunderstandings.
- Note that current analysis focuses on binary classification only, which can be a bit restricted as the noise patterns under binary classification are much simpler: the positive samples can only be mislabeled to negative label, while the negative samples can only be mislabeled to positive label. As such, some discussion on whether the analysis framework can be generalized to multi-class classification is certainly welcome here.
- In addition, I wonder if current analysis can be applied to sample-dependent noise as well, where we can assume the noisy label $\tilde{y}_i$ has some dependencies on the sample noise $\xi$. Some discussion on this direction is also welcome.

---

> ### Author Response · Authors · 2024-11-22
> **Responses to Reviewer AvJg**
>
> Thank you for your great efforts on the review of this paper and for your constructive comments on our work. Below we address your questions and comments in details.
>
> ----
> **Q1**: *“The novelty seems limited: the analysis mostly follows the framework in (Kou et al. 2023) with modifications to settings with label noise”.*
>
> **A1**: First, we would like to highlight that the goal of this paper is to understand the feature learning dynamics in the presence of label noise, whereas [1] focused on characterizing the difference between benign and harmful overfitting. This necessitates significant changes in our theoretical setups compared to [1]. In particular, we emphasize that **both the change in label noise and in SNR are critical** that renders our settings significantly different compared to [1]. We have now added **Appendix B** that explicitly compares the differences in the proof techniques. In particular, [1] works under $n SNR^2 = o(1)$ while our setting requires $n SNR^2 = \Theta(1)$. Without the constant order of $n SNR^2$, either signal or noise dominates throughout all iterations and therefore, we cannot characterize the two-stage distinct behaviors of signal and noise learning.
>
> Such a change in SNR brings new challenges in the convergence and generalization analysis as we have discussed in Appendix B. Specifically, many of the key techniques developed in [2] are no longer applicable in our case, which includes time-invariant of coefficients $\gamma_{j,r}^{(t)}/\sum_{i} \overline{\rho}_{j,r,i}^{(t)} = \Theta(SNR^2)$ and automatic balancing of updates $\ell_i’^{(t)}/\ell_k’^{(t)} \leq C$. These techniques only hold under $n SNR^2 = o(1)$ where signal learning is negligible compared to noise learning and thus implies the learning patterns across all samples are similar. In our case, to characterize the two-stage behaviors, we need to distinguish the learning dynamics of clean and noisy samples, which requires to develop novel proof techniques. We believe our contributions relative to [1] is non-trivial given our new settings and results.
>
> ----
> **Q2**: *Add some explanation on the model setup. "How this model can be realized, as it needs to distinguish signal and noise into different parts for activation in advance?"*
>
> **A2**: Thank you for the question. The synthetic setup is designed to better characterize the feature learning dynamics, enabling us to separately analyze the evolution of signal and noise learning. However, this does not imply that the model must distinguish signal and noise beforehand. The patch positions are irrelevant due to the use of shared weights, which ensures that the model treats all patches uniformly during training.
>
> ----
> **Q3**: *On the class-conditional noise.*
>
> **A3**: Thank you for pointing this out. In the current setting, the class-conditional noise indeed does not lead to different learning behaviors. We have added a remark in Page 6 of the revised version.
>
> ----
> **Q4**: *Generalization to multi-class classification. “Note that current analysis focuses on binary classification only,.... As such, some discussion on whether the analysis framework can be generalized to multi-class classification is certainly welcome here”.*
>
> **A4**: We are glad to discuss more on our theoretical setup.
>
> * **Clarification of our choice of binary classification**: First, we would like to clarify that the current binary classification setting is sufficient to illustrate the role of label noise in the feature learning process of neural networks while maintaining mathematical tractability. While we agree that generalizing to multi-class classification is an interesting direction, it goes beyond the primary focus of this paper and we will leave it for future research.
>
> * **Possibility of generalization to multi-class classification**: Yes, we believe the current framework can be generalized to handle multiple classes with additional setups. That is, we assume each class is represented by a distinct signal vector, i.e., $\mu_1, \mu_2, …, \mu_k$ for $k$ classes, where the labels are encoded as one-hot vectors. The signal vectors are required to be orthogonal such that the current analysis can be naturally adapted to multi-class settings.
>
> ----
> **Q5**: *Application to sample-dependent noise.*
>
> **A5**: Thank you for your insightful question. Yes, we believe that the feature learning framework we adopted in this paper can be applied to sample-dependent label noise. However, a more rigorous analysis would require a more comprehensive definition of such dependence. This necessitates a careful reformation of the original problem, such as by introducing multiple features and might need to develop new proof techniques, which lies beyond the scope of this paper and is left for future research.
>
> ----
> **Reference**
>
> [1] Kou, Yiwen et al. Benign Overfitting in Two-layer ReLU Convolutional Neural Networks. ICML 2023.

---

> > ### Comment · Reviewer_AvJg · 2024-11-25
> >
> > Thank you for your responses that do clarify some of my previous concerns (e.g., the symmetry about class-conditional noise). Nevertheless, I am afraid there are still several issues that make this submission slightly below the acceptance bar:
> > - I have noted that two other reviewers share the same concerns as me on the novelty of this work comparing to previous results like (Kou et al., 2023). After checking your responses, shall I assume that your analysis distinguish clean samples and noisy samples, and the analysis on clean samples should directly follow existing works? That does not sound like a breakthrough novelty, though it should certainly be useful as it leads to some results that match empirical observations.
> > - While you have also mentioned many times that the two-stage results are insightful, it does not sound like a big surprise as there are already many empirical works justifying such observations, and even some more empirical works brought other different observations. It might be better if the theoretical analysis can bring other possibility for empirical observations (that is why I asked about multi-class setting), though I understand it may be too difficult.

---

> > > ### Author Response · Authors · 2024-11-26
> > >
> > > Thank you for your prompt reply. We are glad that some of your previous concerns have been addressed by our responses. Regarding the remaining issues on the technical novelty and significance of our work, we would like to offer further clarifications:
> > >
> > > >  After checking your responses, shall I assume that your analysis distinguish clean samples and noisy samples, and the analysis on clean samples should directly follow existing works? That does not sound like a breakthrough novelty, though it should certainly be useful as it leads to some results that match empirical observations.
> > >
> > > We thank the reviewer for carefully reviewing our responses and for recognizing the value of our work in aligning theoretical results with empirical observations. Regarding your question, we would like to clarify that it is *incorrect* to assume our analyses with label noise directly extend existing works on clean samples.
> > >
> > > *Intuitively*,  training with label noise introduces complex interactions between optimizing on clean and noisy samples, leading to a fundamentally different training regime compared to training solely on clean data. Analyzing the whole dynamics with label noise requires addressing the interplay between these two forces, which inherently poses unique challenges.
> > >
> > > *Technically*, we are working under a significantly different training settings, due to the $nSNR^2=\Theta(1)$ rather than $nSNR^2=o(1)$, as well as our different condition on the label noise probabilities. These differences are critical and leads to the key techniques in prior analysis on clean samples ineffective for our analyses on both clean and noisy samples. Addressing these challenges has required us to develop novel approaches to capture the dynamics with both clean and noisy samples.
> > >
> > > > While you have also mentioned many times that the two-stage results are insightful, it does not sound like a big surprise as there are already many empirical works justifying such observations, and even some more empirical works brought other different observations. It might be better if the theoretical analysis can bring other possibility for empirical observations (that is why I asked about multi-class setting), though I understand it may be too difficult.
> > >
> > > A good question. First of all, we would like to clarify the value of theoretical contributions. While empirical investigations have explored the behavior of neural networks under label noise, theory can explain *why* certain empirical phenomenon arises, offering a *deeper* understanding beyond the observation alone. Simplified theoretical setups also encourage *abstraction*, allowing researchers to isolate key factors and uncover principles that inspire new approaches, algorithms, or techniques.
> > >
> > > In our case, our analysis indeed offers novel and significant understandings beyond empirical observations, along with practical implications:
> > >
> > > - **Early signal learning**: We prove that in the early training phase, models learn signal faster than noise from both clean and noisy samples. This supports early stopping as an effective strategy to prevent fitting noisy samples.
> > > - **Late-stage noise dominance**: We also prove that after sufficient training iterations, models inevitably learn noise from noisy samples faster than signal, while clean samples remain dominated by signal learning. This highlights the importance of targeted noisy sample elimination to mitigate overfitting.
> > >
> > > These insights not only align with existing empirical observations but also offer clear guidance for future research, paving the way for more effective techniques to address the label noise problem.
> > >
> > > **More importantly**, compared to existing theoretical works on label noise (listed in related work section), our theoretical analysis provides a rich and *trackable* view of training dynamics *without* relatively unrealistic assumptions, along with *sufficient* practical insights. We consider this a significant and surprising contribution in itself.
> > >
> > > We hope our explanation clarifies the novelty and significance of our work.

---

> ### Author Response · Authors · 2024-11-25
> **Looking forward your feedback**
>
> Dear Reviewer AvJg,
>
> We hope that our responses could adequately address your concerns. As the deadline of this discussion phase is approaching, we warmly welcome further discussion regarding any additional concerns that you may have, and we sincerely hope you can reconsider the rating accordingly.
>
> Thank you for the time and appreciation that you have dedicated to our work.
>
> Best regards,
>
> Authors of submission 3468

---

### Official Review · Reviewer_N78m · 2024-11-01

**Soundness:** 3
**Presentation:** 3
**Contribution:** 2
**Rating:** 6
**Confidence:** 2

**Summary:**

This work examines the learning dynamics of a 2-layer convolutional network with ReLU activations on a signal-noise data distribution from prior work, with the change that the training data sampled from this distribution has flipped labels with non-trivial probability. The main result is a characterization of the dynamics that shows that there are two stages: one in which the model first fits the "clean" data (no label noise) and then a second stage in which the model fits the noisy data.

**Strengths:**

**Significance:** Non-trivial amounts of label noise in training data is a common occurrence, and this work contributes to our theoretical understanding of how model training works on such data.

**Quality:** This is a mostly theoretical work that is conducted in a well-known setting; the theory seems sound and I did not see any glaring issues (although I did not carefully check details of the calculations in the appendix).

**Clarity:** The paper is overall well-written, and the core proof ideas receive sufficient exposition in the main body of the paper.

**Originality:** Although this work reuses the data model of earlier work [1, 2], it increases the allowable label noise probability of [2].

[1] Cao, Yuan et al. “Benign Overfitting in Two-layer Convolutional Neural Networks.” ArXiv abs/2202.06526 (2022).
[2] Kou, Yiwen et al. “Benign Overfitting in Two-layer ReLU Convolutional Neural Networks.” International Conference on Machine Learning (2023).

**Weaknesses:**

The main weakness I see with this work is its contribution relative to the previous work of Kou et al. (2023). In particular, I outline my concerns regarding the changes in the theory below (along with a few concerns regarding the experiments).

- **Differences in Setting:** The main differences between the setting in this paper and that of Kou et al. (2023) are that the label noise probabilities $\tau^+, \tau^-$ can be arbitrarily large up to the threshold of not completely destroying the signal ($1/2$), and that the SNR for the setting in this paper needs to be $n * SNR^2 = \Theta(1)$. I view the first change as the main contribution of the paper, however,
I feel the second change needs further justification.
The authors claim that this change corresponds to a milder constraint on $d$, but it is not clear to me that this is the case given it could just correspond to a higher signal setting. I agree this would follow if the lower bound on $d$ were only in terms of $||\mu|| \sigma_{\xi}^{-1}$, but the first part of the lower bound in Condition 4.1 does not make sense to me -- the definition of $T^*$ involves $d$ (and if we fix this, we run into the issue I mentioned). Overall, my concern here is that the higher label noise is being counterbalanced by higher signal. It would be very helpful if the authors included something like Tables 1 and 2 from Appendix A of Kou et al. (2023) to have a more granular comparison to the settings of prior work, and justify each change.

- **Differences in Proofs:** Once again, the key difference in the proofs in this paper versus Kou et al. (2023) is handling the different SNR constraint; it would be helpful to provide more exposition regarding the key technique in Kou et al. (2023) to make it clearer how the new proof differs. Also, the authors state that the "signal coefficients are on the same order as noise coefficients", which again goes back to my concern regarding the change to the SNR setting. This does not seem like an extension of the Kou et al. (2023) setting but rather a different setting where we have balanced greater signal strength with label noise.

- **Takeaways:** The main takeaway from the results in this paper is that, because of the two-stage dynamics, early stopping makes sense when training with significant label noise. This is a useful perspective, but it is hard to conclude from the theory itself because we are working in the asymptotic setting. Further experiments investigating early stopping and its relation to the two-stage dynamics would be useful.

- **Experiments:** Minor comments -- neither Figures 1 nor 2 are easy to read by themselves; it would be helpful to either break up the plots more (both figures plot several curves which leads to a lot of visual clutter, at least train/test should be broken up in Figure 2) or provide more textual exposition regarding what's being plotted so that the reader does not have to refer back to the definitions repeatedly. More importantly, I feel it would be useful to have more practical experiments focusing on the two-stage dynamics similar to the synthetic ones. Right now, Figure 2 relies on comparing the accuracies on the noisy/non-noisy samples for a single choice of label noise parameter; it would at the very least be useful to verify the phenomena for a wider range of label noise parameters since this should fall within the theory.

In summary, while I think considering the setting of higher label noise is useful, I have non-trivial concerns with how the theoretical setting in this work differs from that of Kou et al. (2023); if the authors can appropriately address these I would be amenable to increasing my score.

**Questions:**

- It appears that unlike in prior work, the signal patch here is always fixed to be $x^{(1)}$ -- is this correct? If so, why is this necessary?

---

> ### Author Response · Authors · 2024-11-22
> **Responses to Reviewer N78m (Part 1)**
>
> Thank you for your great efforts on the review of this paper and for your constructive comments on our work. Below we address your questions and comments in details.
>
> ----
> **Q1**: *On the contributions relative to [2]. Differences in settings and differences in proofs.*
>
> **A1**: First, we emphasize that the goal of this paper is to understand the feature learning dynamics in the presence of label noise, whereas [2] focused on characterizing the difference between benign and harmful overfitting. This necessitates significant changes in our theoretical setups compared to [2]. Specifically, without the condition that $n SNR^2 = \Theta(1)$, either the signal or noise dominates throughout all training stages, making it infeasible to prove the two-stage behavior. This results in a fundamentally different regime compared to [2], which requires novel proof techniques and refined analysis. Below we discuss differences in settings and proof techniques in details.
>
> * **Differences in setting**: We have added *Table 2 in Appendix A*, explicitly comparing the conditions to [2]. The differences lie in conditions in *label noise ratio*, *SNR* and *dimensionality*. The change in label noise $\tau$ is to ensure the effect of label noise can be visibly characterized. The change in SNR is required to derive the two-stage behaviors of signal and noise learning. The Change in dimensionality is partially due to the change in SNR, as we no longer require $d$ to be sufficiently large for $n SNR^2 = o(1)$, as in [2].  Specifically, the conditions on $d$ differ by $||\mu||/\sigma_\xi$ up to some logarithmic factors. We agree that it may be inappropriate to claim we have a milder condition on $d$, without quantifying the scale of $||\mu||$ and $\sigma_\xi$. To avoid confusion, we have removed the claim on “mild condition” in the revised version.
>
> * **Differences in proofs**: We are encouraged that the reviewer acknowledges that our theoretical setting is significantly different from [2] where the signal learning is always dominated by noise learning. In our case, signal and noise learning are on the same scale due to the constant order $n SNR^2$. Consequently, the key proof techniques employed by [2]--such as time-invariant of coefficients $\gamma_{j,r}^{(t)}/\sum_{i} \overline{\rho}_{j,r,i}^{(t)} = \Theta(SNR^2)$ and automatic balancing of updates $\ell_i’^{(t)}/\ell_k’^{(t)} \leq C$–are no longer valid in our case. In our setting, it is necessary to separately analyze the behaviors of clean and noisy samples in two stages, as balanced behaviors across all samples are not observed. To address this, we developed novel proof techniques tailored for such cases, resulting in a non-trivial and refined analysis. We have included **Appendix A** and **Appendix B** in the revised manuscript to explicitly contrast our work with [2] in terms of the required conditions and proof techniques.
>
> ----
> **Q2**: *“Further experiments investigating early stopping and its relation to the two-stage dynamics would be useful.”*
>
> **A2**: Thank you for the suggestion. To verify the relation between early stopping and the two-stage behavior, we have conducted **new experiments** (see **experiments in Appendix G of the revised manuscript**) to investigate the temporal correlation between the early stopping point and the end point of Stage I. We consider various signal strength and label noise ratio. Across all the settings, we see a clear match between the stages of signal dominating noise learning and zero training accuracy on noisy samples. This strongly suggests that the early stopping point could be reliably identified by the transition from Stage I to Stage II.
>
> ----
> **Q3**: *Separate train/test accuracy in Figure 2.*
>
> **A3**: Thank you for the suggestion. We have now separated the train and test accuracy in Figure 2 to improve readability.
>
> ----
> **Q4**: *Practical experiments focusing on the two-stage dynamics.*
>
> **A4**: To demonstrate the two-stage behavior in practical scenarios, we have added **new interpretations** (see Appendix G) of model predictions (via SHAP) on the CIFAR-10 dataset, across various model checkpoints. The purpose is to illustrate two-stage behavior for noisy samples throughout the training dynamics. We have plotted the results for epoch 1, 41, 81, 121 on noisy samples, i.e., samples with label flip. The figures clearly show that in the first stage (represented by epoch 1, 41), the model learns generalizable features, such as wings of airplanes and contours of automobiles. This stage corresponds to the stage where signal dominates noise learning. In contrast, for the second stage (represented by epoch 81, 121), the model overfits to partial features and the model interpretation via SHAP does not show a clear pattern of the objects.

---

> > ### Author Response · Authors · 2024-11-22
> > **Responses to Reviewer N78m (Part 2)**
> >
> > **Q5**: *Experiments with a wider range of label noise parameters.*
> >
> > **A5**: Thank you for the suggestion. We have now added additional experiments with various label noise ratios for both *synthetic* and *real-world* datasets in **Appendix G of the revised manuscript**. We consider label noise ratio $\tau = 0.1, 0.15, 0.2, 0.25$ for classifying both synthetic data and CIFAR-10 images.
> >
> > * In the **synthetic** setup, we see the two-stage behaviors (signal dominates noise learning in the first stage and noise dominate signal in the second stage for noisy samples) hold for all tested label noise noise ratios.
> >
> > * For the **real-world** setup, we also observe similar two-stage behaviors: In the first stage, training accuracy of noisy samples first decreases before increasing to 1 while the training accuracy is consistently high for clean samples. This suggests the model first fits clean samples while overfits noisy samples in the second stage.
> >
> > These results suggest our findings in this paper hold for a wider range of label noise ratios in both synthetic and real-world settings.
> >
> > ----
> > **Q6**: *Signal patch always fixed to $x^{(1)}$.*
> >
> > **A6**: It is *not necessary* to fix the signal patch to be $x^{(1)}$, as the position of the signal/noise does not affect the analysis when using the CNN network with shared weights for both patches. We have revised the definition on Page 3 of the revised manuscript by stating “one of $x^{(1)}$, $x^{(2)}$ is $y \mu$ and the other is $\xi$”.
> >
> > ----
> > **Reference**
> >
> > [1] Cao, Yuan et al. Benign Overfitting in Two-layer Convolutional Neural Networks. NeurIPS 2022.
> >
> > [2] Kou, Yiwen et al. Benign Overfitting in Two-layer ReLU Convolutional Neural Networks. ICML 2023.

---

> ### Comment · Reviewer_N78m · 2024-11-23
>
> Thank you for the detailed response and revisions. Appendices A and B in the revision are very helpful and definitely clarify the differences with the prior work more than before. The additional experiments are also encouraging. Overall I do think there are some interesting ideas in this work and the authors have addressed my main questions; I have thus updated my rating from 5 $\to$ 6. However, I do feel in agreement with the other reviewers that the contribution in terms of novelty is low relative to the prior work, and I do not feel strongly in favor of acceptance so I have updated my confidence to a 2 to reflect this. Given the borderline nature of the other reviews, I will try my best to do a deeper dive into the technical details of both this work and that of Kou et al. (2023) and come back to update my confidence rating accordingly.
>
> Minor comment: it would be very helpful to make the font size in the plots bigger.

---

> > ### Author Response · Authors · 2024-11-24
> >
> > Thank you for the quick response and for raising your rating accordingly. We appreciate the opportunity to further elaborate on the novelty of our work. Apart from the technical novelty, which we have sufficiently discussed in A1, we would like to highlight several novel and significant insights conveyed by our analysis:
> >
> > - **Insight from First stage**: Models learn signal faster than noise from both clean and noisy samples. This supports early stopping as an effective strategy to prevent fitting noisy samples.
> > - **Insight from Second stage**: Models start learning noise from noisy samples faster than signal, while clean samples remain dominated by signal learning. This highlights the importance of targeted noisy sample elimination to mitigate overfitting.
> >
> > These insights provide a theoretical foundation for developing more effective methods to address label noise. Furthermore, we would like to emphasize the significance of our work.
> >
> > - **Going beyond existing literature**: Unlike most existing theoretical and empirical studies, which rarely address the learning dynamics of over-parameterized neural networks in the presence of label noise, our work characterizes the underlying mechanism of overfitting in such settings. Specifically, we demonstrate that models first fit clean samples in the first stage, then progressively fit noisy samples in the second stage, ultimately leading to overfitting to all samples. This detailed understanding fills a significant gap in the literature.
> > - **Boarder impact on deep learning theory**: More importantly, our work demonstrate the value of formulating theory from realistic nontrivial phenomena that provide valuable insights into the underlying mechanisms of deep learning. In particular, our theory explains the non-trivial phenomena observed in previous studies on label noise, such as the memorization effect (Han et al. 2018).
> >
> > Additionally, we will enlarge the font size in our plots in future revisions.

---

### Official Review · Reviewer_eDxb · 2024-11-09

**Soundness:** 2
**Presentation:** 2
**Contribution:** 2
**Rating:** 5
**Confidence:** 2

**Summary:**

The paper analyses generalisation error of gradient descent with label noise in a binary classification problem setting where the input is split into two patches, with only the first patch being dependent on the class label and the second patch being random noise. The results mainly apply to the setting where the input norm in a noisy patch is much larger than the signal patch and where the number of training points is small relative to the dimension of an input patch. The model is a simple one hidden layer MLP, which is applied separately to both patches and added.


Under this setting the paper goes on to show that the first layer weights pick up the signal component of the data first during training even in the presence of label noise. The mislabelled datapoints are fit by the training algorithm in the second phase which corresponds to overfitting. This motivates the ideas of early stopping during training and outlier identification via large loss.

**Strengths:**

The paper provides reasonably well written theory and experiments to support a commonly used intuitive heuristic of early stopping and outlier identification. The paper is organised well and is mostly easy to follow.

**Weaknesses:**

I am not fully familiar with the papers (Cao et al. and Kou et al.) which is the setting this paper also operates in. The setting is extremely simplistic and the final conclusion is not surprising while it may still be technically challenging to prove. It is not clear the extra contribution over those papers is significant enough.

There are a few other technical issues which could be errors in the Theorems/proofs even if the overall message of the paper is correct.

The setting essentially corresponds to linearly separable data with label noise. (the hyper plane with a normal [\mu, \0]  separates the data perfectly). The model is more convoluted and assumes that you know the split of the data x into the noise component and the signal component. This seems very artificial and unrealistic.

From the way $\overline \rho$ and $\underline \rho$ are defined, they seem like symmetric objects with flipped signs. So lines 230 to 233 (and other places) confuse me by stating that $\overline \rho $ keeps increasing, but $\underline \rho$ is bounded below.

**Questions:**

Please justify this setting of linearly separable data with an unnecessarily complex model in a self contained manner.

The argument effectively says that as the training progresses, $w_{j,r}$ only has positive components along the noise points $\xi_i$. Why is that?  i.e why is $\overline \rho$ eventually larger than $\underline \rho$?

---

> ### Author Response · Authors · 2024-11-22
> **Responses to Reviewer eDxb (Part 1)**
>
> Thank you for your great efforts in reviewing our paper and for your constructive comments on our work. Below we address your questions and comments in detail.
>
> ----
> **Q1**: *“The setting is extremely simplistic. The setting essentially corresponds to linearly separable data with label noise... The model is more convoluted and assumes that you know the split of the data x into the noise component and the signal component. This seems very artificial and unrealistic. Please justify this setting of linearly separable data with an unnecessarily complex model in a self contained manner.”*
>
> **R1**: Thank you for your question.
> * **Clarification of our goal**: First, we would like to emphasize that the goal of this paper is to understand the *feature learning dynamics of neural networks in the presence of label noise*. The use of non-linear and two-layer model is to simulate the behavior of neural networks in practice, in contrast to linear models such as logistic regression.
>
> * **Common practice in theoretical analyses**: In addition, the setting of linearly separable data and two-layer CNN model has been widely adopted for analyzing classification models [1,2,3,4,5]. This simplified data model effectively captures the essential aspects of training dynamics and the generalization ability of neural networks [1,6], while ensuring mathematical tractability.
>
> * **Non-trivial theoretical analyses**: Despite the simplified theoretical setup, significant challenges remain. The non-convex nature of the objective function, combined with the complex dynamics introduced by label noise, makes it highly non-trivial to precisely characterize the entire training dynamics and generalization behavior.
>
> * **Comparison to prior works**: We also highlight that compared to prior theoretical works on label noise, our settings more closely reflect the real-world scenarios. As we have discussed in the related work section, existing studies either assumed lazy training, or focused on the early-stage dynamics, failing to capture the entire training process as comprehensively as demonstrated in this paper.
>
> * **Extensive empirical evidence**: Moreover, the alignment between our real-world experiments and the theoretical results supports the validity of our theoretical setting. This suggests that our framework effectively captures the key factors in characterizing the effects of label noise in more realistic scenarios.

---

> ### Author Response · Authors · 2024-11-22
> **Responses to Reviewer eDxb (Part 2)**
>
> **Q2**: *“The final conclusion is not surprising… It is not clear the extra contribution over those papers is significant enough.”*
>
> **R2**: We think the reviewer’s assertion that our contribution is not surprising/significant is  rather subjective and insufficiently justified. To begin with, we would like to emphasize that the goal of this paper is to understand the feature learning process with label noise, while “those papers” [1, 2] focused on the benign overfitting phenomenon. This distinguishes our different yet significant contributions from others. To further clarify, we highlight our contributions in the following aspects:
>
> **(1) Technical contributions**: Compared to existing theoretical studies [1,2], many of our theoretical settings are notably different. For example, [1] considered an activation function of $(ReLU(\cdot))^q$ with $q \gg 3$ and ignored label noise in the data setup.
>
> [2] is more relevant to our work.  However, [2] assumed $n SNR^2$ to be sufficiently small. Thus, they failed to capture the distinct two-stage behaviors caused by label noise and cannot characterize the adverse effect of label noise on generalization. In contrast, once $n SNR^2$ is of constant order, we can rigorously prove the two-stage behavior induced by label noise. This behavior is more aligned with empirical evidence, where the memorization effect emerges and the generalization ability degrades.
>
> It is worth noting that such significantly *different settings* require **new proof techniques and refined analysis**. Because the key proof techniques of [2], including time-invariant coefficients $\gamma_{j,r}^{(t)}/\sum_{i} \overline{\rho}_{j,r,i}^{(t)} = \Theta(SNR^2)$ and automatic balancing of updates $\ell_i’^{(t)}/\ell_k’^{(t)} \leq C$ are no longer applicable. Consequently, we developed novel methods to analyze clean and noisy samples separately, which are critical to capturing the unique effects of label noise in this regime. In addition, we have listed all the key quantities that are different to [2] in two stages, including the monotonicity of signal learning, signal-to-noise-learning ratio, sample-wise prediction and test error.
>
> We have now added **Appendix A and B in the revised manuscript** to explicitly compare the differences in conditions and proof techniques with [2].
>
> **(2) Valuable insights into practice**: Other than rigorous theoretical contributions, our work offers valuable insights into practice. Specifically, we show that the adverse effect of label noise arises because noise learning overtakes signal learning for noisy samples. This insight not only explains the (in)effectiveness of strategies for addressing label noise, such as early stopping and sample selection, but also lays a theoretical foundation for developing more effective techniques. We believe our analysis offers valuable guidance for future research in mitigating label noise.
>
> ----
> **Q3**: *On other technical issues.*
>
> **R3**: After thoroughly reviewing our paper, we have not identified any technical issues with the theorem or its proof. We would greatly appreciate it if the reviewer could specify the potential issues identified so that we can provide further clarification.
>
> ----
> **Q4**: *On the explanation of $\overline \rho$ and $\underline \rho$.*
>
> **R4**: The dynamics of $\overline \rho$ and $\underline \rho$ are *not symmetric*. This is due to ReLU activation, where the positive part of the inner product keeps increasing while the negative part cannot decrease significantly before it becomes deactivated by ReLU. This eventually results in $\overline \rho$ being significantly larger than $\underline \rho$ in magnitude.
>
> ----
> **Reference**:
>
> [1] Cao et al. Benign overfitting in two-layer convolutional neural networks. NeurIPS 2022.
>
> [2] Kou et al. Benign overfitting in two-layer ReLU convolutional neural networks. ICML 2023.
>
> [3] Meng et al. Benign overfitting in two-layer ReLU convolutional neural networks for XOR data. ICML 2024.
>
> [4] Chen et al. Understanding and improving feature learning for out-of-distribution generalization. NeurIPS 2023.
>
> [5] Chen et al. Why does sharpness-aware minimization generalize better than sgd? NeurIPS 2023.
>
> [6] Allen-Zhu & Li. Towards understanding ensemble, knowledge distillation and self-distillation in deep learning. ICLR 2023.

---

> > ### Comment · Reviewer_eDxb · 2024-11-24
> > **Reply to Rebuttal:**
> >
> > Thanks for your response.
> >
> > 1. The clarification that one of x_1 or x_2 is $y\mu$ helps a little towards justifying the setting. Now the setting is technically not linearly separable. (e.g. if d=1 and \mu=1 we have two dimensional data along the lines x_1=+1, x_1=-1, x_2=+1 and x_2 = -1 with data from lines x_1 = +1 and x_2 = +1 being positive and the data from other two lines are negative.
> >
> >
> > 2. The argument regarding the asymmetry of $\overline \rho$ and $\underline \rho$ still eludes me. Maybe I am missing something. Why should the inner product of $w_{j,r}^t$ with $\xi_i$ be biased towards any sign? Shouldn't this inner product be symmetric around 0 at init, and stay that way on average because of symmetry? As a thought experiment, consider the data where only the $\xi_i$ are negated. Technically, this data has the same distribution as $\xi$ are symmetric about $0$ anyway, but it should have opposite behaviour on the dynamics of $\overline \rho$ and $\underline \rho$.
> >
> > 3. I do agree that the proof techniques are likely quite intricate, but I am not sure that the message is significant/novel enough for a strong accept.

---

> > > ### Author Response · Authors · 2024-11-24
> > > **Responses to Reviewer eDxb**
> > >
> > > Thank you for the quick reply! Here are our further responses to your comments.
> > >
> > > > The clarification that one of $x_1$ or $x_2$ is $y\mu$ helps. Now the setting is technically not linearly separable.
> > >
> > > Yes, we thank the reviewer for recognizing the technical challenge lied in our theoretical setup.
> > >
> > >
> > > > The argument regarding the asymmetry of $\overline{\rho}$ and $\underline{\rho}$ still eludes me.
> > >
> > > We are glad to discuss more on the technical details.
> > >
> > > The reason that sign of $\langle w^{(t)}_{j,r}, \xi_i \rangle$ is asymmetric is due to the ReLU based activation applied. In particular, the iteration of $\overline{\rho}$ and $\underline{\rho}$ are given by
> > >
> > >  $$\overline{\rho}\_{j,r,i}^{(t+1)}   = \overline{\rho}\_{j,r,i}^{(t)} - \frac{\eta}{nm}   {\ell}\_i'^{(t)}   \sigma' (  \langle w\_{j,r}^{(t)},   \xi\_{i}   \rangle  )      || {\xi}\_i ||_2^2   {1}({\tilde{y}\_{i} = j})$$
> > >
> > > $$\underline{\rho}\_{j,r,i}^{(t+1)}   = \underline{\rho}\_{j,r,i}^{(t)} + \frac{\eta}{nm} {\ell}\_i'^{(t)}    \sigma' (   \langle w\_{j,r}^{(t)},   {{\xi}}\_{i}   \rangle   || {\xi}\_i ||_2^2  {1}({\tilde{y}\_{i} = -j})$$
> > >
> > > The key term is $\sigma' (  \langle {w}\_{j,r}^{(t)},   {{\xi}}\_{i}   \rangle  )$ where the sign of $\langle {w}\_{j,r}^{(t)},   {{\xi}}\_{i}   \rangle$ determines whether $\sigma' (  \langle {w}\_{j,r}^{(t)},   {{\xi}}\_{i}   \rangle  )$ is zero or one. From our analysis, $\langle {w}\_{j,r}^{(t)},   {{\xi}}\_{i}   \rangle$ is almost equivalent to $\underline{\rho}_{j,r,i}^{(t)}$ when $\tilde{y}_i = -j$. Once $\underline{\rho}\_{j,r,i}^{(t)} \le 0$, its magnitude cannot be large, otherwise deactivating ReLU activation.
> > >
> > > We lastly highlight that the although the initialization is symmetric, the ReLU activation only selects the neuron where $\langle {w}\_{j,r}^{(0)},   {{\xi}}\_{i}   \rangle > 0$. This already breaks the symmetry.
> > >
> > > We hope this addresses your question on the asymmetry. Please let us know if you require further clarifications.
> > >
> > >
> > >
> > > > I do agree that the proof techniques are likely quite intricate, but I am not sure that the message is significant/novel enough for a strong accept.
> > >
> > > Thank you for acknowledging the complexity of our theoretical analyses. We appreciate the opportunity to clarify and further elaborate on the significance and novelty of our work to enhance your understanding.
> > >
> > > Apart from the technical novelty of our work (which we have sufficiently addressed), we would like to highlight several novel and significant messages conveyed by our analysis. These insights  have been rigorously proved and empirically observed in our paper, and they contribute meaningfully to the broader understanding of neural network learning dynamics in the presence of label noise.
> > >
> > > 1. **Insights from the first stage**: Under the current setting, models learn signal faster than noise of both clean and noisy samples in the *first stage*. This observation underscores the effectiveness of early stopping as a strategy for addressing label noise, by stopping the model before fitting noisy samples.
> > >
> > >
> > > 2. **Insights from the second stage**: In the *second stage*, models begin to learn noise of the noisy samples faster than the signal. In contrast, for clean samples, the signal continues to dominate the learning process over noise. This implies that the model will unavoidably overfit to the noisy samples, causing the irreversible damage to the model performance. This insight highlights the importance of eliminating noisy samples through targeted sample selection to mitigate overfitting.
> > >
> > >
> > > 3. **Solid theoretical analysis beyond existing literature**: Unlike most existing theoretical and empirical studies, which rarely address the learning dynamics of over-parameterized neural networks in the presence of label noise, our work characterizes the underlying mechanism of overfitting in such settings. Specifically, we demonstrate that models first fit clean samples in the first stage, then progressively fit noisy samples in the second stage, ultimately leading to overfitting to all samples. This detailed understanding fills a significant gap in the literature.
> > >
> > > 4. **Broader impact and contribution**: Despite the empirical success of deep neural networks, theoretical understanding of their behaviors remains limited. Just like in many other scientific disciplines, a crucial step towards formulating a comprehensive theory of deep learning lies in meticulous empirical investigations of the learning pipeline, intending to uncover quantitative and reproducible nontrivial phenomena that shed light on the underlying mechanisms. Our work contributes to this effort by analyzing the training dynamics and feature learning processes of deep neural networks under the presence of label noise, shedding light on nontrivial mechanisms underlying their behavior. These findings offer meaningful directions for improving model robustness and understanding the interplay between noise and over-parameterization.

---

> > > > ### Comment · Reviewer_eDxb · 2024-11-24
> > > >
> > > > Thanks for the reply.
> > > >
> > > > I have a last doubt below regarding the  sign of $(\overline \rho^t_{j,r,i} - \underline \rho^t_{j,r,i} )$.
> > > >
> > > > I guess my main confusion is with the implication of Equation 3.
> > > >
> > > > The vector $w^t_{j,r}$ component along $\xi_i$ is $(\overline \rho^t_{j,r,i} - \underline \rho^t_{j,r,i} )$.
> > > >
> > > > The term in paranthesis is claimed to be positive. Which means that for all $j,r,i$ the vector $w^t_{j,r}$ will have positive component along $\xi_i$. Why should this be the case? I can understand if this were the case only for $i,j$ such that $\tilde y_i = j$. But equation 3 applies for all $i,j$.

---

> ### Author Response · Authors · 2024-11-25
> **Further responses**
>
> Thank you for the prompt reply. We are glad that our responses have resolved your other questions.
>
> Regarding your last doubt on the sign of $(\overline{\rho} - \underline{\rho})$, we first emphasize that from the defition of $\overline{\rho}$ and $\underline{\rho}$ (in line 209), *only one* of $\overline{\rho}\_{j,r,i}$ and $\underline{\rho}\_{j,r,i}$ is nonzero for any $j,r,i$. Therefore, $\langle w\_{j,r}, \xi_i \rangle \approx \overline{\rho}\_{j,r,i}$ for $\tilde y_i = j$ and $\langle w\_{j,r}, \xi\_i \rangle \approx \underline{\rho}\_{j,r,i}$ for $\tilde y_i \neq j$. Consequently, the sign of $\langle w_{j,r}, \xi_i \rangle$ is determined by either $\overline{\rho}\_{j,r,i}$ or $\underline{\rho}\_{j,r,i}$, *not both*. This implies that not all the inner products are positive even though equation 3 holds for all $j,r,i$.
>
> We hope this has clarified your doubt. We are happy to elaborate if needed. If no further questions remain, we would be greatly appreciated if you could consider re-evaluating our work.

---

### Meta-Review · Area_Chair_AfHU · 2024-12-17

**Metareview:**

This paper builds on the model of Kou et al. by introducing significant label noise. The main result shows that with label noise training exhibits a two-stage behavior, where the first stage the model fits all the clean samples, and the second stage the training loss converges to 0 (overfitting on noisy samples). The theoretical results supports early stopping and sample selection for handling label noise. While the reviewers find the paper well-written and the theoretical results match empirical observations, there are concerns about the novelty on top of Kou et al., and the fact that the theory does not give new predictions/algorithms. A way to improve the paper would be to motivate an early stopping or sample selection criterion based on the theory that could be applied to non-synthetic settings.

**Additional Comments On Reviewer Discussion:**

A reviewer was not active during the author discussion period but participated in the reviewer discussion period. Overall the reviewers were not fully convinced by author response and agreed that the paper needs a bit more novelty/significance (see suggestion in meta-review).

---

### Decision · Program_Chairs · 2025-01-22

Reject